# Strategies for feeding the world more sustainably with organic agriculture

Adrian Muller[1,2], Christian Schader[1], Nadia El-Hage Scialabba[3], Judith Brüggemann[1], Anne Isensee[1], Karl-Heinz Erb [4], Pete Smith[5], Peter Klocke[1,6], Florian Leiber[1], Matthias Stolze[1] & Urs Niggli[1]

Organic agriculture is proposed as a promising approach to achieving sustainable food systems, but its feasibility is also contested. We use a food systems model that addresses agronomic characteristics of organic agriculture to analyze the role that organic agriculture could play in sustainable food systems. Here we show that a 100% conversion to organic agriculture needs more land than conventional agriculture but reduces N-surplus and pesticide use. However, in combination with reductions of food wastage and food-competing feed from arable land, with correspondingly reduced production and consumption of animal products, land use under organic agriculture remains below the reference scenario. Other indicators such as greenhouse gas emissions also improve, but adequate nitrogen supply is challenging. Besides focusing on production, sustainable food systems need to address waste, crop–grass–livestock interdependencies and human consumption. None of the corresponding strategies needs full implementation and their combined partial implementation delivers a more sustainable food future.

[1] Research Institute of Organic Agriculture (FiBL), Ackerstrasse 113, 5070 Frick, Switzerland. [2] Institute of Environmental Decisions, Department of Environmental Systems Science, ETH Zürich, 8092 Zürich, Switzerland. [3] Food and Agriculture Organization of the United Nations (FAO), Viale Terme di Caracalla, 00153 Rome, Italy. [4] Institute of Social Ecology Vienna (SEC), Alpen-Adria University Klagenfurt-Vienna-Graz, Schottenfeldgasse 29, 1070 Wien, Austria. [5] Institute of Biological and Environmental Sciences, University of Aberdeen, 23 St Machar Drive, AB24 3UU Aberdeen, UK. [6] bovicare GmbH, Hermannswerder Haus 14, 14473 Potsdam, Germany. Correspondence and requests for materials should be addressed to A.M. (email: adrian.mueller@fibl.org)

Intensification of agriculture has greatly increased food availability over recent decades. However, this has led to considerable adverse environmental impacts, such as increases in reactive nitrogen over-supply, eutrophication of land and water bodies, greenhouse gas (GHG) emissions and biodiversity losses[1–6]. It is commonly assumed that by 2050, agricultural output will have to further increase by 50% to feed the projected global population of over 9 billion[7]. This challenge is further exacerbated by changing dietary patterns. It is, therefore, crucial to curb the negative environmental impacts of agriculture, while ensuring that the same quantity of food can be delivered. There are many proposals for achieving this goal, such as further increasing efficiency in production and resource use, or adopting holistic approaches such as agroecology and organic production, or reducing consumption of animal products and food wastage[8–11].

Organic agriculture is one concrete, but controversial, suggestion for improving the sustainability of food systems. It refrains from using synthetic fertilizers and pesticides, promotes crop rotations and focuses on soil fertility and closed nutrient cycles[4, 12]. The positive performance of organic agriculture when measured against a range of environmental indicators has been widely reported[13–16]. However, organic systems produce lower yields[17] and thus require larger land areas to produce the same output as conventional production systems. In consequence, environmental benefits of organic agriculture are less pronounced or even absent if measured per unit of product than per unit of area[14, 18]. Furthermore, abandoning synthetic N-fertilizers could lead to nutrient undersupply, even with increased legume cropping[19]. As a consequence, the ability of organic agriculture to feed the world sustainably has been challenged[19, 20]. Some authors contribute to the discussion on lower yields in organic agriculture by considering nutrient availability, but none of these provide a robust analysis of nutrient availability in organic production systems[19–21]. In addition, these studies do not pursue a detailed food systems approach, and do not address the role that animal feeding regimes, consumption trends and food wastage (i.e. food loss and waste) may play—all of which represent factors for

strategies that could substantially reduce land demand, while alleviating environmental impacts and contributing to global food availability[2, 10, 22–26].

We address this research gap by taking a food systems approach that goes beyond a focus on production, yields and environmental impacts per unit output of specific commodities. We first investigate the impacts of a conversion to organic agriculture on a range of environmental and production indicators. We then complement this scenario of organic conversion with two additional changes to the food system, namely (a) reductions of livestock feed from arable land (i.e. food-competing feed) with corresponding reductions in animal numbers and products supply (and thus human consumption) and in related natural resource use and environmental impacts[25, 26]; and (b) reductions of food wastage, with correspondingly reduced production levels and impacts[10]. Our leading research question is whether producing a certain total amount of food, in terms of protein and calories, with organic agriculture would lead to higher, or lower, impacts than producing the same amount of food with conventional agriculture. We then assess whether, and to which extent a combination of organic agriculture with the two other strategies mentioned above may contribute to mitigating potential adverse effects of a conversion to organic production. We thus assess the contexts in terms of complementary food system changes in which a conversion to organic agriculture may contribute to more sustainable food systems.

Despite the availability of a number of global models to assess various aspects of food production and consumption, few are able to consider organic production[22, 27] and so far, none have captured the main agronomic characteristics of organic agriculture in a systematic way. We apply the SOL-model[26] which is able to simulate important aspects of organic agriculture, such as increased legume shares, absence of synthetic fertilizers, lower yields (the 'yield gap') and lower use of food-competing feed components, such as grain legumes or cereals. The SOL-model is a mass-flow model of the global food system, which is built to cover physical and biological aspects at country level for a large number of commodities, thus allowing assessment and

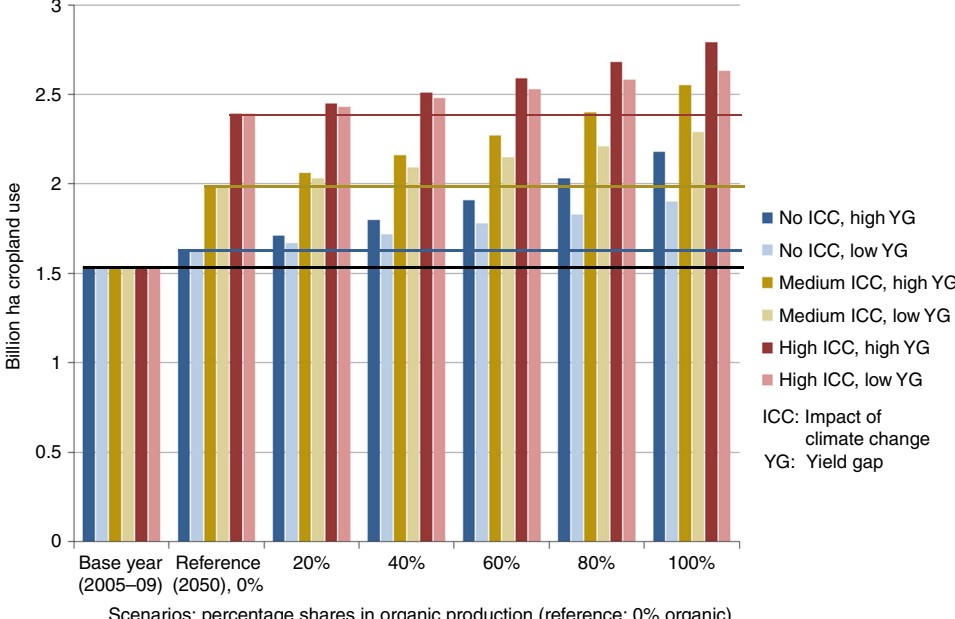

**Fig. 1** Cropland occupation. Cropland occupation (billion ha) for the base year (average 2005–09), the reference scenario 2050 (0% organic) and scenarios with increasing percentages of organic production. Displays scenarios with low and high yield gaps[17, 21] without, with medium and with full impacts of climate change on yields (no/medium/high ICC)

**Fig. 2** Cropland area change. Percentage change in cropland areas with respect to the reference scenario. Scenarios differ in: organic shares (0–100%), impacts of climate change on yields (low, medium, high), food-competing feed reductions (0, 50, 100% reduced from the levels in the reference scenario), and wastage reduction (0, 25, 50% compared to the reference scenario). Colour code for comparison to the reference scenario value (i.e. 0% organic agriculture, no changes in livestock feed and food waste, dotted grey): > +5%: red, < −5% blue, between −5% and +5% yellow; in the reference scenario, cropland areas are 6% higher than in the baseline today

comparison of the physical viability and impacts of different scenarios. The SOL-model explicitly does not cover decisions of farmers and consumers, and price and market effects in an economic sense. The purpose is rather to examine the option space spanned by combining a number of food-system level strategies for increased sustainability, and to assess the potential and contribution of these strategies, and their combinations, towards increased sustainability in food systems. This assessment is undertaken with and without impacts of climate change on yields, to also assess the performance of these strategies under climate change (we refer to the Methods section for further details).

Our results show that adoption of organic agriculture by itself increases land demand with respect to conventional production, but it has advantages in terms of other indicators, such as reduced nitrogen surplus, and pesticide use. But when combined with complementary changes in the global food system, namely changed feeding rations, and correspondingly reduced animal numbers, and changed wastage patterns, organic agriculture can contribute to feeding more than 9 billion people in 2050, and do so sustainably. Such a combination of strategies can deliver adequate global food availability, with positive outcomes across all assessed environmental indicators, including cropland area demand.

Our analysis shows the necessary food system changes at the global level, but we emphasize that structural change in the food system and the pathways that lead to increasing the proportion of organically produced food will differ regionally, so local and regional characteristics need to be accounted for.

## Results

**Feasibility of organic agriculture.** Compared to the base year (calculated using the average of 2005–2009 data; Methods section), cropland occupation increases by 6% in the 2050 reference scenario (which describes agriculture as forecast by the FAO, adopting their assumptions on yield increase, cropping intensities and regional dietary change, and, implicitly, via their production and consumption structure, on underlying elasticities)[7]. Switching to 100% organic production leads to further increases in land use: 16–33%, for low yield gaps (8% lower organic yields on average) to high yield gaps (on average 25% lower), as reported in the literature[17, 21]. Land occupation increases further, if adverse effects of climate change (CC) on yields (modelled by reduced yield increases until 2050, down to zero increases for strong CC impacts) are considered (up to +55% for zero organic, 71–81% for 100% organic, compared to the base year; Fig. 1). The differences in land occupation between scenarios with low and high organic yield gaps decrease with increasing CC impact, as the absolute differences in yields due to the yield gap becomes less with increasing CC impact and thus generally lower yields. Deforestation shows similar patterns to land occupation with 8–15% higher values for 100% organic in comparison to the reference in 2050, depending on assumptions of low or high yield gaps (Supplementary Fig. 9). Deforestation is modelled as the pressure on forests from increased land demand, assuming the same relative deforestation rates, i.e. ha-deforested per-ha cropland increase, in each country as reported in the baseline (using deforestation data from FAOSTAT; Methods section). This likely underestimates deforestation impacts for larger cropland increases, given that additional cropland will largely be sourced from forests, as grasslands are assumed to stay constant. Thus, the land occupation and deforestation indicators as used here serve to assess the pressure on land areas and forests that may arise from the dynamics captured in the different scenarios.

We modelled scenarios that combine conversion to organic production with other systematic interventions, namely the reduction of animal feed grown on arable land and a

| % Wastage reduction | % Reduction in food-competing feed | Zero (% Organic) | | | | | | Medium (% Organic) | | | | | | High (% Organic) | | | | | |
|---|---|---|---|---|---|---|---|---|---|---|---|---|---|---|---|---|---|---|---|
| | | 0 | 20 | 40 | 60 | 80 | 100 | 0 | 20 | 40 | 60 | 80 | 100 | 0 | 20 | 40 | 60 | 80 | 100 |
| 0 | 0 | 25 | 21 | 15 | 10 | 4 | −3 | 23 | 19 | 14 | 8 | 3 | −3 | 21 | 16 | 12 | 7 | 1 | −4 |
| 0 | 50 | 20 | 16 | 12 | 7 | 2 | −4 | 18 | 14 | 10 | 6 | 1 | −4 | 17 | 13 | 9 | 4 | 0 | −5 |
| 0 | 100 | 15 | 11 | 7 | 3 | −1 | −5 | 13 | 10 | 7 | 3 | −1 | −5 | 12 | 9 | 5 | 2 | −2 | −6 |
| 25 | 0 | 23 | 19 | 14 | 8 | 2 | −4 | 21 | 17 | 12 | 7 | 1 | −4 | 19 | 15 | 10 | 5 | 0 | −5 |
| 25 | 50 | 18 | 14 | 10 | 6 | 1 | −5 | 17 | 13 | 9 | 5 | 0 | −5 | 15 | 11 | 7 | 3 | −1 | −5 |
| 25 | 100 | 13 | 10 | 6 | 2 | −2 | −6 | 12 | 9 | 5 | 2 | −2 | −6 | 11 | 8 | 4 | 1 | −3 | −6 |
| 50 | 0 | 21 | 17 | 12 | 7 | 1 | −5 | 19 | 15 | 10 | 5 | 0 | −6 | 17 | 13 | 9 | 4 | −1 | −6 |
| 50 | 50 | 16 | 12 | 8 | 4 | 0 | −6 | 15 | 11 | 7 | 3 | −1 | −6 | 14 | 10 | 6 | 2 | −2 | −6 |
| 50 | 100 | 11 | 8 | 5 | 1 | −3 | −7 | 10 | 7 | 4 | 1 | −3 | −7 | 10 | 7 | 3 | 0 | −3 | −7 |

**Fig. 3** Nitrogen balance. N-surplus (positive values) or deficit (negative values) in kg N/ha. Scenarios differ in: organic shares (0–100%), climate change impacts (low, medium, high), food-competing feed reductions (0, 50, 100% reduced from the levels in the reference scenario), and wastage reduction (0, 25, 50% compared to the reference scenario). Colour code for comparison to the reference scenario value (i.e. 0% organic agriculture, no changes in livestock feed and food waste, dotted grey): >10 kg/ha: red (unsustainably high), between 10 kg/ha and 5 kg/ha blue (optimum, reduction from current average surplus by 60–80%,[59, 60]), between 5 kg/ha and −2 kg /ha yellow (critical, rather low), < −2 kg/ha orange (deficit)

corresponding reduction in animal numbers and production[26], and the reduction of food wastage. As a stand-alone measure, no more than 20% conversion to organic production would be possible if increases in land demand beyond 5% of the land demand in the reference scenario are to be avoided (no impacts of CC (ICC) assumed), and a conversion to 100% organic production without complementary measures would lead to huge land demand increases. Due to the yield gap, fully conventional production will always need less land than if a part of the production is organic, but this is of less importance, in terms of overall sustainability, if the complementary measures are implemented. A partial conversion to organic production (e.g. 40% with 100% reduction of food-competing feed components; medium ICC assumed), and for certain cases even a full conversion (e.g. with 50% food wastage reduction and 100% reduction of food-competing feed components; medium ICC), becomes viable, with equal or even reduced land demand compared to the reference scenario (Fig. 2). Similarly, although land demand would be lower with zero organic agriculture, production systems with positive shares of organic agriculture perform better with respect to a number of other environmental indicators (Figs. 3 and 5 below). To provide a conservative analysis of the potential for organic agriculture, these results are based on high assumed yield gaps for organic agriculture[17]. The results for low yield gaps and lower CC impact on yields for organic than for conventional production (Methods section) are provided in Supplementary Figs. 1–15. We emphasize that grassland areas are held constant in all scenarios, but animal numbers and livestock production decrease in response to reduced food-competing feed supply, and ensuing cropland demand decreases as it is no longer used for feed production. As a consequence of reduced production, consumption of animal products is also reduced.

The N-surplus acts as a proxy for oversupply of reactive nitrogen to ecosystems and related impacts. It is equal to N-inputs minus N outputs, and covers all N flows, including fertilizer inputs and biological fixation, as well as product outputs, emissions and leaching (Methods section). Due to N inputs from reduced mineral fertilizers and substitution by increased legume shares, the N-surplus is reduced with increasing shares of organic production, and reaches a balanced level at an organic share of 80%. It flips to a deficit of −15 to −35% compared to the base year with 100% conversion (Fig. 3). This reduction in N-surplus needs to be considered in the context of where nutrients are sourced and how they are recycled in organic agriculture: Farm yard manure, crop residues (e.g. roots, litter, compost) and nitrogen fixation (via legumes in the crop rotations) are the only sources of nitrogen in organic systems in the scenarios, because synthetic N-fertilizers are prohibited, and food and human waste is not used as fertilizer in the model, nor widespread in reality. This leads to a corresponding reduction of N-availability in the organic system, which is only partly offset by increased biological fixation.

Conversion to organic agriculture thus reduces the contribution of agriculture to the disruption of the nitrogen cycle. However, for high global conversion rates to organic agriculture, N-supply is likely to become challenging, even if food-competing feed and wastage shares are reduced (Fig. 3). Thus, additional measures are needed to ensure adequate N-supply on croplands. Potential measures include optimizing legume management, recycling nutrients from various organic wastes and increasing nutrient use efficiency. Note that in particular the utilization of food and human waste holds substantial further potentials to increase N supply[28–30], but was not modelled in SOLm. In other studies, N-supply has been assessed with optimistic assumptions on N-fixation rates and off-season cover crop potential for legumes[21], which has been contested[19, 20].

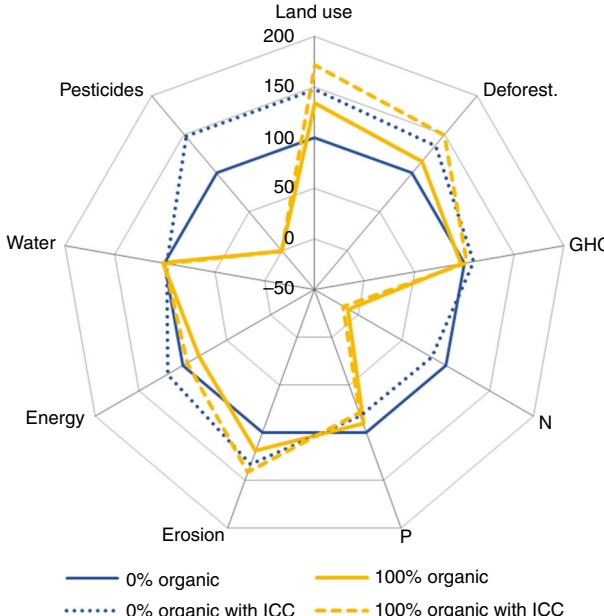

**Fig. 4** Year 2050 environmental impacts of a full conversion to organic agriculture. Environmental impacts of organic scenarios (100% organic agriculture, yellow lines) are shown relative to the reference scenario (0% organic agriculture, blue lines), with (dotted lines) and without (solid lines) impacts of climate change on yields; Calories are kept constant for all scenarios. Indicators displayed: cropland use, deforestation, GHG emissions (incl. deforestation, organic soils), N-surplus and P-surplus, water use, non-renewable energy use, soil erosion, pesticide use

We also emphasize that N-surplus values displayed here are global per-ha averages, including grasslands. They therefore overshadow regional variation and are only adequate as an indicator of impacts on the global nutrient cycle and not for assessing local nutrient supply. Furthermore, atmospheric N-deposition is not included[28]. Including it would increase N-surplus, and N-deficits would be less pronounced.

Overall, the results show that, for example, a food system with a combination of 60% organic production, 50% less food-competing feed and 50% reduced food wastage would need little additional land (Fig. 2) and have an acceptable N-supply (Fig. 3) when medium CC impacts on yields are assumed. When assuming low yield gaps and lower climate change impacts on yields for organic than for conventional agriculture, the viability of high shares of organic agriculture regarding land use correspondingly becomes more viable, while adequate N-supply becomes slightly more challenging, which is due to a relative decline of N-availability from crop residues and N-fixation in relation to yields (Supplementary Figs. 1–4).

**Dietary implications of the different scenarios.** We illustrate how the different food system strategies addressed in this paper may influence the consumption side by analyzing the dietary consumption in the different scenarios. All scenarios discussed here fulfil the condition of providing the same amount of calories as the reference scenario (only corrected accordingly when food wastage is reduced). In the model, legume shares are increased to 20% of the cropping areas in organic production systems. This leads to a slight change in dietary composition (shares in protein supply; Supplementary Fig. 5) and an increased protein/calorie ratio (i.e. the share of calories provided from protein) for full organic production of 12%, compared to 10.9% in the reference

scenario. This is above the minimum level of 10% recommended by the Food and Nutrition Board of the US National Academy of Sciences[31]. The shares of animal products decrease from 38% in the reference scenario to 36% for 100% organic, as the additional legumes substitute meat. Generally, higher or lower impacts of climate change on yields, or high and low yield gaps, do not substantially affect human diets according to our model (Supplementary Figs. 6–8).

With a reduction in food-competing feed, dietary composition changes considerably. The share of animal products in total protein supply drops from 38 to 11% with 100% reduction in food-competing feed[26]. For 100% reduction in food-competing feed, driven by the lower animal numbers, the model increases legume area shares by up to 20% for all production systems to compensate for the loss in animal proteins. Therefore, increasing shares of organic production do not further increase legume area shares, which are already at 20% of total cropland, but leads to lower yields. This explains the decreasing legume shares in diets with increasing organic production shown in Supplementary Figs. 5–8. The role of legumes also shows that scenarios with reduced food-competing feed and scenarios with increasing organic shares ideally complement each other. Increasing legume shares are needed to compensate for decreasing animal protein supply (food-competing feed reduction scenarios) and to assure nitrogen supply (organic scenarios). The effects of climate change and yield gaps on diets are also much smaller than the impact of the level of reducing food-competing feed (Supplementary Figs. 5–8). This is due to the scenario definitions that stay as close as possible to the reference scenario, including relative commodity shares. This also applies to legume shares for which the effects of climate change and yield gaps are much smaller than the impact of the level of food-competing feed.

**Environmental impacts.** A 100% conversion to organic agriculture would lead to reduced impacts for a range of other environmental indicators besides the ones already discussed above (Fig. 4). An exception is the soil erosion potential, which increases by 10–20%, compared to the reference scenario (i.e. a 20–30% increase if compared to the base year; ranges relate to the effects with and without ICC). This is due to the increased land area under organic production and the conservative assumption of similar soil erosion rates under organic and conventional production. P-surplus remains at almost the same level as in the reference situation, due to the assumption that organic systems operate with similar levels of non-renewable P inputs as conventional systems. This is a conservative estimate, because soil-available P, and P from organic inputs is often taken into account by organic producers when deciding on fertilization levels. Due to lack of data, we do not model this. With respect to non-renewable energy demand, a 19–27% decrease can be achieved (mainly due to synthetic fertilizer reduction, and due to differences in energy use as reported in the Ecoinvent 2.0 database), compared to the reference situation (i.e. a 4–14% decrease if compared to the base year). Even GHG emissions can be somewhat reduced with this strategy, by 3–7% compared to the reference scenario if emissions from deforestation and organic soils are included, but still representing an increase of 8–12% in comparison to the base year. This net reduction under 100% conversion to organic agriculture arises because emissions from fertilized soils drop considerably and the emissions from synthetic fertilizer production that also contribute significantly drop to zero, while the emissions from livestock and methane from rice increase only slightly. In sum, these effects offset increased emissions due to higher land use and deforestation. As this reduction is thus mainly due to the generally lower nitrogen fertilization levels (no mineral fertilizers)

with corresponding lower emissions from fertilizer application in organic production, it is important to emphasize that any increase in N-supply to address these critically low N levels in organic agriculture would correspondingly increase $N_2O$-emissions from fertilizer applications. It would thus lessen the reduction in GHG emissions or even change it to a zero or slightly increasing effect. We also emphasize that these emissions calculations follow the IPCC guidelines and do not refer to recent meta-studies on emission factors[32]. Skinner et al.[32] find rather higher emission factors for organic than for conventional production. On the other hand, they find that total N inputs are only a weak determinant for total emissions for organic production while they are a good determinant for conventional systems. However, evidence is not yet robust enough to deviate from the classical IPCC approach in such a global food systems model. We thus do not use adapted emission factors for different production systems and types of fertilizers and do not challenge the proportionality to inputs for organic production. A relatively small part of the difference in GHG emissions again reflects the difference in energy use. Without emissions from deforestation and organic soil loss, GHG emissions are reduced by 11–14% (still representing an increase from the baseline by 12–14%). Water use is similar to that in the reference scenario, which means an increase of 60%, compared to the base year. This occurs because, in the absence of evidence to the contrary, we assumed similar water demand per tonne output for organic and conventional systems. In contrast to total areas, total production volumes do not change much, as by assumption, all scenarios supply the same calorie and protein levels. Since synthetic pesticides are not used in organic agriculture, their impacts correspondingly drop to zero. However, this does not account for increases in non-synthetic pesticides in organic systems, such as copper (organic management allows for some non-synthetic pesticides that can potentially be harmful to the environment).

The impacts on the environmental indicators of complementing the conversion to organic agriculture with the 100% reduction of food-competing feed (FCF) and a 50% reduction of food wastage are shown in Fig. 5 (the top left panel uses the same data as Fig. 4). Supplementary Fig. 10 in addition displays the results for the intermediate scenarios with a 50% reduction of FCF and 25% food wastage reduction. The patterns remain similar, but complementing the conversion to organic agriculture with these additional strategies has the potential to achieve lower impacts along all indicators (at least without ICC). Large improvements are in particular achieved via the reduction of FCF. Supplementary Fig. 11 displays the results on environmental impacts when assuming low instead of high yield gaps. Main differences are the reduced land demand with lower yield gaps and the somewhat more challenging situation regarding N-supply.

Supplementary Figs. 12–15 display these results in another design for easier assessment of which share of organic production may be feasible according to the various environmental impacts for scenarios with 50% food-competing feed reduction, 25% or 50% food wastage reduction, intermediate CC impacts on yields, and high and low yield gaps. Most decisive for feasibility are land use and N-surplus.

We modelled a range of key environmental indicators, but we did not model impacts on biodiversity, given the complexity and —for many indicators—inadequacy to capture such in a global model. However, when linking to impacts that correlate with biodiversity, some indications for impacts on biodiversity can be given: Increased area use and deforestation under organic agriculture rather increase pressure on biodiversity, while the reduced pesticide use and nitrogen surplus reduce this pressure. Less ambiguity is again reached when combining conversion to organic agriculture with the other two food systems strategies,

resulting in overall reduction of all environmental impacts including area use, and thus suggesting a general reduction of pressure on biodiversity under these combined scenarios.

## Discussion

Organic agriculture can only contribute to providing sufficient food for the 2050 population and simultaneously reducing environmental impacts from agriculture, if it is implemented in a well-designed food system in which animal feeding rations, and as a consequence reduced animal numbers and animal product consumption, and food wastage are addressed. Solely converting to 100% organic production within an agricultural production system that should provide the same quantities and composition of outputs as in the reference scenario is not viable and would lead to increased agricultural land use. To be able to comprehensively assess the potential and challenges of a global conversion to organic agriculture, modelling the consequences of such a conversion needs to be based on a comprehensive food systems perspective, as has been adopted here, rather than simply addressing organic yield gaps. The key-challenges of land demand, and to a lesser extent N-supply, for large-scale conversion to organic production also reflect the multi-factorial perspective on maintaining soil fertility, nutrient recycling and ecosystem services, instead of adopting a maximum yield goal for single crops as a stand-alone performance criterion.

Reducing global average demand for animal products and their share in human diets is a strategy for more sustainable food systems on the basis of natural resource use, environmental impact and also human health arguments[9, 33–35]. We have shown that the favourable environmental performance of reduced animal numbers in livestock production that is free from food-competing feed and organic agriculture can be combined to provide a promising blueprint for more sustainable agricultural production, food supply and consumption. In our scenarios, livestock's role is again focused on utilization of resources that otherwise would not be available for human food consumption, namely grasslands and other grazing lands, and by-products from food production[26]. Interestingly, in such a system, the need to reduce animal product output emerges from agronomic and physical/technical characteristics, namely by restricting feed supply to energy and protein that stem from resources that cannot be utilized for food production directly, such as grasslands and a range of processing by-products. It is not driven by dietary changes externally imposed at the consumer level, although such changes are a clear consequence of the production shifts.

Food consumption patterns also play a key role for sustainable agriculture with regard to a second aspect addressed in this model, namely food wastage. In the scenarios with organic conversion and reduction of food-competing feed, agriculture was required to provide the same amount of calories and protein as the reference scenario[7], setting this demand as the benchmark to be met. However, this global average demand of 3028 kcal/cap/day as modelled by the FAO includes food wastage, that amounts to 30–40% globally, according to the most recent estimates from 2011[36]. Reducing food wastage thus offers a complementary approach to reducing resource use and the environmental impact of agriculture.

In summary, our study shows that organic agriculture can contribute to providing sufficient food and improving environmental impacts, only if adequately high proportions of legumes are produced and with significant reductions of food-competing feed use, livestock product quantities, and food wastage. The development of organic agriculture in the future should take up these challenges on the consumption side, and not only focus on sustainable production. This would, in particular, reduce the

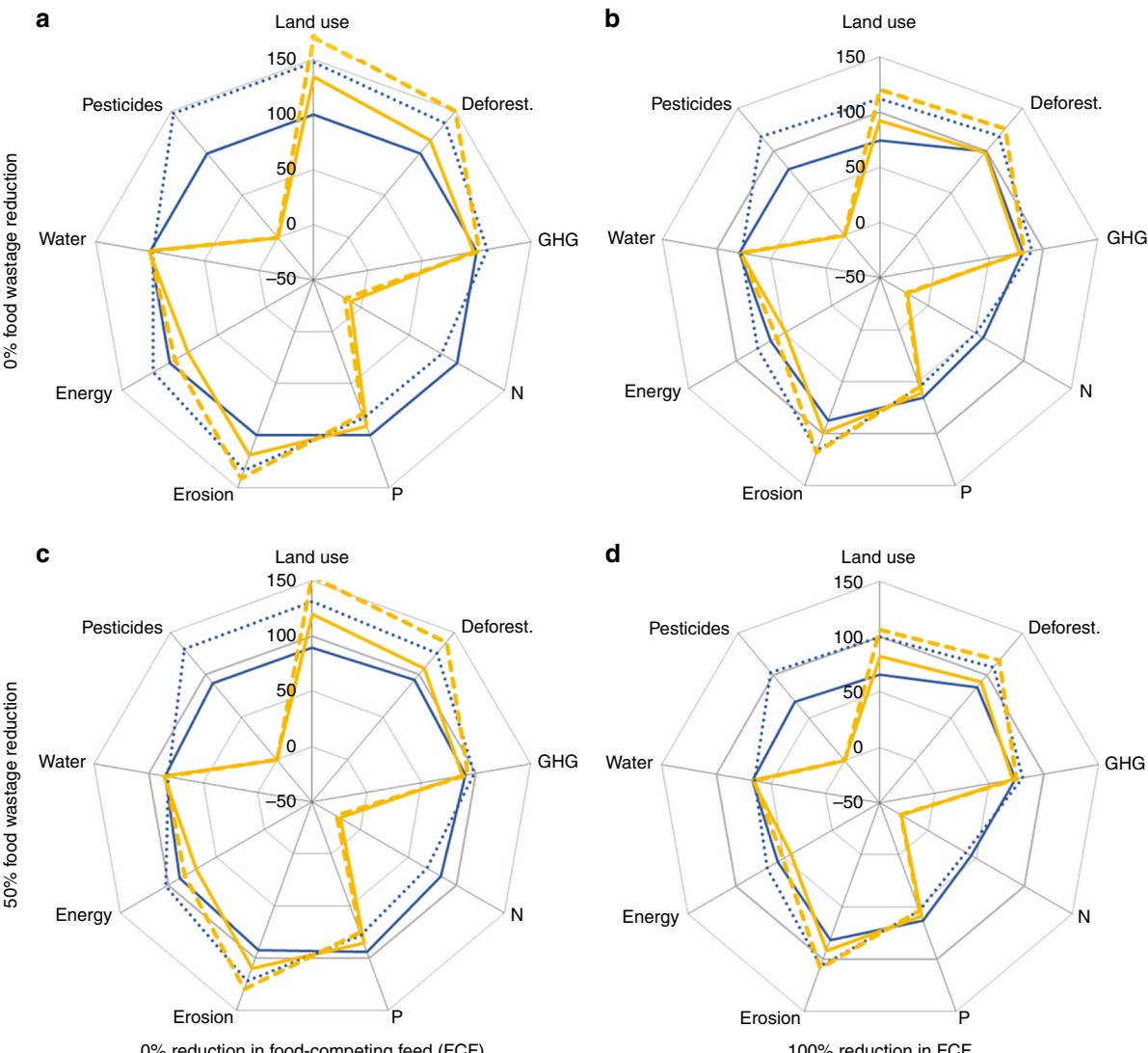

**Fig. 5** Year 2050 relative environmental impacts of a full conversion to organic agriculture in combination with complementary food systems strategies. Environmental impacts of organic (100% organic agriculture, yellow lines) and conventional (0% organic agriculture, blue lines) scenarios with concomitant changes in livestock feed and food waste strategies. All scenarios are shown relative to the reference scenario (i.e. 0% organic agriculture, no changes in livestock feed and food waste; dark grey line), with (dotted lines) and without (solid lines) impacts of climate change on yields; Calories are kept constant for all scenarios. The numbers on the axis indicate % impact, relative to the reference scenario; Calories are kept constant for all scenarios without food wastage reduction. Food-competing feed (FCF) use is at the levels of the reference scenario on the left, **a** and **c**, and changes towards zero FCF use to the right, **b** and **d**; wastage reduction changes from 0%, top **a** and **b**, to 50%, bottom **c** and **d**. Indicators displayed: cropland use, deforestation, GHG emissions (incl. deforestation, organic soils), N-surplus and P-surplus, water use, non-renewable energy use, soil erosion, pesticide use. Results for intermediate scenarios (50% reduction in FCF and 25% food wastage reduction) are displayed in Supplementary Fig. 10

necessity for yield increases, and a wise combination of production and consumption measures could provide an optimal food system. All of the difficult tasks: 'increasing (organic) yields', 'increasing organic production', 'reducing food wastage' and 'reducing animal numbers and animal product consumption' would be implemented together. Thus, none of those would be needed as a single measure at maximal coverage. All could be implemented at partial coverage only and in combination, leading to the improvements needed to increase sustainability of the global food system.

## Methods

**General description of the SOL-model**. The SOL-model[26] is a bottom-up, mass-flow model of the agricultural production and food sector. It is calibrated with

FAOSTAT data[37], in particular the food balance sheets[38], and covers all countries and geographic territories as well as commodities covered in FAOSTAT. Given lack of data for a range of those, this amounts to detailed coverage of 180 primary crop and 22 primary livestock activities in 192 countries. Behind this, data on commodity trees from FAOSTAT are used, covering around 700 intermediate products. In the following, only the main aspects and general traits of the SOL-model are presented, as a detailed description is already available elsewhere[26].

Each crop and livestock activity in the SOL-model is characterized by a set of inputs and outputs, i.e. all physical flows of quantities and nutrients related to the individual activities. Input to livestock activities are feed, energy input for buildings, processes conducted in stables (cleaning, feeding) and fences. Outputs include human-edible (meat, milk, eggs) and human-inedible products (skins, hides, bones, etc.), manure excretion, nutrient losses and GHG emissions (enteric fermentation, manure management; $CH_4$, $N_2O$, $NO_3$ and $NH_3$). Feed is further differentiated into four categories; (a) fodder crops grown on arable land, (b) concentrate feed derived from human-edible food (e.g. grains, pulses) grown on arable land, (c) grassland-based fodder and (d) fodder from agricultural/agri-

industrial by-products. The feed grown on arable land (a and b) is in competition with food production (termed 'food-competing feed'), while grassland-based feed and by-products are not (c and d).

For cattle, pigs, and chickens, country-specific herd structures have been calculated with a maximum entropy model[39]. This derives the most probable distribution of age-classes within the reported number of living and producing animals, as well as reported import and export numbers of living animals, and allows a more detailed assessment of feed and other input demand, as well as environmental impacts.

Fish and seafood is not a focus of the SOL-model and is addressed as described in more detail in the "Fish, seafood and aquaculture supply" section of the Supplementary Material of an earlier paper on the model[26]. The key assumptions are the following: in 2050, it is assumed that 60% of total fish and seafood supply are from aquaculture, and that 75% of aquaculture are fed, thus resulting in a supply of 45% of total fish and seafood supply in 2050 stemming from fed aquaculture. This share is correspondingly reduced with reductions in food-competing feed components, thus resulting in a drop of fish and seafood supply by almost 50% for the scenario with 100% reduction in food-competing feed. These numbers are based on a range of FAO and OECD references[40–43]. Further details, references and arguments for these choices are given in the above-named reference[26].

Input for crop activities include: land areas, mineral and organic fertilizers (manure, crop residues), N-fixation, pesticides and management practices. Outputs include crop yields, residues, and N-losses and P-losses. Each animal and crop activity comes with a range of environmental impacts (land occupation, N-surplus and P-surplus, non-renewable energy use, greenhouse gas emissions, water use, pesticide use, deforestation, soil erosion).

The SOL-model is a physical mass balance model capturing biomass and nutrient flows to assess the physical feasibility of different scenarios. It does not take into account economic restrictions and market effects relating changes in quantities to changes in prices. Economic aspects are key for the social viability of these scenarios, but their inclusion would come at the expense of the detailed commodity and country differentiation, and would require many additional assumptions on price and cross-price elasticities. This would increase model complexity considerably and hamper straightforward interpretation regarding physical viability of the scenarios, which is our focus here.

The following sub-sections describe the additional model parts, data and assumptions used in this paper that are not yet described in the previous section and in an earlier paper and its supporting online material[26].

**Differentiation between organic and conventional agriculture.** For the livestock sector, we do not assume any differences between organic and conventional production, besides a yield gap of 10%. For milk and eggs, the yield gap refers to output per animal per year; for meat, the yield gap refers to slaughter weight. In organic systems, often the same slaughter weight as in conventional systems is reached, but after a longer time than in conventional systems. In our model, this is treated equivalently (lower number of meat animals with same yield and higher number of animals with lower yields). In particular, we do not assume systematically different feeding rations between those two production systems. The yield gap of 10% is somewhat more conservative than the values reported in the literature, which amount to a yield gap of 3.2%[21]. We chose such a more conservative value, because the reference used[21] reports rather low yield gaps in general and as the other, more conservative meta-studies on organic yield gaps[17,44,45] do not report values for animals at all.

*Crop management*: Organic agriculture is characterized by the ban of synthetic fertilizers and pesticides, and a particular focus on soil fertility and crop rotations, nutrient cycling and ecosystem dynamics[46]. In the SOL-model, this is captured by setting synthetic nitrogen fertilizers and pesticides for the organic production shares to zero, as well as assuming 20% of legume crops in crop rotations, i.e. a legume crop every 5 years, and by assumptions on yields. Legume crop-specific N-fixation rates were used[47]. The composition of legumes cropped in the organic systems was chosen to reflect the share between different legume crops as reported in the reference scenario. For the organic system, this rather over-estimates the relative share of food legumes with respect to green manure. Conservatively, it is assumed that rock-phosphate is used as a P-source in organic agriculture in similar relations to P-demand as mineral P-fertilizers are used in conventional agriculture.

*Crop yields*: Yields in organic agriculture are usually lower than in conventional agriculture. For the main results reported in the paper, we assume the most conservative estimates of organic yields that show a yield gap of on average 25%[17]. We do a sensitivity analysis regarding this yield gaps and also calculate with the organic yields for the lower end of the yield gap estimates, i.e. for the highest organic yield estimates (an average yield gap of 8%)[21]. The most recent meta-analysis[44] shows an average yield gap of 20% and thus lies in between those two values. In particular, the analysis reporting low yield gaps, i.e. high organic yields[21] has been highly contested[19,20] and we emphasize that we use the range spanned by the low[17] and high[21] organic yields for sensitivity analysis of potential yield gaps from the most conservative to the most optimistic estimates available in the literature, without further assessing the standing of the more optimistic end of this range. To be conservative, we reported only the results with high yield gaps in the paper, and report those for low yield gaps in the Supplementary Figures.

For the scenarios, we assume the same yield gaps for developing and developed countries, albeit data for the higher yield gap is reported for developed countries only[17]. This is based on the assumption that technological progress likely leads to a convergence of agricultural productivity in developed and developing countries, both for conventional and organic systems. Higher yields for organic production in developing countries are reported in the study with low yield gaps[21], but the data from developing countries often compare optimally managed organic systems with traditional rather inefficient conventional systems, or are based on comparisons of non-conventional systems that however do not qualify as organic (e.g. the System of Rice Intensification SRI) with conventional systems. It can be assumed that optimally managed conventional systems would also perform much better, leading to similar yield gaps to those observed in developed countries.

*Emission factors*: Emission factors, such as for fertilizer applications or manure management, as well as per area soil erosion, deforestation pressure and water use are assumed to be identical between organic and conventional production. This is motivated by the aim to provide a conservative estimate on the performance of organic agriculture and the lack of robust data to motivate utilization of differing parameters between the two systems. Where the data allowed for differentiation, we assumed such, e.g. for energy use (CED) based on Ecoinvent 2.0 data.

**Food wastage.** FAOSTAT reports incomplete food wastage numbers only, and we therefore used the more detailed data from the Food Wastage Footprint[10,48]. This data provides food wastage shares for commodity groups and world regions, that are then applied to all countries and commodities within the respective regions and commodity groups[48]. Wastage data is provided along the whole value chain from production to consumption and dumping (differentiating for five value chain steps: agricultural production, post-harvest handling and storage, processing, distribution, consumption; dumping includes GHG emissions from anaerobic decay of the wasted biomass), and the corresponding shares are added up to derive wastage shares at primary commodity levels. Scenarios with wastage reduction assume 25 and 50% less wastage, i.e. the wastage share for each commodity is reduced by 25% or 50% respectively. The scenarios thus trace effects from wastage reductions from 0% up to 50%. In the model, this results in a corresponding quantity of each commodity not being produced, thus leading to reduced input demand and impacts.

**Climate change impact on crop yields.** A couple of recent publications assess climate change impacts on yields[49–53]. They mainly focus on the most important crops (wheat, maize, rice, soy), and for most commodities no assessment is available. Furthermore, several specific aspects such as the potential inclusion of $CO_2$-fertilization further complicate results. We thus decided to undertake a sensitivity analysis on climate change impacts on crop yields and to assume a broad range from an optimistic extreme of no climate change impacts on yields (as assumed in the reference scenario[7]) to a pessimistic estimate of no further yield increases if compared to the baseline. Some of the literature reports potential yield decreases with respect to current levels, but we decided to not include this possibility in the scenarios. We thus modelled scenarios with no climate change impact and full climate change impact reflecting zero further yield increase as extremes, and an intermediate scenario for illustration, which assumes that yield increases are only 50% of the reference scenario. Due to lack of data, we assumed animal yields to remain unaffected by climate change impacts.

Furthermore, we modelled scenarios where organic agriculture is affected less by climate change than conventional agriculture. This assumption reflects the argument that organic agriculture is better adapted to climate change than conventional agriculture; however, evidence for this is scarce and no conclusive statement on this can be given[54,55]. For illustration, we modelled this situation by assuming that climate change impacts organic yields by merely 60% as much as conventional yields, but we report the corresponding results in the supplementary figures only and not in the main body of the paper.

**Environmental indicators.** This section shortly describes the environmental indicators used in the SOL-model. For further details, we refer to the literature[26].

*Land occupation*: Land occupation measures the cropland and grassland areas utilized in agricultural production. For cropland, land occupation combines areas harvested and cropping intensities. The latter indicate how many times a hectare is harvested on average. Cropping intensities are usually less than one (due to fallow areas) and therefore land occupation reports higher values than areas harvested[7].

In all scenarios, grassland areas are assumed to stay constant[26]. It has to be mentioned that many grasslands and grazing lands currently face high environmental and societal pressures. Focusing global ruminant production on those areas would thus necessitate to adequately address those challenges[56].

Changes in land use are thus between arable and non-agricultural land only (e.g. forests). The indicator cropland occupation captures the total land demand in the scenarios, irrespective of where this may be sourced, while the indicator deforestation captures the pressure from this land demand on forests in countries where deforestation is an issue, assuming similar land sourcing patterns as in the baseline (cf. further down).

*N-surplus*: The N-surplus describes the difference between N-inputs and N-outputs. N-inputs for crops are mineral N-fertilizers, N-fixation, organic fertilizer, crop residues and seeds. N-inputs are derived based on available N (from mineral

fertilizers, fixation, crop residues and manure), assigned to the various crops in relation to their relative N-demand as share of total N-demand of all crops. N from atmospheric deposition is not included in the N-surplus. Total N output of a crop equals the amount of nitrogen that is taken up by a crop during the growing period, i.e. the amount of N in yields and crop residues, as well as emissions ($NO_3$, $NH_3$ and $N_2O$). For organic crop activities only organic N-inputs are possible (manure, N-fixation, crop residues), as no mineral N-fertilizers are allowed. For animals, N-inputs are feed and N-outputs are yields, manure and emissions from manure ($NO_3$, $NH_3$ and $N_2O$). Input and output sources for N thus cover all relevant flows and compounds, in particular direct emissions, volatilization and leaching of $N_2O$, $NH_3$ and $NO_3$ from manure management and fertilizer application of any kind. Emission factors are according to IPCC 2006 Guidelines (Tier 1). SOL-model results for the global aggregate N-surplus and for aggregates of sub-categories, such as N-Fixation, etc. in the base year are consistent with the literature (Supplementary Table 1)[47, 57, 58].

Besides the yield gap, we did not assume any systematic differences between organic and conventional livestock activities that would affect N-surplus.

N-surplus is displayed as a global per-hectare average. This thus covers all N-inputs and outputs from croplands and grasslands and takes an average over all those areas. It thus cannot be directly compared to values for cropland reported in the literature. The use of per-ha numbers illustrates our focus on assessing the viability of organic production from an agronomic point of view, as N-supply is often seen as a challenge to organic production[19-21]. The choice of a surplus of between 10 kg/ha and 5 kg/ha as optimal is motivated from the literature that reports that N-surplus could be decreased by 50–70% for various cereals without affecting yields and as based on the numbers on potential N-input reduction and shares of excess N in relation to inputs, even higher reduction rates can be derived[59, 60]; the optimal range chosen in the model signifies a reduction of 60–80% with respect to the reference scenario and we thus chose this somewhat higher range of 60–80% reduction of N-surplus as an illustrative optimal level to be aimed at in the assessment of how viable changes in N-surplus in the scenarios are from an agronomic perspective.

The potentially challenging situation regarding N-supply in organic agriculture has also been taken up in the literature[21]. They suggest that this challenge could be met when cropping intensities were to increase and fallow land and intercropping were to be systematically used for legume production. We did not incorporate this in the model as it would necessitate a range of additional uncertain assumptions, such as on water availability, overall adequacy of areas for off-season legume cropping and yields. The corresponding assessment[21] is highly contested, as they assume legumes between the main crops on all areas, resulting in additional 1360 million ha legumes, and assuming a very high nitrogen fixing rate of about 100 kg N/ha. For this, we also refer again to the critical assessment of this analysis[19, 20].

*P-surplus*: As for N, P-surplus is defined as the difference between P-inputs and outputs. P-flows are expressed as $P_2O_5$. Inputs and outputs are mineral P fertilizer, $P_2O_5$ in feed, manure, crop residues and yields. When assessing P-surplus, it has to be considered that large quantities of P are fixed in soils and the surplus thus rather expresses a 'loss potential', that can be realized, e.g. through erosion, than actual losses to the environment. SOL-model results for the total P-balance in the base year are consistent with literature values[58].

*Non-renewable energy use*: The life cycle impact assessment methodology 'cumulative energy demand' (CED)[61] is used to calculate non-renewable energy use. Renewable energy components are disregarded. The share of non-renewable energy for fuels and electricity was assumed to stay constant in all scenarios and no technical progress in energy efficiency was assumed.

Inventory data for each activity, including the differentiation between energy use in conventional and organic activities, were taken from LCA-databases, i.e. the ecoinvent 2.0 database and other sources[62-64]. Energy use is linked to farm activities and includes energy use for seeds, crop protection, fertilization, mechanization, organic fertilization, fences, stables and depots for roughage. Data for animal production were taken from the ecoinvent 2.0 database and other sources[64, 65]. Energy use for fertilizer production is modelled specifically for the fertilizer quantities used for each crop in each country. Energy carriers inputs were modelled according to the ecoinvent 2.0 and SALCA inventories[66]. Due to lack of data for trade, transportation energy use was disregarded. We emphasize that absolute numbers on energy use may be biased due to the data quality behind ecoinvent 2.0, which is partly rather old, but relative differences between scenarios are much less sensitive to such data problems.

*Greenhouse gas emmissions*: GHG emissions are based on Tier 1 and 2 approaches from the IPCC-Guidelines from 2006. Emissions for agricultural inputs and infrastructure are taken from the ecoinvent 2.0 database and LCA studies[62-64]. Emissions from deforestation and from agriculturally managed organic soils are taken from FAOSTAT (2). We did not differentiate emission factors between organic and conventional production systems and assumed the same feeding rations and the same shares in different manure management systems for organic and conventional production systems.

For the GHGs, Global Warming Potentials (GWP) from the IPCC2006 100a Tier 1 methodology were used, i.e. 25 t $CO_2e$/t for $CH_4$ and 297 t $CO_2e$/t for $N_2O$. IPCC Tier 1 methods were used to calculate the emissions from manure management and fertilizer application. The Tier 2 methodology was used for enteric fermentation, in order to capture the impacts of different feeding regimes. ecoinvent 2.0 and other data[67] were used to calculate the GHG emissions from the

production of mineral fertilizers and pesticides. GHG emissions from processes and buildings were derived from the respective CED-values and application of process-specific conversion factors derived from ecoinvent 2.0. When aggregating over the common emission categories only, SOL-model results for total GHG emissions in the base year are similar to the values reported in the literature (Supplementary Table 2)[68, 69]. These two literature references differ substantially in the values for emissions from enteric fermentation; SOL-model results are more similar to the values based on the emissions data in FAOSTAT[68].

*Water use*: Water use was calculated from AQUASTAT data[7] on consumptive irrigation water use per ton of irrigated production and data on irrigated areas for various crops and crop categories. We assumed similar irrigation values per ton of irrigated production for organic and conventional production. Differences between the systems then arise due to different yields and different area shares for different crops with different crop-specific irrigation values as reported in AQUASTAT.

*Pesticide use*: There is no consistent data set on pesticide use covering different countries, and we thus developed an impact assessment model for assessing pesticide use incorporating three factors: pesticide use intensity per crop j and farming system k ($PUI_{j,k}$), pesticide legislation in a country i ($PL_i$), and access to pesticides by farmers in a country i ($AP_i$) (Supplementary Table 3).

This model has been described in the supplementary information to Schader et al. (2015)[26] and we quote from this description in the following, as we used the same model in this work.

Each factor was rated on a scale from 0 to 3 by FAO-internal and external experts (Jan Breithaupt, FAO, involving experts from regional FAO offices; Frank Hayer, Swiss Federal Office of the Environment; Bernhard Speiser, Research Institute of Organic Agriculture, FiBL) with experience in different countries and with different methods of calculating pesticide impacts (life cycle assessment, risk assessment). The descriptors for each scale have been designed so that risks from pesticides are 0 if only one of the three model parameters is equal to 0. For instance, if there are no harmful pesticides used in a crop, or if in a country legislation completely bans harmful pesticides or farms do not have access to pesticides at all, the impact factor for a crop-country combination will be 0.

As an example, the pesticide use impact factor ($IF_{i,j,k}$) for coffee production in Ghana was 6 as: PUI was rated as 3, PL was rated as 2 and AP was rated as 1. Values for PL and AP are shown in the Supplementary Table 4 below, values for PUI can be found in Supplementary Table 5. To calculate crop and country-specific pesticide use IF, the three factors were multiplied together (Eqn 1).

$$IF_{i,j,k} = PUI_{j,k} \times PL_i \times AP_i \, \forall \, i, j, k. \qquad (1)$$

Thus, for each crop in each country, a value between 0 and 9 has been assigned on a per-hectare basis, serving as an indicative proxy for overall pesticide use per crop and country (and per-ha). Aggregate values per country were derived by multiplying this pesticide use indicator with the respective crop areas and summing over all crops. The pesticide use intensity of organic activities was evaluated to be zero throughout all activities and countries. This neglects certain aspects of plant protection in organic agriculture, such as the use of copper.

*Deforestation*: FAOSTAT deforestation values for the base period 2005–2009 are set in relation to the change in agricultural land areas over this period in each country. This ratio is then used to derive deforestation values from area changes in the scenarios. Thereby, we have attributed 80% of deforestation to agriculture[70]. Assuming the same deforestation pressure by country in 2050 as in the baseline is a very strong assumption, but due to the lack of better data on a global level, we decided to uses those rates for a first assessment of deforestation pressure.

In some cases, no data on change in agricultural land area has been available for 2005–2009. Then, the ratio between deforestation areas (multiplied by 0.8) and total agricultural land area has been built for the base year. This ratio is then multiplied with the total agricultural land areas in the scenarios to derive values for deforestation. In these cases, we thus used total agricultural area (instead of the change in agricultural area) as a proxy for the pressure of agriculture on forests. In cases where total forest area increased, deforestation values have been set to zero.

Modelling deforestation in relation to agricultural area changes explains the drop in deforestation rates in the reference scenario, as annual land expansion rates to 2050 are projected to be lower than the observed rates for the base years 2005–2009.

Modelling deforestation in this way thus captures the pressure of land increase on forests in countries, where deforestation is an issue, by assuming a similar dynamics as in the baseline. It thus complements the land occupation variable that captures the land demand, irrespective of where it may be sourced from in a specific country. In particular for larger increases in land occupation, this approach may rather underestimate deforestation, as a large part of this additional land likely would have to be sourced from forests, given the assumption of constant grassland areas. A more detailed assessment of the deforestation dynamics with increased cropland demand would necessitate combination with data on the suitability of grassland, forest and other areas for crop production of various kinds, which is however beyond the scope of this paper.

*Soil erosion*: Soil erosion was based on data for soil quantities lost (tonnes soil per-ha per year) via water erosion on a per country basis[26]. Due to lack of data, wind erosion has not been included. Per-ha soil erosion rates were then combined with a soil susceptibility index for different crops to differentiate between crops with lower and higher soil erosion risks. This index was set 0 for permanent grasslands, 1 for crops with a short period of bare fallows and 2 for crops with

**Table 1 Overview of model assumptions for the various scenarios**

| Parameter | Base year | Reference scenario | Scenario assumptions on organic shares (from 0 to 100%), food wastage reduction (0, 25, 50%) and livestock feed and animal numbers (reductions in food-competing feed from 0 to 100% and corresponding reduction in animal numbers) |
|---|---|---|---|
| Year | 2005–2009 | 2050 | 2050 |
| Human population | FAOSTAT | FAOSTAT | As in the reference scenario. |
| Calorie and protein supply per person | FAOSTAT | Alexandratos and Bruinsma 2012[7] | Calorie supply equal to the reference scenario (if no food wastage reduction takes place); these numbers report the total domestically available amounts, *including* amounts that are lost due to food wastage; for comparison, scenarios with constant protein intake are also assessed. With food wastage reduction, calorie/protein supply is reduced accordingly. |
| Food wastage | Food wastage footprint | Food wastage footprint | Relative reduction according to the scenario (25% or 50%). |
| Share of livestock products in human diets | FAOSTAT | Alexandratos and Bruinsma 2012[7] | Model-endogenous calculation of the fraction of livestock-based food energy in total food energy supplied by the food system. |
| Crop yields | FAOSTAT | Alexandratos and Bruinsma 2012[7] | Lower organic yields according to the yield gaps from the literature (sensitivity analysis: 8–25% lower on aggregate levels). |
| Livestock yields | FAOSTAT | Alexandratos and Bruinsma 2012[7] | Based on reference scenario but yields decrease by up to 20% with zero feed from human-edible products and crops from arable land, due to suboptimal feed composition. As a sensitivity analysis, scenarios are also calculated for 0% and up to 40% yield reductions. Results presented in the paper report a mid-range estimate of 20% reduced yields with reductions in food-competing feed. |
| Yield increases | n.a. | Alexandratos and Bruinsma 2012[7] | Percentages increase as in the reference scenario (also for organic). A sensitivity analysis based on projections of climate change impacts on yields is included. The yield gap (cf. above) for organic production is then applied on these increased yields for conventional production in 2050. |
| Increase in cropping intensity | n.a. | Alexandratos and Bruinsma 2012[7] | As in the reference scenario. |
| Legume shares | FAOSTAT | Alexandratos and Bruinsma 2012[7] | Increased legume shares till 20% for 100% conversion to organic production or for 100% conversion to feeding rations without food-competing feed; in combinations, the higher legume share of those two changes is used, thus assuring a minimum level of 20% legumes in the crop rotations for 100% conversion to organic production. |
| Ratio arable land/grassland | FAOSTAT | Alexandratos and Bruinsma 2012[7] (net grassland stays constant, arable land increases) | Net grassland is kept constant as in the reference scenario; arable land change according to the amount of calories/protein supply. |
| Ruminant numbers | FAOSTAT | Alexandratos and Bruinsma 2012[7] | Model-endogenous calculation of the number of animals that can be fed on available feed. |
| Non-ruminant numbers | FAOSTAT | Alexandratos and Bruinsma 2012[7] | Model-endogenous calculation of the number of animals that can be fed on available feed. |
| Share of feed types in feeding rations | Herrero et al. 2013[80] | As in the base year | Based on rations for base year and reference scenario but adapted according to feed supply of human-edible products and crops from arable land dropping gradually to 0%. |
| Utilization shares | FAOSTAT | Alexandratos and Bruinsma 2012[7] | Feed share of primary food crops reduced (0% for 100% reduction of feed from human-edible products and crops from arable land). |
| Deforestation | FAOSTAT | Increased land areas increase deforestation, using the land areas forecasted and deforestation rates from FAOSTAT | If more/less land is needed to satisfy food availability, pressure on forests increases/decreases. |

longer periods of bare fallows such as maize or beets. This classification is based on expert consultations and literature[71–77]. The on average higher soil organic matter contents in organic agriculture are likely to reduce soil erosion rates[78, 79]. However, in order to produce conservative estimates at global level, we did not consider this impact in our model. Potential differences in soil erosion between organic and conventional systems thus arise from the different area allocation to different crops, thus changing relative shares of crops more or less susceptible to erosion.

**Scenario description**. Based on an assessment of (i) the situation today ('base year') capturing the average situation for 2005–2009 as provided by FAOSTAT and

additional data[26], the following scenarios are calculated in the SOL-model: for 2050 (ii) the reference scenario from the FAO[7]; The reference scenario is the basis for (iii) scenarios with an increasing share of organic production, up to a 100% conversion. In addition, scenarios are assessed, assuming increased organic production combined with reduced food wastage by 25 and 50% with respect to the regional and commodity group specific values from the FAO[10] (the latter value of 50% being among the Sustainable Development Goals for 2030), and with reduced animal product supply, modelled via food-competing feed reduction by 50 and 100%. The 100% reduction assumes entirely grass-fed ruminant production, and monogastrics fed only on by-products from food production[26]. The scenarios investigated in the SOL-model take the reference scenario as a starting point and make additional assumptions on specific parts of interest (Table 1). The SOL-model then derives the inputs, outputs and environmental impacts for all crop and livestock activities, given these additional assumptions. For all scenarios, we assess food availability (expressed as calorie and protein supply per capita per day.), dietary patterns, land occupation, animal numbers and a range of environmental impacts such as N-surplus and P-surplus (i.e. the net difference between N/P inflows and out-flows), water use, deforestation and GHG emissions.

All scenarios are then assessed in comparison to the reference scenario. For this, one of three conditions is chosen, namely that the scenarios provide the same amount of calories for food or the same amount protein for food, or use the same acreage of cropland and grassland as the reference scenario. Only in case of scenarios with wastage reduction, the total amount of calories or protein to be produced has been reduced accordingly. These conditions are imposed country-wise. This is achieved by changing cropland areas, production and domestically available quantities upwards or downwards accordingly to meet this goal (grassland areas being kept constant). Patterns for the different commodities thereby remain as close as possible to the pattern observed in the reference scenario (i.e. relative shares between commodity groups and between commodities within these groups). Changes in some of these patterns are however unavoidable in several scenarios, for example if the share of legumes increases and the animal product shares decrease. However, while meat consumption may drop in one scenario, the relative share of chicken and pig meat is retained on country level, just as the relative shares of different legumes is retained on country levels when total legume shares increase.

The main results in the paper are displayed with the same-calorie condition, as this is most illustrative to capture food availability aspects that are in the centre of interest, and as results show that the scenarios meeting the same-calorie condition always produce the same or a higher quantity of protein as the reference scenario and are thus adequate in protein supply. This is due to the higher legume shares in organic production and the lower organic yields for cereals and other staple crops for calorie provision. Table 1 provides an overview of the scenarios calculated and some further information is provided afterwards and in the literature on the SOL-model[26].

The general condition to produce the same amount of calories or protein as in the reference scenario is chosen to assure comparability of viability and impacts of the different scenarios with the reference scenario. Clearly, for many countries, these amounts are very high and it can legitimately be discussed, whether providing such amounts of food is a useful and realistic strategy; this is partly captured in the scenarios that include food wastage reductions. For other countries, the amounts forecasted are clearly at the lower end and for food security reasons should be increased. Given that the total global amount of calories and protein available is well enough to feed the world of 9 billion people, different assumptions on trade and on domestically available quantities in the reference scenarios could in principle deal with these issues. All this is not taken up in the scenarios presented here, for the above-mentioned reasons of comparability.

Commodity quantities in the scenarios are derived from the quantities reported in the reference scenario, by adapting some commodity quantities according to the scenario assumptions: (a) organic production with lower yields leads to reduced quantities from the same areas; (b) reduced wastage leads to reduced production; (c) reduced food-competing feed leads to lower animal numbers that can be fed from this feed and thus to less animal products that can be produced from it; (d) areas that become free due to reduced feed production are cropped with other crops according to their relative distribution in the domestic production; (e) legume shares are increased according to the share of organic production or food-competing feed reduction (up to 20%). Areas used for that are taken proportionally from all other areas of domestic production.

To all products, the same utilization shares, import and export ratios as in the reference scenario are then applied to derive domestically available quantities (utilization of commodities for feed is reduced according to the changes required for food-competing feed reduction, if such is part of the scenario). For each country, the total per capita food calorie supply derived from this domestically available quantity is then scaled to equal the food calorie supply of the reference scenario. The same scaling factor is then applied to the total production within country. This approach thus mimics the production and trade patterns (regarding relative quantities of different commodities) of the reference scenario as close as possible, given the specific scenario assumptions to allow for transparent comparison of the physical changes related to the different scenarios. In this, the SOL-model is explicitly not an economic model, as changes of production is not governed by an explicit trade-module with (cross-)price elasticities, but by direct assumptions on the relative shares of different commodities. This allows for a transparent assessment of the physical and agronomic viability of the scenarios in comparison to the reference scenario and does not entail projections on how the global food trade would adapt to the changes defining the scenarios.

Fertilizer inputs to crops are determined as follows: First, country-specific mineral N-fertilizer use as reported in the reference scenario is allocated to the different crop activities according to their demand and a crop-specific supply/demand ratio for this is then derived. In the scenarios, mineral fertilizers are applied to conventional crops using this same supply/demand ratio. Organic crops do not receive any mineral N-fertilizer. Crop residues and manure are then applied proportionally to the remaining N-demand after taking these mineral fertilizer applications into account. Thereby, a share of 50% of manure is assumed to remain on grasslands. N-fixation is a fertilizer input for legume crops only, as the N fixed by legumes applied to other crops is covered via the crop residues from legumes. Organic P input is then derived from the quantities of crop residues and manure and the respective P contents. Mineral P inputs are also taken from the reference scenario data on country level and are then allocated to the different crops according to the demand remaining after accounting for these P inputs from organic sources.

**Data Availability**. Model code and data used are accessible in the folder 'MullerEtAl_NCOMMS2017' at paper.fibl.ch.

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

## Acknowledgements

We are grateful for the inputs from Caterina Batello, Jan Breithaupt, Carlo Cafiero, Marianna Campeanu, Reto Cumani, Rich Conant, Piero Conforti, Marie-Aude Even, Karen Franken, Andreas Gattinger, Pierre Gerber, Frank Hayer, Jippe Hoogeven, Stefan Hörtenhuber, Mathilde Iweins, John Lantham, Robert Mayo, Eric Meili, Soren Moller, Jamie Morrison, Alexander Müller, Noemi Nemes, Monica Petri, Tim Robinson, Nicolas Sagoff, Henning Steinfeld, Francesco Tubiello, Helga Willer, and thank Robert Home for checking the language. K.-H.E. gratefully acknowledges funding from ERC-2010-Stg-263522 (LUISE). The input of P.S. contributes to the DEVIL project (NE/M021327/1), funded under the Belmont Forum/FACCE-JPI. This paper contributes to the Global Land Project (www.globallandproject.org). We acknowledge funding for open access publication by the Institute of Environmental Decisions, Federal Institutes of Technology, Zurich.

## Author contributions

A.M., C.S. designed the research, collected data, programmed the model and wrote the paper. N.E.-H.S., K.-H.E. designed the research, collected data and wrote the paper. P.S. designed the research and wrote the paper. J.B. collected data and programmed the herd structure sub-model. A.I. and P.K. collected data and designed the animal feed research

part. F.L. collected data, designed the animal feed research part and wrote the paper. M. S., U.N. designed the research and wrote the paper. All authors gave final approval to the manuscript.

## Additional information

**Competing interests::** The authors declare no competing financial interests.

