## [Peer Review File · Nature Communications]

Reviewers' comments:

Reviewer #1 (Remarks to the Author):

The scientific question raised in the paper on whether it is possible to satisfy 2050 FAO scenario food demand with global organic farming is very relevant and important in the context of sustainable future agriculture and the Sustainable Development Goals (SDGs). This scenario was introduced by Alexandaratos and Bruinsma and serves as the "standard" assumption for most groups, which deal with issues of future food security. I do not know of any publication that deals specifically with this relevant question as explicit as the submitted paper. The paper shows with a simple material flow model that organic farming under the assumptions used does not allow to satisfy demand without considerable expansion of farmland. Nevertheless a combination of organic farming, expansion, reduction of food waste and reduction of meat consumption may achieve the goal.

The basic assumption of the paper is that organic farming reduces yield by app. 20%. This assumption reduces and generalizes organic farming solely to a global factor for a loss in yield. This may be mainstream science for intense developed world farming practices. To generalize it to the global farming systems with its large variety of yield gaps and development stages is at least courageous. The paper lacks critical discussion to clarify whether this is a scientifically sound assumption, which I doubt at the moment. The paper then goes through a lengthy analysis of combinations of measures that combine the introduction of global organic farming practices with farmland expansion, reductions in food waste and change in consumption patterns. It shows that organic farming alone will not be able to satisfy global demand without unrealistic expansion of farmland. With the vastly expanded, now 4-dim. scenario space it is not a priori astonishing that the authors find solutions, which allows to satisfy this changing future demand. It should be made transparent that these results are not trivial in the sense that four parameters (expansion, % organic farming, reduction of food waste, reduction of meat diet) will (almost) always create a solution to the given problem.

The elasticities that are used to determine the changes in food supply within this scenario space are not clearly explained in terms of their geographic diversity and, even more important, their stability in a future 2050 world. This has to carefully be clarified to the reader together with a careful analysis on how the result of the study depends on the assumptions on elasticity that were used.

With these clarifications being successful I would like to see the proposed solution be carefully discussed with regard to its realism and impact on implementation in terms of e.g. geographical priorities.

Altogether, I think that the paper in its current form lacks scientific maturity and is not ready to be published in Nature Comms. It needs major revisions, critical discussion and clarification.

Reviewer #2 (Remarks to the Author):

I was very pleased to take on the review of this manuscript along with its associated supplemental materials. As the authors, I am keenly aware of the challenges posed globally by food systems given growing human demands and projected trajectories of production. We clearly need to find viable, lower impact alternative food systems if we are to limit substantial additional degradation of the biosphere. Consequently, I offer my feedback below in the context of wishing to strengthen your article and our collective understanding of the implications of alternative food systems. My comments below are organized into four major sections: general comments on the research, larger scale methodological or presentation issues, then finer-scaled editorial feedback on the main manuscript and finally finer-grained feedback on the text of the Supplemental materials.

General comments.

The manuscript represent an ambitious piece of research that helps us understand broad-scale

environmental implications of ag and food system impacts out to 2050 under a range of variables, including: a) levels of organic agricultural practice adoption, b) effects of climate change on yields, c) levels of aggregate food losses along supply chains, and d) the scale of animal products included in diets. Results indicate that there are combinations of variable adoption that can deliver sufficient calories for projected human population out to 2050 while reducing the scale of the total land under cultivation as well as resource and environmental impacts in comparison with the business as usual reference scenario. While prior research has tackled aspects of what is modeled here, this paper is original in its scope of variable inclusion and insight gained.

As addressed in detail below, I have some concerns with some of the data employed and how some data were handled (see specifically #3 and 4 below) and struggled for a long time in understanding what was actually being modeled in certain key aspects (see #2 below). More generally, from the number of fine-grained editorial comments I've made below, I think that the overall clarity and meaning could be greatly improved so that the value of the overall contribution of the research can be more easily discerned and understood.

Major issues:

1. There is an assumption evident throughout the manuscript, and it arises first in the title, and it is that organic agriculture equals sustainable agriculture. While key attributes of organic ag certainly address issues of current local and global scale unsustainability, many other aspects are simply unaddressed. They may improve or get worse under organic management. Relatedly, it is implicit in the title and elsewhere that the word sustainability is used, that there is a definite sustainability threshold below which we no longer have to be concerned with how food is produced or consumed. Options to address this would be to either simply remove the word sustainability from the title or adding something like 'more' before it.
2. I struggled for a long time to understand what was being modeled, and what results represented, in regards to reducing the percentage of concentrate feeds being fed to livestock. For example, in the text on page 6 and in the Y-axis second column of Figure 2 and elsewhere in the manuscript, the text refers to modeling the impact of reducing the percentage of concentrate fed to livestock. Initially, I assumed that this meant that the total production of livestock was left unchanged with only the feed input mix (forage vs concentrate) being varied. But of course, this made no sense given some of your findings. For example, that greenhouse gas emissions went DOWN with reduced levels of concentrate feeds fed to livestock. This makes no sense at all given that most life cycle assessment research, and particularly that on ruminants, clearly shows that higher levels of forages in diets drives UP GHG emissions. It was only after reading additional details in the supplemental materials that I realized that when you are modeling reductions in concentrate feed use, you are actually modeling absolute reductions in livestock volumes being produced. Indeed, this is the only way that some of your results make any sense at all. IF I now understand your work correctly, you need to be much clearer throughout the text and illustrative materials that what is now described as a percent concentrate feed share is actually a measure of the total amount of future livestock production that is being reduced.
3. While it is a small part of the overall model and results, I don't understand at all how fish and seafood are treated in the model. Indeed they only show up in Figure 4 and while there is a clear reduction in the seeming amount of seafood and fish produced as a function of the amount of concentrate feeds being produced (100% of the business as usual 2050 reference year versus 0% relative to the reference year), how is this arrived at? The clear assumption one could make is that your model is assuming that a large portion of future seafood in the 2050 reference years seafood will be grown on concentrate feeds. But is this at all defensible? Right now ~50% of seafood continues to be sourced from fisheries and while this isn't growing, it's not going away any time soon. And within aquaculture, while fed culture systems are an important and growing sub-sector of current global seafood production, non-fed aquaculture is massive and will continue to dominate global farmed seafood production globally. Perhaps my concern can be dismissed given the much bigger picture that you're working on but the seeming conflict of model outcomes with my understanding of global seafood systems causes me concern re some of the underlying model assumptions.

4. I am concerned with the quality and currency of the data that you've used to model energy and GHG emission impacts from agriculture. It was only once I got to page 5 of the Supplemental materials and then checked the associated references (#s 72-74 in particular) that I realized that you are using results of LCA research on ag systems that are at least 10 years old throughout your models. This not just an issue of inputs and emissions to ag systems changing over time but that over the last decade there has been a rapid increase in the amount of research available to characterize energy inputs and GHG emissions from ag systems around the world. Ultimately this is a question of how confident can we be that models of the future based on limited datasets from the past are reasonable? Somewhat relatedly, how did you go about integrating different observations regarding energy inputs and GHG emission from different studies/sources. If you had say five observations of energy inputs to wheat, did you take a simple arithmetical average? Given that much of the data underpinning your models have been derived from multiple sources, I believe that there is need for some description of how these data were treated/combined before they were used in the model.

Finer grained issues:

A. In line 13, consider adding "and expansion" after the word intensification

B. In line 16, it's really more than organic agriculture's technical feasibility that is contested is it not? More generally, are questions regarding it's ability to address issues beyond the attributes at the core of organic production.

C. In line 19, consider modifying the sentence to be something like "... assure sufficient food availability in 2050, while reducing a broad suite of environmental impacts.

D. Line 27. I may be a bit pedantic but is not the problem oversupply of reactive nitrogen? i.e. consider inserting the word reactive before nitrogen

E. Line 28, consider inserting the words 'land and' before water bodies. i.e. it would then read "... , eutrophication of land and water bodies, ..."

F. Line 38, consider inserting the word 'some' before environmental indicators as clearly not all evidence points to just positive outcomes as it acknowledged in the immediately subsequent sentence (that land areas under organic ag may need to be greater as yields are often lower)

G. Lines 44-47. The sentence that starts "Some authors complement..." is confusing. What is it that you are critiquing when you say that these studies (specifically references 18-20) lack a detailed food system approach? Is it that their purposes were flawed or lacking in some way? Put another way, the sentence reads like a criticism of this earlier work when you say, "... but these studies lack a detailed food systems approach...). My response is why should they have? Why is such an 'oversight' an oversight at all.

H. Lines 59-61. I know that there is more detail re the SOL-model in the methods etc, but I feel that more detail is needed here re how the model works and why it is useful. It is central to the analysis and as such, it's use needs more explication and defence.

I. Further re lines 60-61, what does the phrase "... tendency to reduced concentrate feed use." mean? Tendency is far too vague a word. Does organic ag actually employ less concentrate feed in livestock production or not. Does the SOL-model model this well or not. Finally, what is also particularly unclear to me is what does the tendency to reduced concentrate feed use mean for absolute levels of livestock (and aquaculture) production? Does it mean that these go down as well or just that the use of concentrate feeds is lower in organic ag as modeled but there is then a greater dependency on forages?

J. Line 62. Here you clearly transition to your results but the phrasing is weak. I would suggest starting the sentence more directly with something like "Our results show ..."

K. Line 63. What does "becomes feasible" mean? Speak more directly please. Something like perhaps "... , and less intensive agriculture (organic in this case) also represents a means of feeding 9+ billion people in 2050 if it is combined..."

L. Line 69. Again I may be a pedant but I suggest inserting 'improved' before soil fertility and reduced before pesticide use.

M. Line 72. What is the base year? I know it's in the methods when you dig carefully through them but the reader is left entirely guessing at this stage. Similarly, what does '2050 reference scenario'

mean? Again this is understood when the methods are read but could be made clearer/more accessible here.

N. Figure 1 legend, I suggest changing the acronym 'CC' to ICC so that it is consistent with the intended meaning of Impact of Climate Change. More generally, please consider changing the acronym CC to ICC throughout the document. Referring to CC alone is ambiguous - I did not know whether you were initially referring to the impact of scenarios ON climate change (eg how greenhouse gas emissions would increase or decrease) or the impact of climate change on the scenarios. Changing the acronym throughout should help with this. Further re Fig 1, I also suggest inserting the Base year (eg 20xx) and the reference year of 2050 into the X-axis labels. Similarly, I suggest indicating the base year and reference year in the Fig 1 caption in line 80. Essentially these and many other suggestions I'm making above and below are simply seeking greater clarity, transparency and consistency throughout the manuscript. I was left guessing far too often when reading the manuscript and was only able to discern intended meaning by carefully piecing things together.

O. Line 85. The phrasing "We also calculate scenarios ..." is awkward and not used previously. I suggest something like 'Scenarios were developed that...' and then 'Impacts of these scenarios were modeled ...'

P. Lines 87-88. The sentences that reads "Without these accompanying measures, no more than 20% conversion to organic production would be possible without larger increases in land demand (no CC assumed)." does not appear to be supported by results in Fig 1. My reading of Fig 1 seems to indicate that the two blue bars with 20% conversion to organic in 2050 are higher than the reference scenario. So there is no modeled scenario - not even the 20% conversion scenario - that does not result in increased land demand.

Q. Regarding headings etc in Figure 2 and Fig 3. X axis major heading needs to be more transparent. Consider extending it to 'High yield gap between conventional and organic agriculture'. X axis secondary heading 'Climate change impact' is ambiguous (see N above). Y axis heading '% Concentrates' is ambiguous/misleading (see 2 above).

R. Line 104. The sentence "N-surplus is balanced for organic conversion rates of 80%" is incredibly unclear. There is far too much understanding that is implied.

S. Lines 137-140. I suggest inserting parenthetical references to specific Figures that illustrate points made in this very information dense sentence. Help point the reader to where data are presented that support the points being made.

T. Line 142. I find the title of the section ambiguous as I initially thought Dietary Composition was a section that was addressing a model input rather than implications of the model outputs. Consider "Dietary Implications" or something along these lines.

U. Re Figure 4. Missing Y-Axis labels?

V. Re Figure 5. The resolution of the three panels is poor. This needs to be sharpened.

Furthermore, the use of the red and blue colour scheme here stands in general contrast to how you've used these colours in prior figures and makes it harder to interpret. In Figures 2 and 3, red was used to illustrate model outcomes that could be seen as 'bad' where as blue was used to illustrate model outcomes that could be seen as 'good'. Here in Fig 5 red is model outcomes with full impacts of climate change incorporated, and blue are model outcomes with zero impact of climate change.

W. Lines 221-214. The sentence that starts "Modelling the consequences of a global conversion to organic agriculture needs to be based on a comprehensive food systems perspective, ..." is a normative statement that is not supportable. You may deem it to be a superior approach or an approach that provides richer insight into ways of compensating for lower yield and other drawbacks of organic production but there is nothing in your analysis that objectively supports the argument that modeling HAS to be done this way.

X. Line 221. This is related to the larger issue #2 I raised above - what is actually being modeled and described when concentrate feed use is varied. Here the sentence is very misleading when you say "We have shown that the favourable environmental performance of concentrate-free livestock production and organic ...". What you have shown is that there is substantial environmental performance improvements possible when the absolute quantity of livestock production is reduced - a quantity that corresponds to that which will be produced using concentrate feeds in a business

as usual case in 2050. However as the sentence is written, it implies that all that needs changing is how the livestock is fed i.e. shift from concentrate to forages. You need to find a way to have the text more accurately reflect the specific key attributes of the model.

Y. Line 224. What physical/technical characteristics? This is very vague. Help the reader understand what's in your mind.

Z. Line 223. The range of 30-40% food waste is indicated but what was actually modeled? The range clearly comes from the literature but in this sentence, how can a range of losses then give rise to the very precise global average demand of 3028kcal/cap/day?

AA. Line 237. Consider modifying the end of the sentence to "... reductions livestock production and food wastage are simultaneously addressed."

BB. Re line 383. What is unclear in the methods description is from what starting point losses are you reducing from. The reductions are clear (25 and 50%) but not the starting value. Is it 30% or 40% or something in between these end points?

CC. Line 384. Related to major point #2, this is the very first place that I found an explicit reference to what you actually modeled - a reduction in animal product supply - when you varied the amount of concentrate feed supplied. The main article text was incredibly opaque on this critical issue.

DD. Lines 385-386. I'm confused re the scale of grass fed and concentrate feed fed ruminant production under different models. If the assumption is that ruminant production is entirely grass-fed and all monogastrics are fed on by-products (how is this defined by the way?), then why is there any need for concentrate feed production under a 50% reduction in concentrate feed production scenario?

In Supplemental Materials:

a. There is redundant text in lines 43-46 and then in lines 60-61. Try and reconcile.

b. Please try and keep the naming conventions for the model used consistent throughout. In some places it is SOL-m in other SOL model in others SOLm.

c. Line 124. I think you need the word 'the' between on and most.

d. Line 259 I think the word This at the start of the sentence should be The to make it work with the plural 'rates'.

e. Line 295 - Supplementary Table 1. In keeping with my observation in #2 above, please make the fourth column heading reflect that it is not just a reduction in concentrate feed production that is modeled but corresponding reductions in absolute livestock volumes.

Reviewer #3 (Remarks to the Author):

The study examines the option-space for using organic agriculture in combination with other strategies (i.e. wastage reduction and animal feed changes) and different scenarios of climate change impacts on crop yields to feed the human population in the year 2050. While the study contributes an important novel piece of research to the debate about whether organic agriculture could sustainably feed the world in 2050 and while the study has the potential to provide useful insights into this question, I do not recommend this study for publication at this point due to three major issues. The first issue pertains to inappropriate conclusions given the results of the study. The second issue pertains to poor delivery and poor structure and writing of the paper. Finally, the third issue pertains to insufficient details given in the methods and SI on data inputs, model parameters and equations used to estimate response variables under different scenarios.

The first major issue I have with this study is that the results of the analysis are interpreted assuming that organic agriculture should be part of a solution for sustainably feeding the world (and then the study examines scenarios of how organic agriculture could be combined with other strategies to avoid cropland expansion). But the study fails to clearly ask the question of whether organic agriculture would actually provide a more sustainable strategy to produce food compared to conventional (intensive) management. Yes, less-intensive (aka organic) agriculture is feasible

(in terms of food production) if it is combined with reductions in food demand (eg reduction in waste and changes in human or livestock diets). This is a no brainer. But the actual interesting question to ask is whether a scenario of 20, 40, 60, 80 or 100% organic agriculture would have reduced or increased environmental impacts (in terms of GHG emissions, N and P loss, water use, soil erosion etc.) compared to the counterfactual scenario of conventional (intensive) agriculture. The answer to this question is hidden in the results of the analysis conducted in this paper (as well as in some statements in the text, e.g. lines 209-211). But the authors do not highlight this comparison but instead base the main conclusion of their paper (i.e. organic agriculture is more sustainable than conventional agriculture) on how organic agriculture could be combined with other food system strategies.

By confusing and combining the question of how organic management could be combined with other strategies the authors base some of their conclusions on the sustainability of organic agriculture in feeding the world in 2050 on the impact of other strategies (i.e. changes in feed concentrates and food wastage reduction). It is perfectly acceptable (and actually very interesting) to examine organic agriculture in combination with other food system strategies like food wastage reduction etc. But when talking about the impact of organic agriculture per se it is important to separate the impact of organic management from these other strategies.

The authors state, for example, in the abstract that they show 'that organic agriculture could assure sufficient food availability in 2050, while reducing environmental impacts' (line 18-19). This is, from my understanding, an incorrect interpretation of the results of their analysis. (with the caveat, however, that I am unable to understand the key figure on environmental impacts, i.e. Fig. 5, due to unclear description, see comment below, but I am basing this on an interpretation of Fig. 1). Fig. 1 shows that all scenarios of organic agriculture would always require more agricultural area than the scenario of conventional agriculture (i.e. 0% organic agriculture). Given that natural ecosystems typically deliver ecosystem services (e.g. soil quality, climate regulation, water cycling) at much higher rates, and have considerably lower environmental impacts (e.g. in terms of N loss, soil erosion or GHG emissions) than any form of agricultural land use, I am assuming that environmental impacts are strongly correlated with land area required. From the results depicted in Fig. 2 I would therefore argue that the most sustainable strategy for feeding the human population in 2050 is a scenario of 0% organic agriculture, 50% food wastage reduction and 0% feed concentrates, as this would lead to a reduction in cropland area of 35% compared to the reference scenario (under no climate change). While if we keep other strategies (i.e. food wastage and feed concentrates) constant, then the scenario of 100% organic agriculture would be the most unsustainable scenario given that it would require 33-71% more cropland area compared to conventional agriculture.

The authors appear to base their conclusion that environmental impacts of organic agriculture are lower on the observation that organic agriculture can, when combined with other food system strategies (i.e. different feed concentrate and food waste scenarios), be carried out without an increase in land area. But this observation allows the conclusion that these other food system strategies can reduce cropland area required. The impact of organic agriculture is still one of increased cropland area (as clearly shown in Fig. 1 and clearly stated in paragraph lines 72-78).

Another issue is the strength of statements re the benefits of organic agriculture. On many environmental dimensions the current scientific evidence does not allow a conclusive assessment of whether organic agriculture actually performs better than conventional agriculture. But the authors make it appear (e.g. on lines 37-38) as if we had a strong confidence in the higher environmental performance of organic agriculture per unit area. While in lines 40-41 they make it appear as if performance per unit output is still generally positive, but more uncertain. This is not an entirely appropriate reading of the scientific literature. Even the literature cited by the authors (e.g. Tuomisto et al. 2012, ref 17) states that organic agriculture has a clear environmental benefit on some dimensions per unit land area (e.g. biodiversity, soil organic matter, N leaching, N₂O emissions, energy use) but the benefit is not clear on some other dimensions (e.g. P leaching, CH₄

emissions). While if per unit output impacts are accounted for (which are very important due to the strong negative environmental impacts of land use) the environmental benefit of organic management is either highly uncertain (e.g. for biodiversity) or potentially organic might even perform worse than conventional (e.g. for N leaching and N₂O emissions). This uncertainty and ambivalence in our understanding of the environmental performance of organic agriculture should be clearly stated in the paper.

The second major issue pertains to poor writing and structure of the paper. Based on the current text it is often difficult (and sometimes impossible) to understand what the authors did, and how to interpret results and figures.

It is, for example, not possible to understand what exactly one of the key figures of the paper, i.e. Fig. 5, shows. The x-axis appears to represent %impact relative to reference scenario (i.e. 0% organic agriculture, 100% concentrate, 0% waste reduction), but then the light shaded bars also appear to represent the same reference scenario (of 0% organic shares, according to the Figure legend). Does this figure imply that all environmental impacts (except for N surplus) are higher under organic management and food wastage and feed concentrate scenarios compared to the reference scenario of intensive (conventional) agriculture? As is I do not know how to interpret the figure, I am not able to assess the key results of this paper re environmental impacts.

The same applies to the section on Environmental impacts (lines 172ff) - given the information in the main text it is not possible to interpret the results presented there. Why does soil erosion increase under organic management (line 174ff; given that most studies show reduced soil erosion under organic management, see eg. Reganold et al. 1987, 10.1038/330370a0; Siegrist et al. 1998, doi:10.1016/S0167-8809(98)00113-3)? Why do the authors assume organic agriculture to rely on the same amount of P from non-renewable resources as conventional agriculture (line 177-178)? I would assume that organic agriculture would require fewer non-renewable P resources due to higher nutrient recycling rates (from animal manure, composts and crop residues) in organic systems. Also - how was water use in organic systems modeled in the paper? Organic regulations do not include any water regulations, and there is very limited research on water use in organic versus conventional systems. I therefore do not understand why water use is supposed to be lower under organic management (line 186ff). Finally - given that CO₂ emissions from deforestation (mostly for agriculture) currently accounts for ca. >11% of global anthropogenic GHG emissions, given that organic scenarios require considerable larger land areas, and given that N₂O and CH₄ emissions from organically managed agricultural soils are highly uncertain (see e.g. Skinner et al. 2014, doi:10.1016/j.scitotenv.2013.08.098, a paper that includes authors of this study), and given that the authors do not specify whether and how other emission categories (e.g. enteric livestock fermentation, manure storage) differ between organic and conventional scenarios (see section 1.5.5 in SI), I do not understand why GHG emissions from organic agriculture are supposed to be lower under organic management (line 183ff).

Similarly, it is difficult to understand the relevance of Figure 4 given the limited description of how diet was modelled in the main text. It is not possible to interpret the relevance of the results presented in the section on 'Dietary composition' (lines 142ff) without reading the methods at the end of the paper (or in the SI). Why do these diet changes matter? Are they a function of model prescriptions (e.g. fixing a ceiling for legume area under organic scenarios) or do they give us some indications about what diet changes will be caused by conversion to organic agriculture (or changes in feed concentrates)?

The paper also currently lacks a clear guiding structure. The introduction does not give a clear overview about what the reader is to expect from the rest of the paper, what concrete research questions were examined and what methods/scenarios used. Even the most fundamental aspect of the analysis - i.e. that the paper examines different scenarios to feed the human population in 2050, is not stated clearly in the introduction. While the reader is, for example, made to believe that the current study includes different diet scenarios in its analysis (e.g. line 51, 64/65), which it

does not directly (but only indirectly). While the issue of feed concentrates and why this matters or the question of climate change impacts on yields and cropland area requirements (which both are important components of the ensuing analysis) are not introduced in the introduction.

Finally, in consideration of recent efforts to enhance the reproducibility of scientific research, which are supported by Nature journals (see eg recent editorial from May 26, 2016) I highly encourage a more detailed description of the methods, as well as making the code of the SOL-m model and the full datasets used in the analysis available in online repositories (see e.g. doi:10.1038/515312a, doi:10.1038/514536a). As a minimum, tables showing the parameters used (e.g. for GHG emissions, N inputs or pesticide use), as well as tables showing the final values for different scenarios should be included in the SI. Right now (given the information and data provided in methods and in the SI) it is impossible to reproduce the analysis conducted in the paper or even to assess the assumptions underlying the model. The SI methods frequently refer to SI ref 44 (Schader et al. 2015) for more details on the SOL-m model used. This paper does not, however, provide any details on the organic scenarios (but only discusses methods used for different feed concentrate scenarios). Given the high uncertainty about the environmental performance of organic agriculture on many variables (see comment above) and the poor description of methods used and assumptions made by the SOL-m model on these uncertain processes in the methods section of the paper or the SI, it is not possible to assess whether the results of the SOL-m model presented here are valid, reliable or robust. I have added some examples of specific questions re methods used in the detailed comments below, but these are not complete at all. To truly understand the methods there are many more questions to be answered.

Detailed comments:

- line 16: Its not only the 'technical feasibility' but the feasibility in general (including agronomic, environmental, social, & economic feasibility) that is questioned. Suggest changing to 'its feasibility is contested'.
- line 33: suggest using 'holistic approaches' rather than 'systemic approaches'
- Lines 44-47: The studies that have attempted to quantify the N supply for organic agriculture do not only lack 'a detailed food systems approach' but they also lack a robust analysis of N availability from organic nutrient sources. The study by Badgley et al. (2007, ref 20) was criticised for over-estimating N availability from leguminous cover crops by Connor (2008, ref 18). While Connor (2008, 2013, refs 18 & 19) discussed the topic of N availability and provided a crude estimate, they do not carry out any real analysis. So I would rephrase: "Some authors complement the discussion on lower yields in organic agriculture with considerations on nutrient availability but none of these have to date carried out a robust analysis of nutrient availability from organic services (refs 18-20). In addition, these studies are lacking a detailed food systems approach and do not address animal feeding, consumption aspects and food loss and waste ("wastage")."
- line 51: by listing 'reductions in animal product consumption' here the reader is made to expect that the current paper examines different diet scenarios, which it does not. Rephrase.
- lines 51-53: I would suggest to cite references for each important food system strategy in the place where it is mentioned to support the choice of these particular strategies as important food system leverage points examined in this study. I would also suggest expanding briefly on why each of the strategies examined in this paper are relevant. This is especially pertinent for the strategy of changing feed concentrates, which is not that commonly discussed in the context of important strategies to feed the global population sustainably (and which is, I believe, only mentioned by ref 21 amongst the references cited here).
- line 55: rephrase as 'impacts of single farming systems'
- line 57: The von Lampe et al. (2014, ref 27) study on the AgMIP comparison does not include a representation of organic agriculture. If this sentence is to mean that only few global models have considered organic agriculture and the authors are meaning to cite those models that did include organic agriculture, then ref 27 should be removed. Otherwise the sentence should be rephrased.

Also peer-reviewed studies should be used wherever possible (i.e. ref 21 should be replaced by the more recent paper Erb et al. 2015, DOI: 10.1038/ncomms11382).

- lines 56-61: this concluding paragraphs of the introduction should give a brief summary of what the reader should expect in the remainder of the paper and what type of analysis was conducted in this study. Include clear statements here about the type of analysis and scenarios conducted with the SOL-model and the concrete research questions examined. Right now the results described in lines 72ff are difficult to interpret because the reader does not know what the analysis was about (without reading the methods at the end of the paper).

- line 62-63: awkward phrasing. If intensive agriculture comes with adverse environmental impacts this is not a 'solution'.

- line 64-65: This makes it appear as if the present study examined (1) changed feeding rations, (2) changed food consumption, and (3) changed wastage patterns. But in fact the paper only examines 1 and 3. Rephrase and be clear about the scope of the current analysis.

- line 72: define reference scenario.

- line 73/73: explain what low and high yield gap scenarios represent.

- line 74: briefly explain here (or at the end of the introduction) what type of climate change scenarios were used

- line 76: how is deforestation different from land occupation? The reader should be able to understand the gist of this without having to first read the methods at the end of the paper or in the SI.

- line 86: 'reductions of animal feed grown on arable land' - why does this matter? Need to discuss in introduction.

- line 88: abbreviations (CC) need to be explained where they are first used

- line 93-94: again, so far the reader has no idea what 'high yield gap' and 'low yield gap' means.

- line 104ff: again, the introduction needs to be set up so that the reader knows what type of response variables (i.e. cropland area requirements, N surplus, different environmental impacts) are examined in this analysis as well as some basic background on how these variables were examined. Right now the reader does not know what N surplus is about - Why is this a concern? What broad assumptions/methods were used to estimate this? Why does the N surplus of organic and conventional agriculture differ?

- line 104-105: instead of saying that the N surplus decreases (difficult to interpret) it would be better to say that a scenario of 100% conversion to organic agriculture leads to a N deficit.

- line 110: The exclusion of potential N sources from food waste and human waste needs to be discussed as a caveat of the model somewhere in the paper. Even though nutrients in food waste and human wastes are currently not re-used at very high rates in food production (estimates are around 5-9% in urban areas), some modeling studies have shown high potential recycling rates if urban wastes were used in agriculture (see eg. Faerge et al. 2001, doi:10.1016/S0304-3800(01)00233-2; Magid et al. 2006, doi:10.1016/j.ecoleng.2006.03.009).

- line 144-145: 'increased protein/calorie ratio' - what does this mean? Please explain.

- lines 142: it needs to be clearly stated how diets were modeled for the reader to be able to understand these results. Were the numbers of calories kept constant in different scenarios? Are different dietary compositions an outcome of the model (given livestock and legume production constraints imposed in the model in different scenarios)? Or were different diet compositions prescribed to examine different diet scenarios (e.g. vegan, vegetarian etc.)?

- lines 189: use 'non-synthetic pesticides' instead of 'critical substances' here and justify this statement better (i.e. explaining that organic management allows the use of some pesticides, and that some of these can potentially be harmful to the environment).

- line 241ff: too long and convoluted sentence. Rephrase.

- Fig. 1: this Fig. is difficult to interpret given the relatively simple message it is intended to deliver. I would consider showing bars as % change (or increase/decrease in ha cropland use) relative to base year.

- Fig. 2: in every line of this figure the numbers increase with increasing % organic area, except for the 25% wastage, 50% concentrate and medium climate change impact line (which changes from 4 to 0 and then to 5% area change under 0, 20 and 40% area organic respectively). Is this an error? If not, how do the authors explain this anomaly?

- Fig. 3: the colour scheme used here is confusing as both very high numbers and very low numbers are shaded in the same colour. I would use a different colours to denote high N surplus and high N deficit, as these are associated with very different problems (i.e. high N losses to the environment vs N limitation of crop production).
- Fig. 5: the line at 100% change is confusing; the axis should be centred around zero % change (i.e. no difference) to allow easier interpretation. Also - I am not sure how to interpret this Figure, see comment in major comments above.
- Fig. 5: Why is energy use examined separately from GHG emissions? Isn't energy use (e.g. for fertilizer production) already included in GHG emissions (see lines 217ff in SI)? Does energy use imply any other additional environmental impact that is separate from GHG emissions?
- SI, line 70: what is a 'yield gap' for livestock production? Does this mean lower livestock productivity (e.g. in milk yields) or lower livestock densities? Explain.
- SI, line 76: How is N fixation modelled in SOL-m? This is a variable with high uncertainty and should be discussed further. Also - it seems that the SOL-m model assumes that the additional legume crops grown under organic management are used for human consumption and contribute to calorie delivery? Is that true? Many legume crops in organic systems are, however, not harvested (for human consumption or animal feed) but only used as green manures that are ploughed into the soil at the end of the cropping season and provide fertility for the following crop. How does the SOL-m model deal with this issue?
- SI, line 152ff: How does this assumption of grassland area remaining constant with the scenarios of changes in concentrate feed? If livestock is fed less on concentrate feed, does this not necessarily imply an expansion of pastureland?
- SI, line 156ff: it is not clear to me from this description how N inputs to organic or conventional agriculture were estimated. Was any data on N application in organic versus conventional systems used? Or was N input based on estimated crop N demands?
- SI, lines 165-166: please show this comparison with literature values in a Table
- SI, line 212ff: include a table showing the parameters used for different emission factors, as well as a table showing the final emission values for different scenarios. It is not clear here where the parameters used for organic management come from. Did the authors assume similar emission factors for enteric fermentation for organically and conventionally managed livestock? Were emissions from processes and buildings (assuming this includes manure storage?) assumed to be the same between organic and conventional management?
- SI, line 217: abbreviations (GWP) need to be explained where they are first used; also include units for numbers shown.
- SI, line 224: show this comparison of baseline year GHG emissions from the SOL-m model with literature estimates in a table.
- SI, line 227ff: it is not clear from this description how water use between organic and conventional management was calculated.
- SI, line 230ff: please provide more information on the model used to estimate pesticide use, including the equations used. Also please provide the final values of pesticide use (maybe for different regions) in a Table, and ideally compare with values from the literature.
- SI, line 256ff: it is not clear how soil erosion differs between organic and conventional scenarios? Does the difference only depend on the cropland area occupied?

Authors' responses to reviewer comments:

1 Reviewer #1 (Remarks to the Author):

Dear reviewer,

*thank you very much for your very helpful and detailed comments and suggestions, which helped to substantially clarify and improve our manuscript. Please find below our answers to your comments, each time in **italic blue** print. Line numbers in our answers refer to the revised manuscript and SI text WITH track changes; the main revised manuscript and SI files are cleaned versions without track changes. For completeness, we have uploaded these two versions of the manuscript and the SI, to make all changes easily traceable.*

Reviewers' comments:

The scientific question raised in the paper on whether it is possible to satisfy 2050 FAO scenario food demand with global organic farming is very relevant and important in the context of sustainable future agriculture and the Sustainable Development Goals (SDGs). This scenario was introduced by Alexandaratos and Bruinsma and serves as the "standard" assumption for most groups, which deal with issues of future food security. I do not know of any publication that deals specifically with this relevant question as explicit as the submitted paper. The paper shows with a simple material flow model that organic farming under the assumptions used does not allow to satisfy demand without considerable expansion of farmland. Nevertheless a combination of organic farming, expansion, reduction of food waste and reduction of meat consumption may achieve the goal.

The basic assumption of the paper is that organic farming reduces yield by app. 20%. This assumption reduces and generalizes organic farming solely to a global factor for a loss in yield. This may be mainstream science for intense developed world farming practices. To generalize it to the global farming systems with its large variety of yield gaps and development stages is at least courageous. The paper lacks critical discussion to clarify whether this is a scientifically sound assumption, which I doubt at the moment.

It is true that we primarily referred this high yield gap in the main paper, but we also investigated how results change when adopting more optimistic assumptions on organic yields, that result in a yield gap of only 8% on average. The yield gap is differentiated between different commodity groups and based on recently published meta-analyses of the data available on yield comparisons. In the SI, section 1.2.2, we have some discussion on these different yield gap levels and the rationale to report results in the main paper for the most conservative choice only (while results for the lower yield gaps are displayed in the SI). For clarification, we have added a short discussion on this point in the main paper, referring to the SI (new line 135).

The paper then goes through a lengthy analysis of combinations of measures that combine the introduction of global organic farming practices with farmland expansion, reductions in food waste and change in consumption patterns. It shows that organic farming alone will not be able to satisfy global demand without unrealistic expansion of farmland. With the vastly expanded, now 4-dim. scenario space it is not a priori astonishing that the authors find solutions, which

allows to satisfy this changing future demand. It should be made transparent that these results are not trivial in the sense that four parameters (expansion, % organic farming, reduction of food waste, reduction of meat diet) will (almost) always create a solution to the given problem.

We have now thoroughly reorganized the story-line in the introduction, to also address the comments of other reviewers. On line 59ff, it now reads as follows, to make clear what we aim to do, and how the various combinations along the different dimensions have to be assessed, and that the results are not trivial: “Here we investigate the impacts of changes in the food system on a range of environmental and production indicators. The changes are a) reductions of livestock feed from arable land (i.e. food-competing feed) with corresponding reductions in animal numbers and products supply (and thus consumption) and in related resource use and environmental impacts^{25,26}; b) reductions of food wastage with correspondingly reduced production and production impacts¹⁰; and c) conversion to organic agriculture²². Our main interest is in assessing whether producing the same amount of food in terms of protein and calories with organic agriculture would lead to higher or lower impacts than the conventional counterfactual. This allows us to assess in which context of food system changes, and to what extent organic agriculture may contribute to more sustainable food systems.”

The elasticities that are used to determine the changes in food supply within this scenario space are not clearly explained in terms of their geographic diversity and, even more important, their stability in a future 2050 world. This has to carefully be clarified to the reader together with a careful analysis on how the result of the study depends on the assumptions on elasticity that were used.

We are aware of the high level of uncertainty that is connected to such future projections. Therefore, we calibrate the reference scenario 2050 to be identical with the business as usual projection from Alexandratos and Bruinsma (2012), with all assumptions on elasticities underlying their modelling work. For all scenarios, we start from this reference case. The alternative scenarios do not cover assumptions on elasticities, neither from supply nor demand side, as there is no economic market model behind the calculations. The resulting changes in environmental impacts and food availability are purely based on the changes in physical mass flows. In order to take into account the uncertainty that is connected to such an approach we used sensitivity analyses for the main factors. We changed the text to mention this explicitly (lines 111-113: “[...] adopting their assumptions on [...] and, implicitly, via their production and consumption structure, on underlying elasticities.”).

With these clarifications being successful I would like to see the proposed solution be carefully discussed with regard to its realism and impact on implementation in terms of e.g. geographical priorities.

Indeed, such discussion is relevant, but we focus on assessing the physical feasibility and adopt a global scope. To account for this, we added the following on lines 104-6: “Our analysis shows the necessary global changes, but we emphasize that structural change in the food system and the implementation of organic agriculture on any path to such increased shares of organic production will differ regionally and need to account for local and regional characteristics.”

Altogether, I think that the paper in its current form lacks scientific maturity and is not ready to be published in Nature Comms. It needs major revisions, critical discussion and clarification.

We have thoroughly rewritten the paper based on all of the reviewers' comments.

2 Reviewer #2 (Remarks to the Author):

Dear reviewer,

*thank you very much for your very helpful and detailed comments and suggestions, which helped to substantially clarify and improve our manuscript. Please find below our answers to your comments, each time in **italic blue** print. Line numbers in our answers refer to the revised manuscript and SI text WITH track changes; the main revised manuscript and SI files are cleaned versions without track changes. For completeness, we have uploaded these two versions of the manuscript and the SI, to make all changes easily traceable.*

Reviewers' comments:

I was very pleased to take on the review of this manuscript along with its associated supplemental materials. As the authors, I am keenly aware of the challenges posed globally by food systems given growing human demands and projected trajectories of production. We clearly need to find viable, lower impact alternative food systems if we are to limit substantial additional degradation of the biosphere. Consequently, I offer my feedback below in the context of wishing to strengthen your article and our collective understanding of the implications of alternative food systems. My comments below are organized into four major sections: general comments on the research, larger scale methodological or presentation issues, then finer-scaled editorial feedback on the main manuscript and finally finer-grained feedback on the text of the Supplemental materials.

General comments.

The manuscript represent an ambitious piece of research that helps us understand broad-scale environmental implications of ag and food system impacts out to 2050 under a range of variables, including: a) levels of organic agricultural practice adoption, b) effects of climate change on yields, c) levels of aggregate food losses along supply chains, and d) the scale of animal products included in diets. Results indicate that there are combinations of variable adoption that can deliver sufficient calories for projected human population out to 2050 while reducing the scale of the total land under cultivation as well as resource and environmental impacts in comparison with the business as usual reference scenario. While prior research has tackled aspects of what is modeled here, this paper is original in it's scope of variable inclusion and insight gained.

As addressed in detail below, I have some concerns with some of the data employed and how some data were handled (see specifically #3 and 4 below) and struggled for a long time in understanding what was actually being modeled in certain key aspects (see #2 below). More generally, from the number of fine-grained editorial comments I've made below, I think that the overall clarity and meaning could be greatly improved so that the value of the overall contribution of the research can be more easily discerned and understood.

Major issues:

1. There is an assumption evident throughout the manuscript, and it arises first in the title, and

it is that organic agriculture equals sustainable agriculture. While key attributes of organic ag certainly address issues of current local and global scale unsustainability, many other aspects are simply unaddressed. They may improve or get worse under organic management. Relatedly, it is implicit in the title and elsewhere that the word sustainability is used, that there is a definite sustainability threshold below which we no longer have to be concerned with how food is produced or consumed. Options to address this would be to either simply remove the word sustainability from the title or adding something like 'more' before it.

True; Impacts of organic agriculture being scaled up may be manifold. We tried to concentrate on the main factors that can be modelled with a relatively high certainty and that are dominant in the scientific discourse in the last years. Namely changes in yields were covered using latest results from global meta studies, nitrogen availability was modelled by a consistent mass-flow approach and other factors for live cycle inventories were taken into account, such as the ban of chem.-synthetic pesticides in organic production systems. Further changes may be changes in rotations, soil management, etc. We did not want to make strong assumptions here, as this depends very much on micro-economic decisions made by farmers and market effects, which cannot be seriously modelled for such long-term scenarios. Instead, we used sensitivity analysis for the main assumptions in order to test the robustness of our results. We changed by adding the word “more” to the title, and changing similarly throughout the text, where adequate (and also in one place in the SI, namely in the title).

2. I struggled for a long time to understand what was being modeled, and what results represented, in regards to reducing the percentage of concentrate feeds being fed to livestock. For example, in the text on page 6 and in the Y-axis second column of Figure 2 and elsewhere in the manuscript, the text refers to modeling the impact of reducing the percentage of concentrate fed to livestock. Initially, I assumed that this meant that the total production of livestock was left unchanged with only the feed input mix (forage vs concentrate) being varied. But of course, this made no sense given some of your findings. For example, that greenhouse gas emissions went DOWN with reduced levels of concentrate feeds fed to livestock. This makes no sense at all given that most life cycle assessment research, and particularly that on ruminants, clearly shows that higher levels of forages in diets drives UP GHG emissions. It was only after reading additional details in the supplemental materials that I realized that when you are modeling reductions in concentrate feed use, you are actually modeling absolute reductions in livestock volumes being produced. Indeed, this is the only way that some of your results make any sense at all. IF I now understand your work correctly, you need to be much clearer throughout the text and illustrative materials that what is now described as a percent concentrate feed share is actually a measure of the total amount of future livestock production that is being reduced.

This is true; the reduction in concentrates is directly linked to a reduction in animal numbers. In most product-related attributional LCAs this effect, which becomes clear only at global level, is often not considered, as marginal changes in production systems such global boundaries are do not need to be considered. Hence, there is a trade-off between resource efficiency of animal products and resource efficiency at human diets level, if it comes to concentrate use. In a way, it is a rebound effect of increasing the efficiency of livestock production. See also Schader, C., A. Muller, N. El-Hage Scialabba, J. Hecht and M. Stolze (2014). 'Comparing global and product-based LCA perspectives on environmental impacts of low-concentrate ruminant production', LCAFood, 8-11 October 2014, San Francisco, USA, pp. 1203-1209.). Thus, the model is strongly driven by the reductions in animal numbers, but how much those are reduced is not imposed directly from outside the model, but via the assumptions on how animals are fed and how much feed is available for them. We clarified this throughout the text, also changing “reduction in concentrates” and related formulations to “reductions in food-competing feed”, as this is the key of our argument, thus covering

reduction in forage maize, etc. as well, but not so in byproducts from food processing and in grasslands.

3. While it is a small part of the overall model and results, I don't understand at all how fish and seafood are treated in the model. Indeed they only show up in Figure 4 and while there is a clear reduction in the seeming amount of seafood and fish produced as a function of the amount of concentrate feeds being produced (100% of the business as usual 2050 reference year versus 0% relative to the reference year), how is this arrived at? The clear assumption one could make is that your model is assuming that a large portion of future seafood in the 2050 reference years seafood will be grown on concentrate feeds. But is this at all defensible? Right now ~50% of seafood continues to be sourced from fisheries and while this isn't growing, it's not going away any time soon. And within aquaculture, while fed culture systems are an important and growing sub-sector of current global seafood production, non-fed aquaculture is massive and will continue to dominate global farmed seafood production globally. Perhaps my concern can be dismissed given the much bigger picture that you're working on but the seeming conflict of model outcomes with my understanding of global seafood systems causes me concern re some of the underlying model assumptions.

We added a short paragraph to the Supplementary material providing further details on how we calculated the fish and seafood part. Basically, we assume a considerable share of total fish and seafood supply stemming from fed aquaculture in 2050, namely 45%. This is based on a range of FAO and OECD reports and projections, as explained in this short additional paragraph and the underlying more detailed description in the supplementary material of Schader et al. 2015 (section 1.4.1.7 in http://rsif.royalsocietypublishing.org/highwire/filestream/35634/field_highwire_adjunct_files/0/rsif20150891suppl.pdf).

4. I am concerned with the quality and currency of the data that you've used to model energy and GHG emission impacts from agriculture. It was only once I got to page 5 of the Supplemental materials and then checked the associated references (#s 72-74 in particular) that I realized that you are using results of LCA research on ag systems that are at least 10 years old throughout your models. This not just an issue of inputs and emissions to ag systems changing over time but that over the last decade there has been a rapid increase in the amount of research available to characterize energy inputs and GHG emissions from ag systems around the world. Ultimately this is a question of how confident can we be that models of the future based on limited datasets from the past are reasonable?

Yes, we completely agree that ecoinvent 2.0 data is not very comprehensive and a bit outdated. We cross-checked with ecoinvent 3 for some key categories, such as mineral fertilizer GWP and CED, and differences are minor to a few percent, but can also reach 10% in some cases. For some fertilisers ecoinvent 3 inventories result in higher impacts and for some vice versa. At the time we developed the model, ecoinvent 2 was the most reliable and transparent database we could use. But we do not think that it is beneficial in terms of quality of the results to collect many different LCA studies and average them. Instead we used a different approach which calculates inventories for each activity in each country based on different general data sources (see answer below). We compared the most influential inventory data components (e.g. mineral fertilizers) and think that the impact of including more recent data would be marginal. With respect to future technologies: The assumption behind our calculations is new technologies, e.g. GMOs or fractions of renewable energy in country specific electricity mixes, stay constant. Although the absolute level of energy use and GHG emissions will be affected through this, the relative performance of the scenarios against each other is likely to be the same (the relative performance is likely neither affected

much by the differences between ecoinvent 2 and 3). We explicitly mentioned this reservation with respect to absolute figures in our research in the SI.

Somewhat relatedly, how did you go about integrating different observations regarding energy inputs and GHG emission from different studies/sources. If you had say five observations of energy inputs to wheat, did you take a simple arithmetical average? Given that much of the data underpinning your models have been derived from multiple sources, I believe that there is need for some description of how these data were treated/combined before they were used in the model.

Indeed, different studies report different energy demand and GHG values for different products in different systems. Different methodological assumptions (e.g. allocation rules, system boundaries) and inventory data (e.g. mostly such LCA studies refer to a specific farm type or region in a country and are by no means representative) are responsible for this. For many combinations of countries and products no values exist at all. Therefore, we developed an approach, which has been recently presented at the LCAFood2016 Conference in Dublin (http://www.lcafood2016.org/wp-content/uploads/2016/10/LCA2016_BookOfAbstracts.pdf#354). This approach basically models the different components for energy use and GHG emissions based on transparent data sources and assumptions. For GHG emissions, from the agricultural sector we used mostly IPCC Guidelines, for inputs (fertilizer, infrastructure) we used mostly ecoinvent data and extrapolated it. We agree that there is some uncertainty connected to this procedure but we think it is much better than taking averages from non-representative studies from different countries. Furthermore, for the type of scenarios we model, it is unlikely that different assumptions may have a substantial impact on the relative performance of the scenarios.

Finer grained issues:

A. In line 13, consider adding "and expansion" after the word intensification

Done

B. In line 16, it's really more than organic agriculture's technical feasibility that is contested is it not? More generally, are questions regarding it's ability to address issues beyond the attributes at the core of organic production.

True, we reformulated as follows: "Organic agriculture has been proposed as a promising approach to achieve sustainable food systems, but its feasibility to contribute to such is contested."

C. In line 19, consider modifying the sentence to be something like "... assure sufficient food availability in 2050, while reducing a broad suite of environmental impacts.

This part of the abstract has been considerably changed due to other reviewers' comments.

D. Line 27. I may be a bit pedantic but is not the problem oversupply of reactive nitrogen? i.e. consider inserting the word reactive before nitrogen

Done.

E. Line 28, consider inserting the words 'land and' before water bodies. i.e. it would then read "..., eutrophication of land and water bodies, ..."

Done.

F. Line 38, consider inserting the word 'some' before environmental indicators as clearly not all evidence points to just positive outcomes as it acknowledged in the immediately

subsequent sentence (that land areas under organic ag may need to be greater as yields are often lower)

Done, we added "...a broad range of..."

G. Lines 44-47. The sentence that starts "Some authors complement..." is confusing. What is it that you are critiquing when you say that these studies (specifically references 18-20) lack a detailed food system approach? Is it that their purposes were flawed or lacking in some way? Put another way, the sentence reads like a criticism of this earlier work when you say, "... but these studies lack a detailed food systems approach...). My response is why should they have? Why is such an 'oversight' an oversight at all.

We rephrased this passage, also address comments from another reviewer; it is indeed rather a statement that such assessments are not yet done rather than a critique that those earlier studies should have done it but did not. We, therefore, in particular replaced "...lack a detailed..." with "...do not pursue a detailed....".

H. Lines 59-61. I know that there is more detail re the SOL-model in the methods etc, but I feel that more detail is needed here re how the model works and why it is useful. It is central to the analysis and as such, it's use needs more explication and defence.

We added a short paragraph to provide further information on how the model works and what the main aim is, and what this approach may deliver and what not.

I. Further re lines 60-61, what does the phrase "... tendency to reduced concentrate feed use." mean? Tendency is far too vague a word. Does organic ag actually employ less concentrate feed in livestock production or not. Does the SOL-model model this well or not. Finally, what is also particularly unclear to me is what does the tendency to reduced concentrate feed use mean for absolute levels of livestock (and aquaculture) production? Does it mean that these go down as well or just that the use of concentrate feeds is lower in organic ag as modeled but there is then a greater dependency on forages?

We changed "a tendency to reduced" to "lower"; certified organic agriculture has in general upper limits on concentrate use for ruminants and use of imported feedstuff, that differ between countries; though the EU is much less strict on this than the Swiss Biosuisse-label, for example. However, it is in the principles of organic agriculture to rather feed grass to ruminants and to not use imported arable crops as feed. In the SOL-model, reduced concentrate feed also covers a reduction in other feed from arable land (e.g. forage maize), since the main focus is on addressing the potential feed-food competition for land, which does not arise on grasslands that cannot be used for direct food production. We changed formulations throughout the manuscript to clarify that "reduced concentrates" refers to "food-competing feed", i.e. "feed from arable land"; by-products from food production, for example, are not reduced.

J. Line 62. Here you clearly transition to your results but the phrasing is weak. I would suggest starting the sentence more directly with something like "Our results show ..."

Done

K. Line 63. What does "becomes feasible" mean? Speak more directly please. Something like perhaps "..., and less intensive agriculture (organic in this case) also represents a means of feeding 9+ billion people in 2050 if it is combined..."

We reformulated as follows: "...and less intensive agriculture (organic in this case) can contribute to feeding more than 9 billion people in 2050 sustainably in a context of

complementary changes in the global food system, namely -"

L. Line 69. Again I may be a pedant but I suggest inserting 'improved' before soil fertility and reduced before pesticide use.

Done

M. Line 72. What is the base year? I know it's in the methods when you dig carefully through them but the reader is left entirely guessing at this stage. Similarly, what does '2050 reference scenario' mean? Again this is understood when the methods are read but could be made clearer/more accessible here.

Done, we added the years over which the average is taken to form the base year and we added a short explanation and the reference for the 2050 scenario, making explicit that it is from the FAO.

N. Figure 1 legend, I suggest changing the acronym 'CC' to ICC so that it is consistent with the intended meaning of Impact of Climate Change. More generally, please consider changing the acronym CC to ICC throughout the document. Referring to CC alone is ambiguous - I did not know whether you were initially referring to the impact of scenarios ON climate change (eg how greenhouse gas emissions would increase or decrease) or the impact of climate change on the scenarios. Changing the acronym throughout should help with this. Further re Fig 1, I also suggest inserting the Base year (eg 20xx) and the reference year of 2050 into the X-axis labels. Similarly, I suggest indicating the base year and reference year in the Fig 1 caption in line 80. Essentially these and many other suggestions I'm making above and below are simply seeking greater clarity, transparency and consistency throughout the manuscript. I was left guessing far too often when reading the manuscript and was only able to discern intended meaning by carefully piecing things together.

We changed the figure and caption and the text as suggested

O. Line 85. The phrasing "We also calculate scenarios ..." is awkward and not used previously. I suggest something like 'Scenarios were developed that...' and then 'Impacts of these scenarios were modeled ...'

We changed as suggested for the first point; the part of the text the second point refers to has been rephrased in line with the bigger reorganization of the text, as already mentioned above.

P. Lines 87-88. The sentences that reads "Without these accompanying measures, no more than 20% conversion to organic production would be possible without larger increases in land demand (no CC assumed)." does not appear to be supported by results in Fig 1. My reading of Fig 1 seems to indicate that the two blue bars with 20% conversion to organic in 2050 are higher than the reference scenario. So there is no modeled scenario - not even the 20% conversion scenario - that does not result in increased land demand.

We thoroughly rephrased this part and it now clearly states that increasing shares of organic production always go along with increased land demand if compared to a fully conventional situation.

Q. Regarding headings etc in Figure 2 and Fig 3. X axis major heading needs to be more transparent. Consider extending it to 'High yield gap between conventional and organic agriculture'. X axis secondary heading 'Climate change impact' is ambiguous (see N above). Y axis heading '% Concentrates' is ambiguous/misleading (see 2 above).

We changed these headings (cf. also our answer to 2 above, where we indicate that we replaced "concentrate shares" with "reductions in food-competing feed" throughout the

paper and in the figures).

R. Line 104. The sentence "N-surplus is balanced for organic conversion rates of 80%" is incredibly unclear. There is far too much understanding that is implied.

We reformulated for clarification; it now reads as follows: "Due to reduced N inputs from mineral fertilizers (and somewhat counter-acted by increased legume shares), the N-surplus is reduced with increasing shares of organic production and reaches a balanced level for an organic share of 80%"

S. Lines 137-140. I suggest inserting parenthetical references to specific Figures that illustrate points made in this very information dense sentence. Help point the reader to where data are presented that support the points being made.

Done

T. Line 142. I find the title of the section ambiguous as I initially thought Dietary Composition was a section that was addressing a model input rather than implications of the model outputs. Consider "Dietary Implications" or something along these lines.

We changed as suggested to "Dietary implications of the different scenarios"

U. Re Figure 4. Missing Y-Axis labels?

We added them (% shares in per-capita protein supply)

V. Re Figure 5. The resolution of the three panels is poor. This needs to be sharpened. Furthermore, the use of the red and blue colour scheme here stands in general contrast to how you've used these colours in prior figures and makes it harder to interpret. In Figures 2 and 3, red was used to illustrate model outcomes that could be seen as 'bad' where as blue was used to illustrate model outcomes that could be seen as 'good'. Here in Fig 5 red is model outcomes with full impacts of climate change incorporated, and blue are model outcomes with zero impact of climate change.

We improved the resolution of the figure and changed coloring to be neutral regarding "good/bad" outcomes as suggested.

W. Lines 221-214. The sentence that starts "Modelling the consequences of a global conversion to organic agriculture needs to be based on a comprehensive food systems perspective, ..." is a normative statement that is not supportable. You may deem it to be a superior approach or an approach that provides richer insight into ways of compensating for lower yield and other drawbacks of organic production but there is nothing in your analysis that objectively supports the argument that modeling HAS to be done this way.

We changed as follows: "To be able to comprehensively assess the potential and challenges of a global conversion to organic agriculture, modelling the consequences of such a conversion needs..."

X. Line 221. This is related to the larger issue #2 I raised above - what is actually being modeled and described when concentrate feed use is varied. Here the sentence is very misleading when you say "We have shown that the favourable environmental performance of concentrate-free livestock production and organic ...". What you have shown is that there is substantial environmental performance improvements possible when the absolute quantity of livestock production is reduced - a quantity that corresponds to that which will be produced using concentrate feeds in a business as usual case in 2050. However as the sentence is written, it implies that all that needs changing is how the livestock is fed i.e. shift from

concentrate to forages. You need to find a way to have the text more accurately reflect the specific key attributes of the model.

We have incorporated this aspect here, and changed to “We have shown that the favourable environmental performance of reduced animal numbers in livestock production that is free from food-competing feed and organic agriculture can be combined....” - we also changed the text at other points, where needed, to clarify this aspect, emphasizing that the reduction of food-competing feed directly relates to a reduction in animal numbers that ultimately drive the environmental advantages. However, it is important to note, as mentioned in the paper, that this reduction in animal numbers is not imposed exogenously, but that it is derived from the strategy to feed animals only on feed resources that do not compete with food production, i.e. on grasslands and byproducts, etc. – the availability of those then drives the animal numbers.

Y. Line 224. What physical/technical characteristics? This is very vague. Help the reader understand what's in your mind.

We added, after “characteristics”, the following: “..., namely by restricting feed supply to energy and protein that stem from resources that cannot be utilized for food production directly, such as grasslands and a range of processing by-products and wastes.”

Z. Line 223. The range of 30-40% food waste is indicated but what was actually modeled? The range clearly comes for the literature but in this sentence, how can a range of losses then give rise to the very precise global average demand of 3028kcal/cap/day?

The precise average demand is taken from the 2050 projections from Alexandratos and Bruinsma 2012. They do not explicitly account for wastage but derived this from their own modelling runs based on assumptions on future income levels, population growth, observed supply levels, and elasticities between commodity groups as well as price elasticities of those. The wastage level is taken from another source and more explicitly comes with large uncertainties. In this sense, it is not a range of losses that gives rise to this precise estimate, it is rather that their models give such a precise estimate, but that a range of percentages thereof is food wastage. We reformulated somewhat to make this clearer.

AA. Line 237. Consider modifying the end of the sentence to "... reductions livestock production and food wastage are simultaneously addressed."

Done, but we kept the feed reduction explicit as well; it now reads as follows: “...and significant reductions of food-competing feed use and thus livestock production, and food wastage were simultaneously addressed.”

BB. Re line 383. What is unclear in the methods description is from what starting point losses are you reducing from. The reductions are clear (25 and 50%) but not the starting value. Is it 30% or 40% or something in between these end points?

We added the information on what it refers to (the values from the FAO that largely lie between 30 – 40 % on average, but differ by commodity group and region – we used this data with this differentiation as a basis).

CC. Line 384. Related to major point #2, this is the very first place that I found an explicit reference to what you actually modeled - a reduction in animal product supply - when you varied the amount of concentrate feed supplied. The main article text was incredibly opaque on this critical issue.

We changed throughout the text to make this clear from the beginning

DD. Lines 385-386. I'm confused re the scale of grass fed and concentrate feed fed ruminant

production under different models. If the assumption is that ruminant production is entirely grass-fed and all monogastrics are fed on by-products (how is this defined by the way?), then why is there any need for concentrate feed production under a 50% reduction in concentrate feed production scenario?

We reformulated for clarification, making explicit that a 100% reduction in concentrates (or as formulated now, food-competing feed) means entirely grass-fed ruminants and monogastrics being fed on by-products only; while with a 50% reduction, half the food-competing feed from the reference scenario are still there to be used as feed. By-products are identified on the basis of the FAOSTAT commodity trees and their availability and utilization for feed is taken from the detailed data underlying the food balance sheets. Soybean meal thereby is defined to be a main product, not a by-product, different to other oil-crop meals.

In Supplemental Materials:

a. There is redundant text in lines 43-46 and then in lines 60-61. Try and reconcile.

Done; we dropped the repetition in lines 60-61.

b. Please try and keep the naming conventions for the model used consistent throughout. In some places it is SOL-m in other SOL model in others SOLm.

Done; we changed to Sol-model throughout the paper and the SM.

c. Line 124. I think you need the word 'the' between on and most.

Yes, we added as suggested.

d. Line 259 I think the word This at the start of the sentence should be The to make it work with the plural 'rates'.

Yes, we changed as suggested.

e. Line 295 - Supplementary Table 1. In keeping with my observation in #2 above, please make the fourth column heading reflect that it is not just a reduction in concentrate feed production that is modeled but corresponding reductions in absolute livestock volumes.

We changed this by making explicit that not only feed supply is changed, but, in consequence, animal numbers change as well.

3 Reviewer #3 (Remarks to the Author):

Dear reviewer,

*thank you very much for your very helpful and detailed comments and suggestions, which helped to substantially clarify and improve our manuscript. Please find below our answers to your comments, each time in **italic blue** print. Line numbers in our answers refer to the revised manuscript and SI text WITH track changes; the main revised manuscript and SI files are cleaned versions without track changes. For completeness, we have uploaded these two versions of the manuscript and the SI, to make all changes easily traceable.*

Reviewers' comments:

The study examines the option-space for using organic agriculture in combination with other strategies (i.e. wastage reduction and animal feed changes) and different scenarios of climate change impacts on crop yields to feed the human population in the year 2050. While the study contributes an important novel piece of research to the debate about whether organic agriculture could sustainably feed the world in 2050 and while the study has the potential to provide useful insights into this question, I do not recommend this study for publication at this point due to three major issues. The first issue pertains to inappropriate conclusions given the results of the study. The second issue pertains to poor delivery and poor structure and writing of the paper. Finally, the third issue pertains to insufficient details given in the methods and SI on data inputs, model parameters and equations used to estimate response variables under different scenarios.

The first major issue I have with this study is that the results of the analysis are interpreted assuming that organic agriculture should be part of a solution for sustainably feeding the world (and then the study examines scenarios of how organic agriculture could be combined with other strategies to avoid cropland expansion). But the study fails to clearly ask the question of whether organic agriculture would actually provide a more sustainable strategy to produce food compared to conventional (intensive) management. Yes, less-intensive (aka organic) agriculture is feasible (in terms of food production) if it is combined with reductions in food demand (eg reduction in waste and changes in human or livestock diets). This is a no brainer. But the actual interesting question to ask is whether a scenario of 20, 40, 60, 80 or 100% organic agriculture would have reduced or increased environmental impacts (in terms of GHG emissions, N and P loss, water use, soil erosion etc.) compared to the counterfactual scenario of conventional (intensive) agriculture. The answer to this question is hidden in the results of the analysis conducted in this paper (as well as in some statements in the text, e.g. lines 209-211). But the authors do not highlight this comparison but instead base the main conclusion of their paper (i.e. organic agriculture is more sustainable than conventional agriculture) on how organic agriculture could be combined with other food system strategies.

True, indeed; thank you for pointing this out. We totally changed the narrative to change the focus, as suggested.

By confusing and combining the question of how organic management could be combined with other strategies the authors base some of their conclusions on the sustainability of organic agriculture in feeding the world in 2050 on the impact of other strategies (i.e. changes in feed concentrates and food wastage reduction). It is perfectly acceptable (and actually very interesting) to examine organic agriculture in combination with other food system strategies like food wastage reduction etc. But when talking about the impact of organic agriculture per se it is important to separate the impact of organic management from these other strategies. *Cf. above; we changed the text throughout the paper to reflect this. The impact of organic agriculture is separated from the other strategies and thus made transparent via the different combinations of scenarios.*

The authors state, for example, in the abstract that they show 'that organic agriculture could assure sufficient food availability in 2050, while reducing environmental impacts' (line 18-19). This is, from my understanding, an incorrect interpretation of the results of their analysis. (with the caveat, however, that I am unable to understand the key figure on environmental impacts, i.e. Fig. 5, due to unclear description, see comment below, but I am basing this on an interpretation of Fig. 1). Fig. 1 shows that all scenarios of organic agriculture would always require more agricultural area than the scenario of conventional agriculture (i.e. 0% organic agriculture). *True, we revised the abstract and main body of the text accordingly, cf. also comments and replies above.*

Given that natural ecosystems typically deliver ecosystem services (e.g. soil quality, climate regulation, water cycling) at much higher rates, and have considerably lower environmental impacts (e.g. in terms of N loss, soil erosion or GHG emissions) than any form of agricultural land use, I am assuming that environmental impacts are strongly correlated with land area required. From the results depicted in Fig. 2, I would therefore argue that the most sustainable strategy for feeding the human population in 2050 is a scenario of 0% organic agriculture, 50% food wastage reduction and 0% feed concentrates, as this would lead to a reduction in cropland area of 35% compared to the reference scenario (under no climate change). While if we keep other strategies (i.e. food wastage and feed concentrates) constant, then the scenario of 100% organic agriculture would be the most unsustainable scenario given that it would require 33-71% more cropland area compared to conventional agriculture. *This is true for land use, and for other environmental impacts of livestock production systems. We changed the text accordingly (cf. replies above). However, for other environmental impacts the resource efficiency in organic crop production systems is not necessarily lower than in conventional agriculture, e.g. N, surplus, cf. figure 3) a higher share of organic agriculture performs better than a purely conventional production system. This is consistent e.g. with the review by Meier, M. S., F. Stoessel, N. Jungbluth, R. Juraske, C. Schader and M. Stolze (2015). 'Environmental impacts of organic and conventional agricultural products—Are the differences captured by life cycle assessment?'. *Journal of environmental management* 149, pp. 193-208.*

The authors appear to base their conclusion that environmental impacts of organic agriculture are lower on the observation that organic agriculture can, when combined with other food system strategies (i.e. different feed concentrate and food waste scenarios), be carried out without an increase in land area. But this observation allows the conclusion that these other food system strategies can reduce cropland area required. The impact of organic agriculture is still one of increased cropland area (as clearly shown in Fig. 1 and clearly stated in paragraph lines 72-78).

We changed the text thoroughly to make this clear and make more explicit the higher land demand of organic agriculture, cf. also above.

Another issue is the strength of statements re the benefits of organic agriculture. On many environmental dimensions the current scientific evidence does not allow a conclusive assessment of whether organic agriculture actually performs better than conventional agriculture. But the authors make it appear (e.g. on lines 37-38) as if we had a strong confidence in the higher environmental performance of organic agriculture per unit area. While in lines 40-41 they make it appear as if performance per unit output is still generally positive, but more uncertain. This is not an entirely appropriate reading of the scientific literature. Even the literature cited by the authors (e.g. Tuomisto et al. 2012, ref 17) states that organic agriculture has a clear environmental benefit on some dimensions per unit land area (e.g. biodiversity, soil organic matter, N leaching, N₂O emissions, energy use) but the benefit is not clear on some other dimensions (e.g. P leaching, CH₄ emissions). While if per unit output impacts are accounted for (which are very important due to the strong negative environmental impacts of land use) the environmental benefit of organic management is either highly uncertain (e.g. for biodiversity) or potentially organic might even perform worse than conventional (e.g. for N leaching and N₂O emissions). This uncertainty and ambivalence in our understanding of the environmental performance of organic agriculture should be clearly stated in the paper.

*You are right. Scientific literature is not conclusive in this respect (see comment above). We have reformulated to take up this criticism. It now reads “The positive performance of organic agriculture regarding **a range of** environmental indicators...” and we also added a recent reference adding to the methodological discussion on comparing organic and conventional agriculture, as well as updating the results when compiling and aggregating available studies (Meier et al. 2015). We then also rephrased in the next section, which reads now “...and ~~it is argued that~~ environmental benefits of organic agriculture are less pronounced **and partly ambiguous** per unit of product...”*

The second major issue pertains to poor writing and structure of the paper. Based on the current text it is often difficult (and sometimes impossible) to understand what the authors did, and how to interpret results and figures.

We thoroughly rewritten the paper to clarify and improve the narrative throughout the paper.

It is, for example, not possible to understand what exactly one of the key figures of the paper, i.e. Fig. 5, shows. The x-axis appears to represent % impact relative to reference scenario (i.e. 0% organic agriculture, 100% concentrate, 0% waste reduction), but then the light shaded bars also appear to represent the same reference scenario (of 0% organic shares, according to the Figure legend). Does this figure imply that all environmental impacts (except for N surplus) are higher under organic management and food wastage and feed concentrate scenarios compared to the reference scenario of intensive (conventional) agriculture? As is I do not know how to interpret the Figure, I am not able to assess the key results of this paper re environmental impacts.

The light bars indicate the changes when going from 0% organic (light shaded) to 100% organic (dark shaded), GIVEN the reduction in food wastage and food-competing feed as indicated in the labels for each panel (e.g. “50% food wastage reduction” and “0% conc. feed”, etc.). Therefore, the light-blue areas in the panel top left equal the 100%-line, i.e. the reference scenario; in the other panels, however, the light-blue line represents 0% organic, but not anymore the situation of the reference scenario regarding food-competing feed and wastage. That is why the light blue bars are always equal (top left) or below the 100%-line that represents the reference scenario. The light orange bar indicates the same, but with

impacts of climate change on yields, therefore it is often above the 100% line. We changed the legend as follows: “[...] the bars show the range of impacts for 0% (light colour) to 100% organic shares (dark colour) under the respective reduction of food-competing feed and wastage as indicated for each panel;” We also made clear that the bars refer to the IMPACTS relative to the reference scenario, which is assigned 100%, and not to the CHANGE in impacts, as originally formulated; this has also been confusing (cf. also the comment further down).

The same applies to the section on Environmental impacts (lines 172ff) - given the information in the main text it is not possible to interpret the results presented there. Why does soil erosion increase under organic management (line 174ff; given that most studies show reduced soil erosion under organic management, see eg. Reganold et al. 1987, 10.1038/330370a0; Siegrist et al. 1998, doi:10.1016/S0167-8809(98)00113-3)? *Some of these aspects are explained in the SI; section 1.5; soil erosion, for example, has been assumed to be similar in organic and conventional systems, as we aimed not to overestimate potential positive impacts of organic agriculture.*

Why do the authors assume organic agriculture to rely on the same amount of P from non-renewable resources as conventional agriculture (line 177-178)? I would assume that organic agriculture would require fewer non-renewable P resources due to higher nutrient recycling rates (from animal manure, composts and crop residues) in organic systems. *As above, we addressed P conservatively; we mentioned that in organic agriculture, non-renewable P likely is lower, as the organic farmers additionally take P from recycling plant biomass from compost and crop residues into account (manure is applied in both systems), but there are large uncertainties related to that and we thus refrained from reflecting this in our calculations.*

Also - how was water use in organic systems modeled in the paper? Organic regulations do not include any water regulations, and there is very limited research on water use in organic versus conventional systems. I therefore do not understand why water use is supposed to be lower under organic management (line 186ff).

We reported water use to be same in both production systems, as there is indeed only very limited research on water use in organic vs. conventional systems; we thus did not model it differently for the two systems; small but negligible differences are due to different crop area shares in the two systems and thus depends on the differences of water use between crops, not between the production systems.

Finally - given that CO₂ emissions from deforestation (mostly for agriculture) currently accounts for ca. >11% of global anthropogenic GHG emissions, given that organic scenarios require considerable larger land areas, and given that N₂O and CH₄ emissions from organically managed agricultural soils are highly uncertain (see e.g. Skinner et al. 2014, doi:10.1016/j.scitotenv.2013.08.098, a paper that includes authors of this study), and given that the authors do not specify whether and how other emission categories (e.g. enteric livestock fermentation, manure storage) differ between organic and conventional scenarios (see section 1.5.5 in SI), I do not understand why GHG emissions from organic agriculture are supposed to be lower under organic management (line 183ff).

Differences in GHG emissions are driven by differences in N inputs that are lower in the organic system, resulting in correspondingly lower soil N₂O emissions from N fertilizer application. This is due to using IPCC methods which are mainly driven by total N inputs. Thus, the total N input is more important than the total area used and the lower per-area emissions are not compensated by higher area use. The livestock sector has been modelled

using the same IPCC Guidelines both for organic and conventional production systems. Finally, the systems show differences in the production emissions from synthetic N-fertilizer production as those do not occur in the organic system.

Similarly, it is difficult to understand the relevance of Figure 4 given the limited description of how diet was modelled in the main text. It is not possible to interpret the relevance of the results presented in the section on 'Dietary composition' (lines 142ff) without reading the methods at the end of the paper (or in the SI). Why do these diet changes matter? Are they a function of model prescriptions (e.g. fixing a ceiling for legume area under organic scenarios) or do they give us some indications about what diet changes will be caused by conversion to organic agriculture (or changes in feed concentrates)?

We added some explanation and motivation in these paragraphs (and briefly also in the introduction), to better explain the results and why we present them in this paper.

The paper also currently lacks a clear guiding structure. The introduction does not give a clear overview about what the reader is to expect from the rest of the paper, what concrete research questions were examined and what methods/scenarios used. Even the most fundamental aspect of the analysis - i.e. that the paper examines different scenarios to feed the human population in 2050, is not stated clearly in the introduction. While the reader is, for example, made to believe that the current study includes different diet scenarios in its analysis (e.g. line 51, 64/65), which it does not directly (but only indirectly). While the issue of feed concentrates and why this matters or the question of climate change impacts on yields and cropland area requirements (which both are important components of the ensuing analysis) are not introduced in the introduction.

We changed the text thoroughly - and the introduction in particular - to provide better guidance and to more explicitly reflect these key points.

Finally, in consideration of recent efforts to enhance the reproducibility of scientific research, which are supported by Nature journals (see eg recent editorial from May 26, 2016) I highly encourage a more detailed description of the methods, as well as making the code of the SOL-m model and the full datasets used in the analysis available in online repositories (see e.g. doi:10.1038/515312a, doi:10.1038/514536a). As a minimum, tables showing the parameters used (e.g. for GHG emissions, N inputs or pesticide use), as well as tables showing the final values for different scenarios should be included in the SI. Right now (given the information and data provided in methods and in the SI) it is impossible to reproduce the analysis conducted in the paper or even to assess the assumptions underlying the model. The SI methods frequently refer to SI ref 44 (Schader et al. 2015) for more details on the SOL-m model used. This paper does not, however, provide any details on the organic scenarios (but only discusses methods used for different feed concentrate scenarios). Given the high uncertainty about the environmental performance of organic agriculture on many variables (see comment above) and the poor description of methods used and assumptions made by the SOL-m model on these uncertain processes in the methods section of the paper or the SI, it is not possible to assess whether the results of the SOL-m model presented here are valid, reliable or robust. I have added some examples of specific questions re methods used in the detailed comments below, but these are not complete at all. To truly understand the methods there are many more questions to be answered.

We put the model code, data and results on an ftp-Server (cf. the link in the main paper, after the Acknowledgments, right before the methods section). We also added some further details to the description of the organic scenarios in table S6 in the SI (page 14 in the SI). See also the answers to the related comments above/below. GHG emissions, N inputs and pesticide use are calculated as described in the respective paragraphs in the SI, where we added some

further details.

Detailed comments:

- line 16: Its not only the 'technical feasibility' but the feasibility in general (including agronomic, environmental, social, & economic feasibility) that is questioned. Suggest changing to 'its feasibility is contested'.

Done, we changed to “its feasibility to contribute to more sustainable food systems is contested”

- line 33: suggest using 'holistic approaches' rather than 'systemic approaches'

Done

- Lines 44-47: The studies that have attempted to quantify the N supply for organic agriculture do not only lack 'a detailed food systems approach' but they also lack a robust analysis of N availability from organic nutrient sources. The study by Badgley et al. (2007, ref 20) was criticised for over-estimating N availability from leguminous cover crops by Connor (2008, ref 18). While Connor (2008, 2013, refs 18 & 19) discussed the topic of N availability and provided a crude estimate, they do not carry out any real analysis. So I would rephrase: "Some authors complement the discussion on lower yields in organic agriculture with considerations on nutrient availability but none of these have to date carried out a robust analysis of nutrient availability from organic services (refs 18-20). In addition, these studies are lacking a detailed food systems approach and do not address animal feeding, consumption aspects and food loss and waste ("wastage")."

We rephrased this passage adding the part on the lack of robust analysis of N availability; we further reformulated somewhat based on comments from another reviewer.

- line 51: by listing 'reductions in animal product consumption' here the reader is made to expect that the current paper examines different diet scenarios, which it does not. Rephrase. *We changed “[...] such as reductions of livestock feed from arable land (“concentrates”), reductions of food wastage, and reductions in animal product consumption.” to “[...] such as reductions of livestock feed from arable land (i.e. “food-competing feed”) with corresponding reductions in animal product supply and consumption and reductions of food wastage.” (we changed from concentrate feed to “food-competing feed” throughout the text due to a comment by reviewer 2)*

- lines 51-53: I would suggest to cite references for each important food system strategy in the place where it is mentioned to support the choice of these particular strategies as important food system leverage points examined in this study. I would also suggest expanding briefly on why each of the strategies examined in this paper are relevant. This is especially pertinent for the strategy of changing feed concentrates, which is not that commonly discussed in the context of important strategies to feed the global population sustainably (and which is, I believe, only mentioned by ref 21 amongst the references cited here).

In the revised introduction, we have added some more explanation of these strategies and added the references right to where the respective strategies are mentioned (line 59 ff).

- line 55: rephrase as 'impacts of single farming systems'

We changed to “...impacts per unit output”.

- line 57: The von Lampe et al. (2014, ref 27) study on the AgMIP comparison does not include a representation of organic agriculture. If this sentence is to mean that only few global

models have considered organic agriculture and the authors are meaning to cite those models that did include organic agriculture, then ref 27 should be removed. Otherwise the sentence should be rephrased. Also peer-reviewed studies should be used wherever possible (i.e. ref 21 should be replaced by the more recent paper Erb et al. 2015, DOI: 10.1038/ncomms11382).

We changed as suggested

- lines 56-61: this concluding paragraphs of the introduction should give a brief summary of what the reader should expect in the remainder of the paper and what type of analysis was conducted in this study. Include clear statements here about the type of analysis and scenarios conducted with the SOL-model and the concrete research questions examined. Right now the results described in lines 72ff are difficult to interpret because the reader does not know what the analysis was about (without reading the methods at the end of the paper).

We thoroughly reformulated and reorganized the whole introduction and this part of the introduction in particular and these aspects should now be clearer.

- line 62-63: awkward phrasing. If intensive agriculture comes with adverse environmental impacts this is not a 'solution'.

We changed "solution" to "approach"

- line 64-65: This makes it appear as if the present study examined (1) changed feeding rations, (2) changed food consumption, and (3) changed wastage patterns. But in fact the paper only examines 1 and 3. Rephrase and be clear about the scope of the current analysis.

We changed as suggested; it reads now "...changed feeding rations and correspondingly reduced animal numbers (and thus reduced animal product consumption) and changed wastage patterns."

- line 72: define reference scenario.

We added "[...reference scenario] describing agriculture as forecast by the FAO, adopting their assumptions on yield increase, cropping intensities and regional dietary change" and referred to Alexandratos and Bruinsma 2012.

- line 73/73: explain what low and high yield gap scenarios represent.

We changed to "for low yield gaps (8% lower organic yields on average) to high yield gaps (on average 25% lower), as reported in the literature^{16,20}).

- line 74: briefly explain here (or at the end of the introduction) what type of climate change scenarios were used

Done, we added an explanation as follows: "...adverse effects of climate change (CC) on yields (modelled by reduced yield increases till 2050, down to zero increases for strong CC impacts) are considered"

- line 76: how is deforestation different from land occupation? The reader should be able to understand the gist of this without having to first read the methods at the end of the paper or in the SI.

We added "...deforestation (captured as deforestation rates linked to land area change, not to total land occupation)..."

- line 86: 'reductions of animal feed grown on arable land' - why does this matter? Need to discuss in introduction.

We thoroughly rewrote the introduction and this should now be clearer.

- line 88: abbreviations (CC) need to be explained where they are first used

Done

- line 93-94: again, so far the reader has no idea what 'high yield gap' and 'low yield gap' means.

This has now been remedied where it is used for the first time, cf. comment above.

- line 104ff: again, the introduction needs to be set up so that the reader knows what type of response variables (i.e. cropland area requirements, N surplus, different environmental impacts) are examined in this analysis as well as some basic background on how these variables were examined. Right now the reader does not know what N surplus is about - Why is this a concern? What broad assumptions/methods were used to estimate this? Why does the N surplus of organic and conventional agriculture differ?

We added an explanation of the N-surplus when first mentioning it on line 155 “(equaling N-inputs minus N outputs, thus acting as a proxy for oversupply of reactive nitrogen to ecosystems and related impacts)”. It differs between the two production systems because the N inputs in the organic systems tend to be lower than in conventional ones, due to the absence of mineral fertilizers, which is only partly compensated by increased legume shares (we made this explicit on lines 154-155).

- line 104-105: instead of saying that the N surplus decreases (difficult to interpret) it would be better to say that a scenario of 100% conversion to organic agriculture leads to a N deficit.

We changed as suggested

- line 110: The exclusion of potential N sources from food waste and human waste needs to be discussed as a caveat of the model somewhere in the paper. Even though nutrients in food waste and human wastes are currently not re-used at very high rates in food production (estimates are around 5-9% in urban areas), some modeling studies have shown high potential recycling rates if urban wastes were used in agriculture (see eg. Faerge et al. 2001, doi:10.1016/S0304-3800(01)00233-2; Magid et al. 2006, doi:10.1016/j.ecoleng.2006.03.009).

We added a sentence on that and two references.

- line 144-145: 'increased protein/calorie ratio' - what does this mean? Please explain.

We explain by adding “(i.e. the share of calories provided via protein)” right after “ratio”.

- lines 142: it needs to be clearly stated how diets were modeled for the reader to be able to understand these results. Were the numbers of calories kept constant in different scenarios? Are different dietary compositions an outcome of the model (given livestock and legume production constraints imposed in the model in different scenarios)? Or were different diet compositions pre-scribed to examine different diet scenarios (e.g. vegan, vegetarian etc.)?

We added some text to clarify that in this assessment, the scenarios provide the same amount of calories as the reference scenario. We also added some explanation on what drives the results – it is the conditions on legumes and assumptions on animal feed, resulting in lower animal numbers and lower animal product supply. It is not driven by any external assumptions on dietary composition such as vegetarian or vegan, etc.

- lines 189: use 'non-synthetic pesticides' instead of 'critical substances' here and justify this statement better (i.e. explaining that organic management allows the use of some pesticides, and that some of these can potentially be harmful to the environment).

Done, we changed and added as suggested.

- line 241ff: too long and convoluted sentence. Rephrase.

We rephrased and broke into several shorter sentences.

- Fig. 1: this Fig. is difficult to interpret given the relatively simple message it is intended to deliver. I would consider showing bars as % change (or increase/decrease in ha cropland use) relative to base year.

Changing as suggested would basically just remove the lower part of the figure, up to the reference values around 1.5 bn. We think that giving the absolute numbers are more informative in this case than the percentage values, and also that it is more informative to give the complete values and not only the difference to the reference scenario. We have therefore opted not to change this figure.

- Fig. 2: in every line of this figure the numbers increase with increasing % organic area, except for the 25% wastage, 50% concentrate and medium climate change impact line (which changes from 4 to 0 and then to 5% area change under 0, 20 and 40% area organic respectively). Is this an error? If not, how do the authors explain this anomaly?

Thank you for pointing this out. It should read “-4” instead of “4”, we corrected this.

- Fig. 3: the colour scheme used here is confusing as both very high numbers and very low numbers are shaded in the same colour. I would use a different colours to denote high N surplus and high N deficit, as these are associated with very different problems (i.e. high N losses to the environment vs N limitation of crop production).

We changed as suggested, using a different colour for N-deficits

- Fig. 5: the line at 100% change is confusing; the axis should be centred around zero % change (i.e. no difference) to allow easier interpretation. Also - I am not sure how to interpret this Figure, see comment in major comments above.

We clarified how the figure is to be read by amending the caption (cf. comment above). In particular, we changed wording from “Percentage change of environmental impacts with respect to [...]” to “Percentage ~~change~~ of environmental impacts with respect to [...]” to make clear that the bars refer to the IMPACTS and not the CHANGE in impacts and that they are displayed as percentage in relation to the reference scenario, which is assigned 100% impact. We therefore did not change the figure by replacing the 100%-line by the 0% line. We changed colors in the figure as suggested by reviewer 2.

- Fig. 5: Why is energy use examined separately from GHG emissions? Isn't energy use (e.g. for fertilizer production) already included in GHG emissions (see lines 217ff in SI)? Does energy use imply any other additional environmental impact that is separate from GHG emissions?

Energy use is a small part of GHG emission only, with GHG emissions mainly driven by enteric fermentation (CH₄), fertilized soils (N₂O) and manure management (CH₄, N₂O). It thus provides additional information on a specific part whose GHG emissions can be mitigated by specific measures relating to the energy system; the indicator is non-renewable energy-use, as common in LCA assessments.

- SI, line 70: what is a 'yield gap' for livestock production? Does this mean lower livestock productivity (e.g. in milk yields) or lower livestock densities? Explain.

We added the following paragraph to the SI: “For milk and eggs, the yield gap refers to output per animal per year; for meat, the yield gap refers to slaughter weight. In organic

systems, often the same slaughter weight as in conventional systems is reached, but after a longer time than in conventional systems. In our model, this is treated equivalently (lower number of meat animals with same yield and higher number of animals with lower yields)."

- SI, line 76: How is N fixation modelled in SOL-m? This is a variable with high uncertainty and should be discussed further. Also - it seems that the SOL-m model assumes that the additional legume crops grown under organic management are used for human consumption and contribute to calorie delivery? Is that true? Many legume crops in organic systems are, however, not harvested (for human consumption or animal feed) but only used as green manures that are ploughed into the soil at the end of the cropping season and provide fertility for the following crop. How does the SOL-m model deal with this issue?

We added the following to the SI: "Legume-crop specific N-fixation rates were used⁵⁷. The composition of legumes cropped in the organic system was chosen to reflect the share between different legume crops as reported in the reference scenario. For the organic system, this rather over-estimates the relative share of food legumes with respect to green manure." The reference used (57) is Herridge et al. 2008, Global inputs of biological nitrogen fixation in agricultural systems, Plant Soil 311:1-18.

- SI, line 152ff: How does this assumption of grassland area remaining constant with the scenarios of changes in concentrate feed? If livestock is fed less on concentrate feed, does this not necessarily imply an expansion of pastureland?

We made the assumption of constant grassland areas in order to make our scenarios more comparable to the reference scenario. Reducing concentrate feed increases the share of grass in the feeding rations of ruminants, but there need not be a total expansion of grassland, as we do not have an assumption that animal numbers must stay constant.

- SI, line 156ff: it is not clear to me from this description how N inputs to organic or conventional agriculture were estimated. Was any data on N application in organic versus conventional systems used? Or was N input based on estimated crop N demands?

N-inputs are derived based on available N, assigned to the various crops in relation to their relative N-demand as a share of total N-demand of all crops. We added this explanation to the SI.

- SI, lines 165-166: please show this comparison with literature values in a Table

We added these numbers in a table in this section 1.5.2

- SI, line 212ff: include a table showing the parameters used for different emission factors, as well as a table showing the final emission values for different scenarios. It is not clear here where the parameters used for organic management come from. Did the authors assume similar emission factors for enteric fermentation for organically and conventionally managed livestock? Were emissions from processes and buildings (assuming this includes manure storage?) assumed to be the same between organic and conventional management?

We assumed similar emission factors for enteric fermentation for organically and conventionally managed livestock, and that emissions from processes and buildings (including manure storage) were the same between organic and conventional management. We clarified this in the revised SI (line 243 ff).

- SI, line 217: abbreviations (GWP) need to be explained where they are first used; also include units for numbers shown.

Done

- SI, line 224: show this comparison of baseline year GHG emissions from the SOL-m model with literature estimates in a table.

We added these numbers in a table towards the end of section 1.5.5

- SI, line 227ff: it is not clear from this description how water use between organic and conventional management was calculated.

We added the following for clarification “We assumed similar irrigation values per ton of irrigated production for organic and conventional production. Differences between the systems then arise due to different yields and different area shares for different crops with different crop-specific irrigation values as reported in AQUASTAT.”

- SI, line 230ff: please provide more information on the model used to estimate pesticide use, including the equations used. Also please provide the final values of pesticide use (maybe for different regions) in a Table, and ideally compare with values from the literature.

We added this information to the SI.

- SI, line 256ff: it is not clear how soil erosion differs between organic and conventional scenarios? Does the difference only depend on the cropland area occupied?

It depends on the differences in areas and on the differences in shares of crops with different susceptibility for erosion in organic and conventional systems (organic systems have more legumes, for example). We added this clarification in the revised SI.

Reviewers' comments:

Reviewer #1 (Remarks to the Author):

The paper and supplement has (also thanks to the remarks of Reviewer 2) much improved and is now ready to be accepted for publication with 2 minor revisions.

1) The text on line 88-90 in the article-pdf w/o track change should be substituted by the proposed formulation in the rebuttal-pdf:

"Our analysis shows the necessary global changes, but we emphasize that structural change in the food system and the implementation of organic agriculture on any path to such increased shares of organic production will differ regionally and need to account for local and regional characteristics." This formulation, which I think is much better, has not found its way into the final text.

2) The language should be polished and some grammar errors corrected.

Reviewer #2 (Remarks to the Author):

First, my apologies to the journal editors and authors of this paper for the length of time it has taken me to review the revised version of the manuscript. It is a busy time of year.

I have reviewed the full list of original reviewers comments and the response by the authors and have read the revised manuscript carefully.

For my part, I am quite satisfied that the authors have addressed both the more major issues that I raised in my original review along with the minor editorial and clarity issues I identified. Overall the new revised manuscript is far more transparent and clear regarding the study's objectives, methods and insights that can be made. The authors are to be commended. The revisions were a big piece of work.

I have however, identified a number of small issues in the revised manuscript that I think need to be addressed before the paper can be accepted. These are all of an editorial nature and should be easily addressed (in my below, all line references are to the clean revised manuscript version, not the version in track changes):

1. Line 21. Is the 'or' (i.e. "N-surplus or GHG emissions") truly an or in that either N-surplus can be improved OR GHG emissions can be improved but not both or is it an and?
2. Line 21. Should GHG be spelled out here?
3. Lines 59-61. The sentence that begins "This allows us to ..." is unclear and in particular the second part of it. Also note that it has a spare period at the end.
4. Line 115. To help the reader, can you indicate what the 5% of increased land demand is relative to? Is it the reference year land area or the base year? I'm pretty sure that it's the reference year but it will help the reader understand the constraint you're applying.
5. Re Fig 2 data reported. Have a look at the data reported in the line associated with zero Climate Change impact, 50% reduction in waste and 50% reduction in Concentrates. The values go from -25, -23, -29(!), -14, -9, -4. Please review the actual values from you models and I think that the increased value associated with 40% adoption of organic production is an error.
6. Line 142. The phrase 'specific nutrient recycling at the basis' is unclear.
7. Line 154. The Sentence that starts: 'Not utilizing waste as a fertilizer...' could be made clearer as here, I think the waste you are referring to is food and human waste as per line 145. This is the first and only time though that you're referring to the use of human waste in the paper (all other uses of the word waste are related to food losses). As such, I think it might be useful to insert the words 'food and human' before waste on line 154
8. Line 157. The sentence that starts on this line is, I think part of the paragraph above.
9. Line 185. The sentence that ends '...level of 10% recommended in33.'" If really awkward. Better to include the authors name in the text along these lines: '... recommended in Smith and

Jones33.'

10. Line 293. I don't think that the sentence that starts: 'This would, in particular, ...' is a new paragraph but is part of the above one.

That's it. Good luck.

Reviewer #3 (Remarks to the Author):

Thank you very much for revising your article. In your revisions you addressed some of the issues I had with the previous version of your manuscript. By re-writing the article and phrasing your conclusions in a more careful and nuanced manner you, for example, do not anymore provide conclusions that are not justified by the results of your study, as you previously did.

But I still do not recommend this paper for publication in the current format due to two main issues – (1) lack of minimal description of the working and key assumptions of the model, which makes it difficult for the reader to interpret the results provided, (2) the lack of a clear storyline answering the question posed in the title of the article.

Major issue 1

For one, it is still very difficult for the reader to understand what you did and to interpret your results. I personally now have a better understanding of what exactly you modelled, how the model works and thus how to interpret the results shown. But it has taken me a lot of time and effort to get here. You should make it possible for the reader of the article to easily understand your methodological approach and to easily be able to interpret the results presented – especially for a broad-impact journal like Nature Communications. This is not currently the case. I therefore suggest further re-writing of the manuscript to include a clearer storyline. One thing that is, for example, missing is a clear statement of what the model does and what basic assumptions the model is based upon. Without understanding at least the minimal workings of the model used in the analysis, and the key assumptions underlying it, the reader does not know how to interpret the results presented (without diving into the details provided in the SM). This is particularly important given that a lot of the results of the current analysis are based on certain assumptions (e.g. on the impacts of organic on yields, on soil erosion or on GHG emissions) that are highly debated. As the cited review by Meiers et al. (and others) have highlighted – there is still very high uncertainty on the impacts of organic agriculture on variables like yields in developing countries, soil erosion, N leaching or N₂O/CH₄ emissions. Given this high uncertainty the conclusions of your analysis entirely depend on the assumptions made in the model. Therefore these assumptions need to be very clearly stated in the article (and not hidden somewhere in the SM).

As an example: from the key panel A in Fig. 5 it seems that a world with 100% organic leads to (1) more land use, (2) higher deforestation, (3) basically equal GHG emissions, (4) lower N surplus, (5) slightly lower P surplus, (6) same water use, (7) lower energy use (although shouldn't this already be included in GHG emissions?), (8) higher erosion and (9) lower pesticide use.

These to me seem to be the first key results of your study. Now from these results the question arises why this is so. But given that the reader does not know on what assumptions your model is based (without diving deep into the SM) it is difficult to understand why we see these differences (or lack of differences), for example, in erosion, GHG emissions, water use. So many questions emerge from this that are not answered in the paper (although I guess that the answers are hidden somewhere in the SM). Do GHG emissions include emissions from land use change? How does land use differ from deforestation? Where does the considerable additional land area used in the different scenarios come from, if not from forests? Does N loss refer purely to NO₃ leaching or

does it also include N₂O, NH₃ and NO_x emissions? All of these questions are essential to be able to adequately interpret the results of the analysis presented here. I already asked a lot of these questions in my first review. And while you provided answers to these questions in your response to reviewers, I want to see these same answers in the article itself to allow every reader to make sense of the results presented. So please include a brief but clear description of the model used and of the key assumptions of the model that are relevant for the interpretation of results in the main paper. This discussion of key assumptions of the model does not have to happen in a traditional methods section (as the journal probably does not require such a section) but these key assumptions needed to interpret results should be mentioned when discussing the results (as you have, for example, already done for soil erosion).

Another issue that shows the importance of clearly describing the model used is the question of animal feed concentrate scenarios. As I pointed out in my previous review, it appears counter-intuitive that a reduction in animal feed concentrates leads to an decrease in land use, as one would think that if animals are fed less grains they will require more grassland (which would lead to an increase in land area required). The assumption that livestock populations and consumption of animal products are not held constant but can be reduced in scenarios with lower animal feed concentrates is a key (I repeat key!) assumption of your model that needs to be clearly stated directly in the article in order to allow the reader to interpret the results. I see that you added a sentence saying that the animal feed scenarios are associated with a reduction in animal numbers (line 128-129), but this sentence should be even more clear about the fact that livestock and livestock consumption are not held constant in the model.

Major issue 2

Secondly, the answer to the key question of the paper – i.e. whether organic agriculture would provide a more sustainable means of producing food in the future (at least this is what your title suggests) – is too hidden in the figures presented. The figures you use include far too many panels/cells that are not relevant to this core question of the paper.

Fig. 4 is also almost entirely irrelevant to the key question of the manuscript, i.e. the question about organic agriculture. This figure – as well as the rather detailed discussion of implications of different scenarios for dietary composition - could easily be moved to the SM. Instead the space could be used for a more in-depth discussion of the environmental impacts of organic scenarios and the assumptions underlying these results (see comments above).

As a reader I read your paper because I am looking for an answer to the question about organic agriculture (given the title of your paper). As a reader I am (given the way that you lay out your paper) not very interested in the scenarios on waste reduction or changes in animal concentrate feed. Do not get me wrong – both of these are very relevant and interesting topics. But the way that you have set up the paper so far you focus on the question of organic agriculture. This question should thus be at the centre of the discussion and of the presentation of results.

Lets take Fig. 5 as an example (which provides the most comprehensive overview of the results): in 8 out of 9 panels of Fig. 5 you provide answers to the question of how changes in food wastage and livestock concentrate feed would impact the sustainability of our food system. The only panel that really matters to the question of whether organic agriculture would be more sustainable is panel A (i.e. the right panel in the first row). What I would suggest is making panel A of Fig. 5 (i.e. scenarios of 0% and 100% organic under 0% food waste reduction and 0% change in concentrate feed) into a separate figure that is easier to read than the current display. This is the key figure of the paper as it actually answers the question of how the world would change under an entirely organic scenario.

I also must say that I am still confused about how to read Fig. 5. My difficulty in doing so shows

that this Figure is not chosen well in conveying the conclusions from this study. In your reply to my question you talk about blue bars. I do not see any blue bars in the Figure. Do you mean the grey bars? In addition there is some confusion regarding the concentrate feed use panel titles. In the legend you write that concentrate feed use changes from 0% to 100% from left to right. But the panel titles say 100% to 0% from right to left. Also: are these %concentrate feed scenarios (i.e. 100% means that all animal feed comes from concentrate?) or are these changes in %concentrate feed scenarios (i.e. 100% means that there is a 100% increase in concentrate feed use compared to the reference scenario)?

I do understand now that the light yellow bars in panel A appear to represent the reference scenario (i.e. 0% change in food waste, animal concentrate feed & organic agriculture). But given that the comparison of light yellow and dark yellow bars in this graph is the key result that answers the main question of the paper, this result should be highlighted more and depicted in a manner that is easier to understand.

After the question of the impact of a 100% organic world on the sustainability of the food production system, the second key question of your paper is whether organic agriculture could be combined with other food system strategies (eg reduction in animal concentrate feed and food waste) to create a more sustainable food system. I am assuming that this is the secondary research question as the title focuses solely on the question of organic agriculture without mentioning other food system strategies. Even though most of the figures focus mostly on this second question, so maybe you want to make this the core question of the paper (which would require changing the title)?

But if indeed this is the secondary question of your paper, then you should have a second Figure examining this question and providing an overview of the different combinations of scenarios used (as you are currently doing in Figs 2, 3 and 5). I would suggest retaining Figs 2 and 3, as they nicely show the impacts of different combinations of scenarios on two key variables. While I would leave out the other panels in Fig. 5 (except for panel A, which should be made – in a modified form that is easier to interpret – into the main figure of the paper, see comments above). There are tons of other interesting questions that could be examined with Fig. 5 (eg relating to the environmental impact of changes in animal diets) but those are not the topic of the current paper and should thus not distract from the main comparison (i.e. the comparison of organic scenarios).

My bottom-line is that I would suggest going back to the drawing board and thinking about how to simplify and mainstream the key messages of the paper, revisiting the figures used and re-writing the storyline of the paper more clearly focusing on clearly defined research questions.

I generally have support for this research and I think this analysis provides an interesting contribution to the literature. This analysis does not only provide the most comprehensive model of a world under organic agriculture that I have seen to date, but it also makes important conclusions on the importance of combining different food system strategies to move towards enhanced sustainability.

Minor comments:

I see that you changed the term 'concentrate feed' to 'food-competing feed' in some places in the manuscript. If you use this new term you need to consistent use it throughout the manuscript and figures.

Please check spelling in the manuscript, eg:

Line 106: 'Such structural changes may be regionally different and needs to account for local and regional characteristics.'

Authors' responses to reviewer comments, second revision:

1 Reviewer #1 (Remarks to the Author):

Dear reviewer,

*Thank you very much for your comments and suggestions on the revised version of the manuscript. Please find below our answers to your comments, each time in **italic blue** print. The main revised manuscript and SI files are cleaned versions without track changes. For completeness, we have also uploaded the versions of the manuscript and the SI with track changes, to make all changes easily traceable (some minor issues such as punctuation and spelling errors, etc. are not track-changed).*

Reviewers' comments:

The paper and supplement has (also thanks to the remarks of Reviewer 2) much improved and is now ready to be accepted for publication with 2 minor revisions.

Thank you for this overall positive judgment.

1) The text on line 88-90 in the article-pdf w/o track change should be substituted by the proposed formulation in the rebuttal-pdf:

“Our analysis shows the necessary global changes, but we emphasize that structural change in the food system and the implementation of organic agriculture on any path to such increased shares of organic production will differ regionally and need to account for local and regional characteristics.”

This formulation, which I think is much better, has not found its way into the final text.

Thank you for pointing this out; we replaced the text as suggested, with some additional amendments in the formulation.

2) The language should be polished and some grammar errors corrected.

Two native speakers (co-author Pete Smith and Robert Home, who is not a co-author) did a final proof read of the manuscript.

2 Reviewer #2 (Remarks to the Author):

Dear reviewer,

*Thank you very much for your comments and suggestions on the revised version of the manuscript. Please find below our answers to your comments, each time in **italic blue print**. The main revised manuscript and SI files are cleaned versions without track changes. For completeness, we have also uploaded the versions of the manuscript and the SI with track changes, to make all changes easily traceable (some minor issues such as punctuation and spelling errors, etc. are not track-changed).*

Reviewers' comments:

First, my apologies to the journal editors and authors of this paper for the length of time it has taken me to review the revised version of the manuscript. It is a busy time of year.

I have reviewed the full list of original reviewers comments and the response by the authors and have read the revised manuscript carefully.

For my part, I am quite satisfied that the authors have addressed both the more major issues that I raised in my original review along with the minor editorial and clarity issues I identified. Overall the new revised manuscript is far more transparent and clear regarding the study's objectives, methods and insights that can be made. The authors are to be commended. The revisions were a big piece of work.

Thank you for your positive judgment on this revised version.

I have however, identified a number of small issues in the revised manuscript that I think need to be addressed before the paper can be accepted. These are all of an editorial nature and should be easily addressed (in my below, all line references are to the clean revised manuscript version, not the version in track changes):

1. Line 21. Is the 'or' (i.e. "N-surplus or GHG emissions") truly an or in that either N-surplus can be improved OR GHG emissions can be improved but not both or is it an and?

It is an "and" – we changed accordingly.

2. Line 21. Should GHG be spelled out here?

Yes, it may be better to write it out in the abstract. We changed this.

3. Lines 59-61. The sentence that begins "This allows us to ..." is unclear and in particular the second part of it. Also note that it has a spare period at the end.

We changed to "We thus assess the contexts in terms of complementary food system changes in which a conversion to organic agriculture may contribute to more sustainable food systems." and removed the spare period at the end.

4. Line 115. To help the reader, can you indicate what the 5% of increased land demand is relative to? Is it the reference year land area or the base year? I'm pretty sure that it's the

reference year but it will help the reader understand the constraint you're applying.
It is the reference year; We made this explicit by adding "...of the land demand in the reference scenario" to the 5%.

5. Re Fig 2 data reported. Have a look at the data reported in the line associated with zero Climate Change impact, 50% reduction in waste and 50% reduction in Concentrates. The values go from -25, -23, -29(!), -14, -9, -4. Please review the actual values from you models and I think that the increased value associated with 40% adoption of organic production is an error.

Indeed, this is an error, thank you for pointing this out. It should read "-19", not "-29". We changed this in the figure (we did replace the figure without track change mode, thus only the new figure is displayed, not marked as being changed).

6. Line 142. The phrase 'specific nutrient recycling at the basis' is unclear.

We rephrased for clarification and it reads now as follows: "...in the context of where nutrients are sourced and how they are recycled in organic agriculture."

7. Line 154. The Sentence that starts: 'Not utilizing waste as a fertilizer...' could be made clearer as here, I think the waste you are referring to is food and human waste as per line 145. This is the first and only time though that you're referring to the use of human waste in the paper (all other uses of the word waste are related to food losses). As such, I think it might be useful to insert the words 'food and human' before waste on line 154

We changed as suggested and added "food and human" before "waste".

8. Line 157. The sentence that starts on this line is, I think part of the paragraph above.

True, we changed this (also without track change, it is then more evident).

9. Line 185. The sentence that ends '...level of 10% recommended in³³.' If really awkward. Better to include the authors name in the text along these lines: '... recommended in Smith and Jones³³.'

We changed accordingly to "...recommended by the Food and Nutrition Board of the US National Academy of Sciences³³." The full name of the institution would even be longer (Food and Nutrition Board of the Institute of Medicine of the US National Academy of Sciences), but we think it is more readable and not problematic to omit the explicit reference to the institute here.

10. Line 293. I don't think that the sentence that starts: 'This would, in particular, ...' is a new paragraph but is part of the above one.

True, we dropped starting a new paragraph and included the last part to the previous one (again without track change, it is then more evident).

That's it. Good luck.

3 Reviewer #3 (Remarks to the Author):

Dear reviewer,

Thank you very much for your very helpful and detailed comments and suggestions on the revised version of the paper. Please find below our answers to your comments, each time in italic blue print. Line numbers in our answers refer to the revised manuscript and SI text WITH track changes; the main revised manuscript and SI files are cleaned versions without track changes. For completeness, we have uploaded these two versions of the manuscript and the SI, to make all changes easily traceable (some minor issues such as punctuation and spelling errors, etc. are not track-changed; we neither track-changed the reorganization of sections – the methods section is now before the reference section, as requested by the journal).

Reviewers' comments:

Thank you very much for revising your article. In your revisions you addressed some of the issues I had with the previous version of your manuscript. By re-writing the article and phrasing your conclusions in a more careful and nuanced manner you, for example, do not anymore provide conclusions that are not justified by the results of your study, as you previously did.

Thank you for your generally positive judgment on this revised version, and for the further detailed and very helpful criticism that you provide in the comments below.

But I still do not recommend this paper for publication in the current format due to two main issues – (1) lack of minimal description of the working and key assumptions of the model, which makes it difficult for the reader to interpret the results provided, (2) the lack of a clear storyline answering the question posed in the title of the article.

Major issue 1

For one, it is still very difficult for the reader to understand what you did and to interpret your results. I personally now have a better understanding of what exactly you modelled, how the model works and thus how to interpret the results shown. But it has taken me a lot of time and effort to get here. You should make it possible for the reader of the article to easily understand your methodological approach and to easily be able to interpret the results presented – especially for a broad-impact journal like Nature Communications. This is not currently the case. I therefore suggest further re-writing of the manuscript to include a clearer storyline. One thing that is, for example, missing is a clear statement of what the model does and what basic assumptions the model is based upon. Without understanding at least the minimal workings of the model used in the analysis, and the key assumptions underlying it, the reader does not know how to interpret the results presented (without diving into the details provided in the SM). This is particularly important given that a lot of the results of the current analysis are based on certain assumptions (e.g. on the impacts of organic on yields, on soil erosion or on GHG emissions) that are highly debated. As the

cited review by Meiers et al. (and others) have highlighted – there is still very high uncertainty on the impacts of organic agriculture on variables like yields in developing countries, soil erosion, N leaching or N₂O/CH₄ emissions. Given this high uncertainty the conclusions of your analysis entirely depend on the assumptions made in the model. Therefore these assumptions need to be very clearly stated in the article (and not hidden somewhere in the SM).

We added further information on the assumptions in the methods section towards the end of the main text. Due to the limited number of words for the main paper, we cannot add much detail there, but we provide the most important information on the assumptions the model is based on, in particular for organic production. Furthermore, we added some information on the assumptions to the results section, as suggested further down in the referee's comment.

As an example: from the key panel A in Fig. 5 it seems that a world with 100% organic leads to (1) more land use, (2) higher deforestation, (3) basically equal GHG emissions, (4) lower N surplus, (5) slightly lower P surplus, (6) same water use, (7) lower energy use (although shouldn't this already be included in GHG emissions?), (8) higher erosion and (9) lower pesticide use.

These to me seem to be the first key results of your study. Now from these results the question arises why this is so. But given that the reader does not know on what assumptions your model is based (without diving deep into the SM) it is difficult to understand why we see these differences (or lack of differences), for example, in erosion, GHG emissions, water use. So many questions emerge from this that are not answered in the paper (although I guess that the answers are hidden somewhere in the SM). Do GHG emissions include emissions from land use change? How does land use differ from deforestation? Where does the considerable additional land area used in the different scenarios come from, if not from forests? Does N loss refer purely to NO₃ leaching or does it also include N₂O, NH₃ and NO_x emissions? All of these questions are essential to be able to adequately interpret the results of the analysis presented here. I already asked a lot of these questions in my first review. And while you provided answers to these questions in your response to reviewers, I want to see these same answers in the article itself to allow every reader to make sense of the results presented. So please include a brief but clear description of the model used and of the key assumptions of the model that are relevant for the interpretation of results in the main paper. This discussion of key assumptions of the model does not have to happen in a traditional methods section (as the journal probably does not require such a section) but these key assumptions needed to interpret results should be mentioned when discussing the results (as you have, for example, already done for soil erosion).

As suggested, we added short descriptions of the key assumptions in lines 263-293 directly with the results, where such details on assumptions were not previously presented. We also added further information on the assumptions in the methods section towards the end of the paper (lines 403-431), following the structure required by the journal, which suggests such separate results and methods sections. Nitrogen, land use and deforestation are discussed separately further up in the main text, and we added further information on the assumptions also there (lines 126-135; 163-167; 177-180). We emphasized that the N-surplus covers all relevant flows and compounds, i.e. N in inputs and product and residue output, etc. as well as emissions, volatilization and leaching of NO₃, N₂O, NH₃.

Energy use is reported separately from GHG emissions using the life cycle impact assessment method "Cumulative Energy Demand (CED)" which is reported as MJ. Furthermore, GHG emissions from energy use are included in total GHG emissions (but comprise only a small

fraction of those). We report both indicators GHG emissions (in CO₂-eq) and energy use (in MJ) as they address different aspects of sustainability.

GHG emissions are reported including land use change emissions, namely those from deforestation and management of organic soils (lines 278-279). GHG emissions do not include emissions from changing from grasslands to cropland, as those are set equal zero, because no reliable data on such change was available and we decided to keep grassland areas constant in all scenarios, thus having land use change between forest and cropland only.

We further added explanations on the land use and deforestation variables used in this paper and how they differ, cf. lines 126-135 in the main paper (“Deforestation is modelled as the pressure on forests from increased land demand, assuming the same relative deforestation rates, i.e. ha deforested per ha cropland increase, in each country as reported in the baseline. This likely underestimates deforestation impacts for larger cropland increases, given that additional cropland will largely be sourced from forests, as grasslands are assumed to stay constant. A more detailed assessment of the deforestation dynamics with increased cropland demand would necessitate combination with data on the suitability of grassland, forest and other areas for crop production of various kinds, which is beyond the scope of this paper. Thus, the land occupation and deforestation indicators as used here serve to assess the pressure on land areas and forests that may arise from the dynamics captured in the different scenarios.”) and lines 325-331 in the SI (“Modelling deforestation in this way thus captures the pressure of land increase on forests in countries, where deforestation is an issue, by assuming a similar dynamics as in the baseline. It thus complements the land occupation variable that captures the land demand, irrespective of where it may be sourced from in a specific country. In particular for larger increases in land occupation, this approach may rather underestimate deforestation, as a large part of this additional land likely would have to be sourced from forests, given the assumption of constant grassland areas.”).

Another issue that shows the importance of clearly describing the model used is the question of animal feed concentrate scenarios. As I pointed out in my previous review, it appears counter-intuitive that a reduction in animal feed concentrates leads to an decrease in land use, as one would think that if animals are fed less grains they will require more grassland (which would lead to an increase in land area required). The assumption that livestock populations and consumption of animal products are not held constant but can be reduced in scenarios with lower animal feed concentrates is a key (I repeat key!) assumption of your model that needs to be clearly stated directly in the article in order to allow the reader to interpret the results. I see that you added a sentence saying that the animal feed scenarios are associated with a reduction in animal numbers (line 128-129), but this sentence should be even more clear about the fact that livestock and livestock consumption are not held constant in the model.

We added these assumptions on lines 163-167 to make it fully clear what happens with grassland and cropland areas, livestock numbers, production and animal product consumption.

Major issue 2

Secondly, the answer to the key question of the paper – i.e. whether organic agriculture would provide a more sustainable means of producing food in the future (at least this is what your title suggests) – is too hidden in the figures presented. The figures you use include far too

many panels/cells that are not relevant to this core question of the paper.

Fig. 4 is also almost entirely irrelevant to the key question of the manuscript, i.e. the question about organic agriculture. This figure – as well as the rather detailed discussion of implications of different scenarios for dietary composition - could easily be moved to the SM. Instead the space could be used for a more in-depth discussion of the environmental impacts of organic scenarios and the assumptions underlying these results (see comments above).

Although the focus of the general discussion on potentials and challenges of switching to organic production largely is on environmental issues, we think that potential impacts on dietary patterns should be addressed as well, thus contributing to a food system assessment rather than a mere production system assessment. We thus opt to retain this section, while still adding the methodological details to the paper, as suggested in the various reviewer comments above (cf. above). In particular, figure 4b on the scenarios with reduced food-competing feed and the corresponding discussion could be moved to the SI, if needed due to space restrictions, but we would prefer to keep this in the main paper.

As a reader I read your paper because I am looking for an answer to the question about organic agriculture (given the title of your paper). As a reader I am (given the way that you lay out your paper) not very interested in the scenarios on waste reduction or changes in animal concentrate feed. Do not get me wrong – both of these are very relevant and interesting topics. But the way that you have set up the paper so far you focus on the question of organic agriculture. This question should thus be at the centre of the discussion and of the presentation of results.

Indeed, the feasibility of organic farming is the major focus of our study, and we first want to analyse the impacts of a conversion to organic agriculture without complementary measures. As a second focus of our study, we then investigate complementary measures in combination with which organic farming would become feasible in environmental terms. We revised the paper accordingly in order to more explicitly present the conclusions on the feasibility of organic farming, without and with complementary measures (cf. also comments below).

Lets take Fig. 5 as an example (which provides the most comprehensive overview of the results): in 8 out of 9 panels of Fig. 5 you provide answers to the question of how changes in food wastage and livestock concentrate feed would impact the sustainability of our food system. The only panel that really matters to the question of whether organic agriculture would be more sustainable is panel A (i.e. the right panel in the first row). What I would suggest is making panel A of Fig. 5 (i.e. scenarios of 0% and 100% organic under 0% food waste reduction and 0% change in concentrate feed) into a separate figure that is easier to read than the current display. This is the key figure of the paper as it actually answers the question of how the world would change under an entirely organic scenario.

We added the upper left panel of the former figure 5 (this former figure 5 is now figure 6) as a separate figure (the new figure 5), and the results presented in the section on environmental impacts mainly refer to this (cf. also comments below).

I also must say that I am still confused about how to read Fig. 5. My difficulty in doing so shows that this Figure is not chosen well in conveying the conclusions from this study. In your reply to my question you talk about blue bars. I do not see any blue bars in the Figure. Do you mean the grey bars? In addition there is some confusion regarding the concentrate feed use panel titles. In the legend you write that concentrate feed use changes from 0% to 100% from left to right. But the panel titles say 100% to 0% from right to left. Also: are these %concentrate feed scenarios (i.e. 100% means that all animal feed comes from concentrate?)

or are these changes in %concentrate feed scenarios (i.e. 100% means that there is a 100% increase in concentrate feed use compared to the reference scenario?)?

We beg your pardon – mentioning “blue” was a mistake caused by referring to the old layout of the graph. This is now changed in the current version as we also choose another layout for this figure (cf. comment below). Regarding the 0% and 100% concentrate feed: we reformulated somewhat in the legend to further clarify that it goes from the original level of food-competing feed (FCF), i.e. 0% reduction in FCF, on the left, to no FCF, i.e. a 100% reduction in FCF, to the right.

I do understand now that the light yellow bars in panel A appear to represent the reference scenario (i.e. 0% change in food waste, animal concentrate feed & organic agriculture). But given that the comparison of light yellow and dark yellow bars in this graph is the key result that answers the main question of the paper, this result should be highlighted more and depicted in a manner that is easier to understand.

We changed the layout of the graph to spider graphs for easier interpretation and understanding.

After the question of the impact of a 100% organic world on the sustainability of the food production system, the second key question of your paper is whether organic agriculture could be combined with other food system strategies (eg reduction in animal concentrate feed and food waste) to create a more sustainable food system. I am assuming that this is the secondary research question as the title focuses solely on the question of organic agriculture without mentioning other food system strategies. Even though most of the figures focus mostly on this second question, so maybe you want to make this the core question of the paper (which would require changing the title)?

We keep the conversion to organic production as the first main question, and then address how FCF reduction and wastage reduction may contribute to sustainable food systems as complementary strategies as a second question. We reformulated the main paper accordingly, in particular in the introduction (lines 60-77), and also implemented your suggestion regarding figure 5; cf. also the next comments.

But if indeed this is the secondary question of your paper, then you should have a second Figure examining this question and providing an overview of the different combinations of scenarios used (as you are currently doing in Figs 2, 3 and 5). I would suggest retaining Figs 2 and 3, as they nicely show the impacts of different combinations of scenarios on two key variables. While I would leave out the other panels in Fig. 5 (except for panel A, which should be made – in a modified form that is easier to interpret – into the main figure of the paper, see comments above). There are tons of other interesting questions that could be examined with Fig. 5 (eg relating to the environmental impact of changes in animal diets) but those are not the topic of the current paper and should thus not distract from the main comparison (i.e. the comparison of organic scenarios).

We added the upper left panel of figure 5 as a separate figure (the new figure 5), and the results presented in the section on environmental impacts mainly refer to this. We retained the other panels as an additional figure 6, though, to illustrate the effect of the combination of strategies. The interest is on the performance of organic agriculture, but another key aspect is the potential of the combination of organic production with other food systems strategies to contribute to sustainable food systems. While starting with the question related to organic agriculture, we conclude with the discussion of the systemic view on the potential of a range of food system strategies, and of which role organic agriculture may play in such a combination. The results clearly show that a conversion to organic agriculture without

complementary measures would have a number of adverse impacts (besides a number of positive ones).

My bottom-line is that I would suggest going back to the drawing board and thinking about how to simplify and mainstream the key messages of the paper, revisiting the figures used and re-writing the storyline of the paper more clearly focusing on clearly defined research questions.

We reformulated the text to simplify and clarify the key messages and to clearly tell the story of a conversion to organic agriculture as our first research question, subsequently complemented with two additional strategies for increasing sustainability in food systems. We also revisited the figures to reflect this, in particular by providing the results on environmental impacts of a 100% conversion to organic agriculture without complementary strategies as a separate figure. For further details, cf. the comments above.

I generally have support for this research and I think this analysis provides an interesting contribution to the literature. This analysis does not only provide the most comprehensive model of a world under organic agriculture that I have seen to date, but it also makes important conclusions on the importance of combining different food system strategies to move towards enhanced sustainability.

Thank you again for your generally positive judgment on this revised version,

Minor comments:

I see that you changed the term ‘concentrate feed’ to ‘food-competing feed’ in some places in the manuscript. If you use this new term you need to consistent use it throughout the manuscript and figures.

We changed this where it had not yet been changed, i.e. in figures 3 and 5 (and the new figure 6) in the main body of the text and in figures 1-5, 7, 8-11 in the Supporting Information (and in the respective captions, where needed)), and a few places in the SI text. Replacement of figures has been undertaken without “track change”-mode, but changed wording in the figure captions and the text are displayed with “track change”-mode.

Please check spelling in the manuscript, eg:

Line 106: ‘Such structural changes may be regionally different and needs to account for local and regional characteristics.’

Two native speakers (co-author Pete Smith and Robert Home, who is not a co-author) did a final proof read of the manuscript.

Reviewers' comments:

Reviewer #3 (Remarks to the Author):

Thank you very much for your thorough revision of your paper. In your reviews you have addressed most of my concerns. I have a couple of additional small remarks below that should be addressed before publication.

Line 58: 'complement this conversion'; rephrase to 'complement this scenario of organic conversion'

Line 86ff: This paragraph does not quite interpret the key results accurately and objectively enough but tries to portray the results too much in favour of organic. I would suggest rewording: "Our results show that adoption of organic agriculture by itself increases land demand with respect to conventional production (with resulting negative impacts on biodiversity and greenhouse-gas emissions), but it has advantages in terms of other indicators, such as reduced nitrogen surplus, and pesticide use. But when combined with complementary changes in the global food system, namely changed feeding rations, and correspondingly reduced animal numbers, and changed wastage patterns, organic agriculture can contribute to feeding more than 9 billion people in 20150, and do so sustainably. Such a combination of strategies can deliver adequate global food availability, with positive outcomes across all assessed environmental indicators, including cropland area demand."

Line 115-116: Where does this baseline relationship between cropland expansion and deforestation come from? What type of deforestation data is used? The wording here is not quite clear.

Line 113-122: This paragraph could be shortened & some of the details only mentioned in the methods or SM. Only mentioning the key assumptions and key difference between land use and deforestation is enough here.

Lines 161ff: I very much like the re-written section on N-surplus. Very clear and well-discussed.

Line 208-214: There is some repetition in these 3 sentences. Could be shortened.

Line 227ff: important point.

Figs. 4a & b: I still think that these figures are not essential to the main paper. These figures basically show (1) adoption of organic does not change diet composition much (not surprising given the assumptions of the model), and (2) adoption of FCF scenarios leads to huge changes in diet composition (also not really surprising given the assumptions of the model). These conclusions are already discussed in the text, and do not necessarily require two additional large figures, in my opinion (particularly as the paper already contains 6 large figures). If the authors want to retain the figures, I would suggest to combine both Fig. 4a and 4b into a single panel, and simply add 1 more bar showing composition under 0% FCF and 100% organic to the graph 4b.

Line 257: GHG emissions from LUC (which is mostly related to agriculture) account for ca. 12% of all anthropogenic GHG emissions, while agricultural emissions from farm management account for another ca. 14%. How is it possible that an increase in cropland area of 16-33% leads to a reduction of GHG emissions under the 100% organic scenario of 3-7%? Given that GHG emissions from deforestation for agriculture are about the same magnitude as emissions from agricultural management, GHG emissions from management would need to be decreased by 19-40% to result in a net decrease of 3-7% under the organic scenario. But existing quantitative reviews of the primary literature suggest that N₂O emissions, for example, are only reduced by ca. 14-30% per unit area (depending on the study, i.e. Skinner et al. 2014,

<http://dx.doi.org/10.1016/j.scitotenv.2013.08.098> ; and Tuomisto et al. 2012, <http://dx.doi.org/10.1016/j.jenvman.2012.08.018>), but they might actually increase per unit product (ibid). While CH₄ emissions from rice paddies might actually increase per unit area and per unit product (according to Skinner et al.; a study which includes some of the same authors as this paper). So I do not understand how the model comes to the conclusion of reduced GHG emissions under organic management given the larger land use requirements, as well as the relatively small difference in management GHG emissions observed in experimental studies (and as e.g. synthesizes by Skinner et al). Please discuss this point in more detail and discuss the results on GHG emissions in relation to current GHG emissions from land use for agriculture and agricultural management, as well as existing evidence on GHG emissions from organic management.

Fig. 6: These new figures are much easier to interpret. There are, however, too many different panels, which means the reader is overwhelmed by too much information. I would suggest reducing the number of panels and only showing scenarios of 0% and 100% of each scenario (and remove the panels showing the 0% implementation of each scenario).

Lines 325-330: good discussion

Line 413: this does not represent a sensitivity analysis, as a sensitivity analysis is related to assessing the uncertainty in a model, and the sensitivity of model results to model input data and parameters.

Authors' responses to reviewer comments, third revision:

Reviewer #3 (Remarks to the Author):

Dear reviewer,

*Thank you very much once more for your additional helpful and detailed comments and suggestions on the second revision version of the paper. As in the earlier rounds, please find below our answers to your comments, each time in **italic blue** print. Line numbers in our answers refer to the revised manuscript and SI text with track changes; given the few changes in this third round, we uploaded the version with track changes only, both of the manuscript and the SI.*

Reviewers' comments:

Thank you very much for your thorough revision of your paper. In your reviews you have addressed most of my concerns. I have a couple of additional small remarks below that should be addressed before publication.

Thank you for this overall positive judgment.

Line 58: 'complement this conversion'; rephrase to 'complement this scenario of organic conversion'

We changed as suggested.

Line 86ff: This paragraph does not quite interpret the key results accurately and objectively enough but tries to portray the results too much in favour of organic. I would suggest rewording: "Our results show that adoption of organic agriculture by itself increases land demand with respect to conventional production (with resulting negative impacts on biodiversity and greenhouse-gas emissions), but it has advantages in terms of other indicators, such as reduced nitrogen surplus, and pesticide use. But when combined with complementary changes in the global food system, namely changed feeding rations, and correspondingly reduced animal numbers, and changed wastage patterns, organic agriculture can contribute to feeding more than 9 billion people in 20150, and do so sustainably. Such a combination of strategies can deliver adequate global food availability, with positive outcomes across all assessed environmental indicators, including cropland area demand."

We changed as suggested, besides that we dropped the bracketed remark "(with resulting negative impacts on biodiversity and greenhouse-gas emissions)" as we do not address biodiversity explicitly and as the GHG-effects are complex and interlinked with the N-aspects, so that we think that it is better not to mention them here very briefly, but only further down in more detail (also taking up this new issue raised by the reviewer in a remark further down and newly included in the text: the interlinkage of N-Surplus and GHG emissions).

Line 115-116: Where does this baseline relationship between cropland expansion and deforestation come from? What type of deforestation data is used? The wording here is not

quite clear.

It is data from FAOSTAT on deforestation values, which we then put in relation to cropland area expansion or total cropland areas, if no expansion has been reported in FAOSTAT. We now mention in the text that the data is sourced from FAOSTAT and added a specific reference to the supplementary information, section 1.5.8., where this is described in more detail.

Line 113-122: This paragraph could be shortened & some of the details only mentioned in the methods or SM. Only mentioning the key assumptions and key difference between land use and deforestation is enough here.

Done - we shortened by moving one sentence providing some further details to section 1.5.8 in the SI.

Lines 161ff: I very much like the re-written section on N-surplus. Very clear and well-discussed.

Thank you.

Line 208-214: There is some repetition in these 3 sentences. Could be shortened.

Done – we shortened this part.

Line 227ff: important point.

Thank you.

Figs. 4a & b: I still think that these figures are not essential to the main paper. These figures basically show (1) adoption of organic does not change diet composition much (not surprising given the assumptions of the model), and (2) adoption of FCF scenarios leads to huge changes in diet composition (also not really surprising given the assumptions of the model). These conclusions are already discussed in the text, and do not necessarily require two additional large figures, in my opinion (particularly as the paper already contains 6 large figures). If the authors want to retain the figures, I would suggest to combine both Fig. 4a and 4b into a single panel, and simply add 1 more bar showing composition under 0% FCF and 100% organic to the graph 4b.

We dropped this figure in the main text as suggested, and moved it to the supplementary information, adjusting the references to the figures accordingly (in the SI, we did it without track changes, as no change in contents was involved).

Line 257: GHG emissions from LUC (which is mostly related to agriculture) account for ca. 12% of all anthropogenic GHG emissions, while agricultural emissions from farm management account for another ca. 14%. How is it possible that an increase in cropland area of 16-33% leads to a reduction of GHG emissions under the 100% organic scenario of 3-7%? Given that GHG emissions from deforestation for agriculture are about the same magnitude as emissions from agricultural management, GHG emissions from management would need to be decreased by 19-40% to result in a net decrease of 3-7% under the organic scenario. But existing quantitative reviews of the primary literature suggest that N₂O emissions, for example, are only reduced by ca. 14-30% per unit area (depending on the study, i.e. Skinner et al. 2014, <http://dx.doi.org/10.1016/j.scitotenv.2013.08.098> ; and Tuomisto et al. 2012, <http://dx.doi.org/10.1016/j.jenvman.2012.08.018>), but they might actually increase per unit product

(ibid). While CH₄ emissions from rice paddies might actually increase per unit area and per unit product (according to Skinner et al.; a study which includes some of the same authors as this paper). So I do not understand how the model comes to the conclusion of reduced GHG

emissions under organic management given the larger land use requirements, as well as the relatively small difference in management GHG emissions observed in experimental studies (and as e.g. synthesizes by Skinner et al). Please discuss this point in more detail and discuss the results on GHG emissions in relation to current GHG emissions from land use for agriculture and agricultural management, as well as existing evidence on GHG emissions from organic management.

Thank you for highlighting this important aspect. Under 100% conversion to organic agriculture, the emissions from the livestock part and methane from rice slightly increase in our model, while emissions from fertilized soils drop considerably and the emissions from synthetic fertilizer production that also contribute significantly drop to zero. In sum, this indeed offsets increased emissions due to higher land use.

We have to emphasize, however, that this strongly links to the low N input levels in the 100% conversion scenarios, where synthetic fertilizers are avoided and only partly replaced by increased fixation from increased legume shares. When looking at the N-balance, it can be seen that this is in a critically low range for the 100% conversion to organic agriculture. It is important to highlight this interaction with GWP as the reduction in GWP would be less, or even slightly reversed, if higher N inputs would be provided to avoid these critically low N levels. We addressed the critically low N levels as a particular challenge for such conversion to organic agriculture when discussing the N-surplus in the paper, but we did not establish this important link to the GHG emissions, which we suggest to add to the paper as phrased below.

It is also important to relate to the emissions data as reported in Skinner et al. (2014), for example, as the reviewer suggests, which we did not yet do in the paper. We thus also suggest to add this as phrased below. We did not use the numbers reported in this literature, as we decided to calculate according to the IPCC guidelines, thus in particular using the emission factors reported there (e.g. for N₂O emissions per kg N applied to soils). We thus, in particular, use identical emission factors for both organic and conventional production and for all fertilizer types. Whether these emission factors may differ between N from different organic and mineral fertilizers is subject to ongoing research. However, we think that there is not yet enough evidence to choose different factors in such a global food systems model; hence our decision to rely on the IPCC guidelines for these calculations.

Interestingly, the difference in average total N input levels between organic and conventional trials reported in Skinner et al. (2014) is almost 35%, thus reflecting a similar difference in soil-borne emissions if the same IPCC factor was applied. The observed difference in emissions according to Skinner et al. (2014) is then lower, as the organic systems show somewhat higher emission factors per ton N applied than the conventional ones. On the other hand, the total N input seems a much less important determinant for the soil borne emissions for organic systems than for conventional ones. This may challenge the approach of strict proportionality of emissions to N inputs as suggested by the IPCC guidelines. To avoid these complexities and as the evidence base is still quite weak, we decided to use the classical IPCC approach.

We thus suggest amending the text by adding the following on line 260. It is rather long, but we think this is an important point and thus decided to fully add it to the main text, not moving part of it to the SI:

New text (new part in green): “[...]. This is mainly due to the generally lower nitrogen fertilization levels (no mineral fertilizers) with corresponding lower emissions from fertilizer

application in organic production (cf. above). It is thus important to emphasize that any increase in N supply to address these critically low N levels in organic agriculture would correspondingly increase N₂O-emissions from fertilizer applications. It would thus lessen the reduction in GHG emissions or even change it to a zero or slightly increasing effect. We also emphasize that these emissions calculations follow the IPCC guidelines and do not refer to recent meta-studies on emission factors³². Skinner et al.³² find rather higher emission factors for organic than for conventional production. On the other hand, they find that total N inputs are only a weak determinant for total emissions for organic production while they are a good determinant for conventional systems. However, evidence is not yet robust enough to deviate from the classical IPCC approach in such a global food systems model. We thus do not use adapted emission factors for different production systems and types of fertilizers and do not challenge the proportionality to inputs for organic production. A relatively small part of the difference in GHG emissions again reflects the difference in energy use. [...]

Fig. 6: These new figures are much easier to interpret. There are, however, too many different panels, which means the reader is overwhelmed by too much information. I would suggest reducing the number of panels and only showing scenarios of 0% and 100% of each scenario (and remove the panels showing the 0% implementation of each scenario).

Done, we reduced the panels showing the intermediate scenarios with 50% reduction in FCF and 25% food wastage reduction, thus keeping 4 of the original 9 panels only. We moved the original larger figure including the intermediate scenarios to the supplementary information and display it there (without tracked changes, as no change in contents was involved), making a note on this in the figure caption in the main text.

Lines 325-330: good discussion

Thank you.

Line 413: this does not represent a sensitivity analysis, as a sensitivity analysis is related to assessing the uncertainty in a model, and the sensitivity of model results to model input data and parameters.

Thank you for pointing this out. We accordingly changed the wording to “We assessed how the model results may change due to different assumptions regarding climate change impacts on yields [...]” (the old formulation being “We conduct sensitivity analyses regarding climate change impacts on yields [...]”).

REVIEWERS' COMMENTS:

Reviewer #3 (Remarks to the Author):

The authors have thoroughly addressed my remaining concerns and I can now recommend publication of this paper. I want to emphasize that I think this is a very important piece of work, as it represents the first global model that examines the potential influence of organic management in such a comprehensive manner, including such a wide range of indicators. All the issues I had were purely related to the delivery and interpretation of the results. I still have a few minor suggestions for small additional changes below, but these are at the author's discretion.

I would appreciate if the authors could mention in the section where they discuss the GHG emissions under the organic scenario (i.e. line 264ff) what they wrote in their reply to my comment in their Response to Reviewers, i.e. that under the organic scenario the increased GHG emissions from additional land use and deforestation (as well as the slightly increased emissions from livestock and rice production) are offset by reduced emissions from fertilized soils due to lower N application rates, as well as reduced emissions from synthetic fertilizer production. Currently the authors do not, I believe, mention anywhere in the manuscript that CO₂ emissions from land use and deforestation are increased under the organic scenario.

Secondly, I would like the authors to mention the potential negative impact of increased deforestation rates on biodiversity. The authors do not mention potential implications of their results for biodiversity anywhere in the paper. But as the authors themselves state in the introduction (line 31), biodiversity loss is an important sustainability issue that is driven by agricultural land use. Please add a few sentences on this issue as related to the results from this analysis. This can also include a discussion of the potentially positive impacts of reduced pesticide use and water pollution from N runoff on biodiversity.

Also note that in the figure caption to Fig. 4 and 5 you refer to the yellow lines showing the organic scenario as "bright brown". Also in Figure caption 5, I would suggest deleting the sentence in lines 315-317, as this is already highlighted by the panel titles.

I think that some of the figure captions are not very clear. Figures should be as self-explanatory as possible and allow a reader to understand what the figures show, without having to read the full article. But the reader will, for example, not know what the 'reference scenario' refers to if they do not read the full article. I would therefore suggest to clearly state in each Figure caption what the reference scenarios is, and also for what year the results are (2005? 2050?). In addition, the first sentence of the Figure caption should provide a brief summary of the figure. But Fig. 4 and 5 have, for example, the same first sentence in the figure caption.

I would therefore suggest, for example, changing the figure caption of Fig. 4 to something like "Year 2050 environmental impacts of different organic scenarios. Organic scenarios (yellow lines) are shown relative to the reference scenario (i.e. conventional agriculture, blue lines), with (dotted lines) and without (solid lines) impacts of climate change."

While Fig. 5 could read something like: "Year 2050 environmental impacts of different organic (yellow lines) and conventional (blue lines) scenarios with concomitant changes in livestock feed and food waste strategies. All scenarios are shown relative to the reference scenario (i.e. conventional agriculture, no changes in livestock feed and food waste; dark grey line), with (dotted lines) and without (solid lines) impacts of climate change."

Authors' responses to reviewer comments, final revision:

Reviewer #3 (Remarks to the Author):

Dear reviewer,

*Thank you very much for your additional helpful comments and for your recommendation to now accept the paper for publication. As in the earlier rounds, please find below our answers to your comments, each time in **italic blue** print. As before, line numbers in our answers refer to the revised manuscript and SI text with track changes.*

Reviewers' comments:

The authors have thoroughly addressed my remaining concerns and I can now recommend publication of this paper. I want to emphasize that I think this is a very important piece of work, as it represents the first global model that examines the potential influence of organic management in such a comprehensive manner, including such a wide range of indicators. All the issues I had were purely related to the delivery and interpretation of the results. I still have a few minor suggestions for small additional changes below, but these are at the author's discretion.

Thank you for this positive judgement. Your critical reading over the various review rounds has substantially improved the manuscript and its presentation – thank you very much for all the effort you put into reviewing it in such detail and that thoroughly!

I would appreciate if the authors could mention in the section where they discuss the GHG emissions under the organic scenario (i.e. line 264ff) what they wrote in their reply to my comment in their Response to Reviewers, i.e. that under the organic scenario the increased GHG emissions from additional land use and deforestation (as well as the slightly increased emissions from livestock and rice production) are offset by reduced emissions from fertilized soils due to lower N application rates, as well as reduced emissions from synthetic fertilizer production. Currently the authors do not, I believe, mention anywhere in the manuscript that CO₂ emissions from land use and deforestation are increased under the organic scenario. *We added these explanations to the manuscript on lines 237-241.*

Secondly, I would like the authors to mention the potential negative impact of increased deforestation rates on biodiversity. The authors do not mention potential implications of their results for biodiversity anywhere in the paper. But as the authors themselves state in the introduction (line 31), biodiversity loss is an important sustainability issue that is driven by agricultural land use. Please add a few sentences on this issue as related to the results from this analysis. This can also include a discussion of the potentially positive impacts of reduced pesticide use and water pollution from N runoff on biodiversity.

We added a short paragraph on this at the end of the results section, lines 277-285: “We modelled a range of key environmental indicators, but we did not model impacts on biodiversity, given the complexity and – for many indicators – inadequacy to capture such in

a global model. However, when linking to impacts that correlate with biodiversity, some indications for impacts on biodiversity can be given: Increased area use and deforestation under organic agriculture rather increase pressure on biodiversity, while the reduced pesticide use and nitrogen surplus reduce this pressure. Less ambiguity is again reached when combining conversion to organic agriculture with the other two food systems strategies, resulting in overall reduction of all environmental impacts including area use, and thus suggesting a general reduction of pressure on biodiversity under these combined scenarios.”

Also note that in the figure caption to Fig. 4 and 5 you refer to the yellow lines showing the organic scenario as “bright brown”. Also in Figure caption 5, I would suggest deleting the sentence in lines 315-317, as this is already highlighted by the panel titles.

We adjusted from “bright brown” to “yellow” and deleted part of this sentence to reduce redundancy with the information given in the figure. Due to a comment from the editor referring to journal formatting requirements, we also reorganized the figure and changed the titles to labels for rows and columns of the 4 panels displayed.

I think that some of the figure captions are not very clear. Figures should be as self-explanatory as possible and allow a reader to understand what the figures show, without having to read the full article. But the reader will, for example, not know what the ‘reference scenario’ refers to if they do not read the full article. I would therefore suggest to clearly state in each Figure caption what the reference scenario is, and also for what year the results are (2005? 2050?). In addition, the first sentence of the Figure caption should provide a brief summary of the figure. But Fig. 4 and 5 have, for example, the same first sentence in the figure caption.

I would therefore suggest, for example, changing the figure caption of Fig. 4 to something like “Year 2050 environmental impacts of different organic scenarios. Organic scenarios (yellow lines) are shown relative to the reference scenario (i.e. conventional agriculture, blue lines), with (dotted lines) and without (solid lines) impacts of climate change.”

While Fig. 5 could read something like: “Year 2050 environmental impacts of different organic (yellow lines) and conventional (blue lines) scenarios with concomitant changes in livestock feed and food waste strategies. All scenarios are shown relative to the reference scenario (i.e. conventional agriculture, no changes in livestock feed and food waste; dark grey line), with (dotted lines) and without (solid lines) impacts of climate change.”

We adjusted as suggested; we also added short titles to the figures, thus providing brief summaries of what the figures are about. This has also been an editorial requirement.